# Accelerating Diffusion Models with Parallel Sampling: Inference at Sub-Linear Time Complexity

**Haoxuan Chen**[*]
ICME
Stanford University
haoxuanc@stanford.edu

**Yinuo Ren**[*†]
ICME
Stanford University
yinuoren@stanford.edu

**Lexing Ying**
Department of Mathematics and ICME
Stanford University
lexing@stanford.edu

**Grant M. Rotskoff**
Department of Chemistry and ICME
Stanford University
rotskoff@stanford.edu

## Abstract

Diffusion models have become a leading method for generative modeling of both image and scientific data. As these models are costly to train and *evaluate*, reducing the inference cost for diffusion models remains a major goal. Inspired by the recent empirical success in accelerating diffusion models via the parallel sampling technique [1], we propose to divide the sampling process into $\mathcal{O}(1)$ blocks with parallelizable Picard iterations within each block. Rigorous theoretical analysis reveals that our algorithm achieves $\widetilde{\mathcal{O}}(\text{poly} \log d)$ overall time complexity, marking *the first implementation with provable sub-linear complexity w.r.t. the data dimension $d$*. Our analysis is based on a generalized version of Girsanov's theorem and is compatible with both the SDE and probability flow ODE implementations. Our results shed light on the potential of fast and efficient sampling of high-dimensional data on fast-evolving modern large-memory GPU clusters.

## 1 Introduction

Diffusion and probability flow based models [2–11] are now state-of-the-art in many fields, such as computer vision and image generation [12–22], natural language processing [23, 24], audio and video generation [25–29], optimization [30, 31], sampling and learning of fixed classes of distributions [32–41], solving high-dimensional partial differential equations [42–46], and more recently several applications in physical, chemical and biological fields [47–63]. For a more comprehensive list of related work, one may refer to the following review papers [64–66]. While there are already many variants, such as denoising diffusion probabilistic models (DDPMs) [7], score-based generative models (SGMs) [9], diffusion schrödinger bridges [67], stochastic interpolants and flow matching [2–4], *etc.*, the recurring idea is to design a stochastic process that interpolates between the data distribution and some simple distribution, along which *score functions* or alike are learned by neural network-based estimators, and then perform inference guided by the learned score functions.

Due to the sequential nature of the sampling process, the inference of high-quality samples from diffusion models often requires a large number of iterations and, thus, evaluations of the neural network-based score function, which can be computationally expensive [68]. Efforts have been

---

[*]Equal contribution, alphabetical order.
[†]Corresponding author.

38th Conference on Neural Information Processing Systems (NeurIPS 2024).

| Work | Implementation | Measure | Approx. Time Complexity |
|:---:|:---:|:---:|:---:|
| [100, Theorem 2] | SDE | $\mathrm{TV}(p_0, \widehat{q}_T)^2$ | $\widetilde{\mathcal{O}}(d\delta^{-1})$ |
| [104, Theorem 2] | SDE | $D_{\mathrm{KL}}(p_\eta \| \widehat{q}_{T-\eta})$ | $\widetilde{\mathcal{O}}(d^2\delta^{-2})$ |
| [107, Corollary 1] | SDE | $D_{\mathrm{KL}}(p_\eta \| \widehat{q}_{T-\eta})$ | $\widetilde{\mathcal{O}}(d\delta^{-2})$ |
| [111, Theorem 3] | ODE w/UMLC correction | $\mathrm{TV}(p_\eta, \widehat{q}_{T-\eta})^2$ | $\widetilde{\mathcal{O}}(\sqrt{d}\delta^{-1})$ |
| **Theorem 3.3** | **SDE w/parallel sampling** | $D_{\mathrm{KL}}(p_\eta \| \widehat{q}_{T-\eta})$ | $\widetilde{\mathcal{O}}(\mathrm{poly}\log(d\delta^{-2}))$ |
| **Theorem 3.5** | **ODE w/parallel sampling** | $\mathrm{TV}(p_\eta, \widehat{q}_{T-\eta})^2$ | $\widetilde{\mathcal{O}}(\mathrm{poly}\log(d\delta^{-2}))$ |

Table 1: Comparison of the approximate time complexity (*cf.* Definition 2.1) of different implementations of diffusion models. $\eta$ is a small parameter that controls the smooth approximation of the data distribution (*cf.* Section 3.1.1).

made to accelerate this process by resorting to higher-order or randomized numerical schemes [69–79], augmented dynamics [80], adaptive step sizes [81], operator learning [82], restart sampling [83], self-consistency [84–87] and knowledge distillation [88–90]. Recently, several empirical works [1, 91–94] leverage the Picard iteration and triangular Anderson acceleration to parallelize the sampling procedure of diffusion models and achieve empirical success in large-scale image generation tasks. Some other recent work [95, 96] also combine the parallel sampling technique with the randomized midpoint method [97] to accelerate the inference of diffusion models.

This efficiency issue is closely related to the problem of bounding the required number of steps and evaluations of score functions to approximate an arbitrary data distribution on $\mathbb{R}^d$ to $\delta$-accuracy, which has been analyzed extensively in the literature [98–115]. In terms of the dependency on the dimension $d$, the current state-of-the-art result for the SDE implementation of diffusion models is $\widetilde{\mathcal{O}}(d)$ [107], improved from the previous $\widetilde{\mathcal{O}}(d^2)$ bound [104]. [111] gives a $\widetilde{\mathcal{O}}(\sqrt{d})$ bound for the probability flow ODE implementation by considering a predictor-corrector scheme with the underdamped Langevin Monte Carlo (UMLC) algorithm.

In this work, we aim to provide parallelization strategies, rigorous analysis, and theoretical guarantees for accelerating the inference process of diffusion models. The time complexity of previous implementations of diffusion models has been largely hindered by the discretization error, which requires the step size to scale with $\widetilde{\mathcal{O}}(1/d)$ for the SDE implementation and $\widetilde{\mathcal{O}}(1/\sqrt{d})$ for the probability flow ODE implementation. We show that the inference process can be first divided into $\mathcal{O}(1)$ blocks with parallelizable evaluations of the score function within each, and thus reduce the overall time complexity to $\widetilde{\mathcal{O}}(\mathrm{poly}\log d)$. We provide **the first implementation of diffusion models with poly-logarithmic complexity**, a significant improvement over the current state-of-the-art polynomial results that sheds light on the potential fast and efficient sampling of high-dimensional distributions with diffusion models on fast-developing memory-efficient modern GPU clusters.

## 1.1 Contributions

- We propose parallelized inference algorithms for diffusion models in both the SDE and probability flow ODE implementations (PIADM-SDE/ODE) with exponential integrators, a shrinking step size scheme towards the data end, and the early stopping technique;

- We provide a rigorous convergence analysis of PIADM-SDE, showing that our parallelization strategy yields a diffusion model with $\widetilde{\mathcal{O}}(\mathrm{poly}\log d)$ approximate time complexity;

- We show that our strategy is also compatible with the probability flow ODE implementation, and PIADM-ODE could improve the space complexity from $\widetilde{\mathcal{O}}(d^2)$ to $\widetilde{\mathcal{O}}(d^{3/2})$ while maintaining the poly-logarithmic time complexity.

## 2 Preliminaries

In this section, we briefly recapitulate the framework of score-based diffusion models, define notations, and discuss related work.

## 2.1 Diffusion Models

In score-based diffusion models, one considers a diffusion process $(\boldsymbol{x}_s)_{s\geq 0}$ in $\mathbb{R}^d$ governed by the following stochastic differential equation (SDE):

$$\mathrm{d}\boldsymbol{x}_s = \boldsymbol{\beta}_s(\boldsymbol{x}_s)\mathrm{d}s + \boldsymbol{\sigma}_s\mathrm{d}\boldsymbol{w}_s, \quad \text{with} \quad \boldsymbol{x}_0 \sim p_0, \tag{2.1}$$

where $(\boldsymbol{w}_s)_{s\geq 0}$ is a standard Brownian motion, and $p_0$ is the target distribution that we would like to sample from. The distribution of $\boldsymbol{x}_s$ is denoted by $p_s$. Once the drift $\boldsymbol{\beta}_s(\cdot)$, the diffusion coefficient $\boldsymbol{\sigma}_s$, and a sufficiently large time horizon $T$ are specified, (2.1) also corresponds to a backward process $(\bar{\boldsymbol{x}}_t)_{0\leq t\leq T}$ for another arbitrary diffusion coefficient $(\boldsymbol{v}_s)_{s\geq 0}$ [116]:

$$\mathrm{d}\bar{\boldsymbol{x}}_t = \left[-\bar{\boldsymbol{\beta}}_t(\bar{\boldsymbol{x}}_t) + \frac{\bar{\boldsymbol{\sigma}}_t\bar{\boldsymbol{\sigma}}_t^\top + \bar{\boldsymbol{v}}_t\bar{\boldsymbol{v}}_t^\top}{2}\nabla\log\bar{p}_t(\bar{\boldsymbol{x}}_t)\right]\mathrm{d}t + \bar{\boldsymbol{v}}_t\mathrm{d}\boldsymbol{w}_t, \tag{2.2}$$

where $\bar{*}_t$ denotes $*_{T-t}$, with $\bar{p}_0 = p_T$ and $\bar{p}_T = p_0$.

For notational simplicity, we adopt a simple choice of the drift and the diffusion coefficients in what follows: $\boldsymbol{\beta}_t(\boldsymbol{x}) = -\frac{1}{2}\boldsymbol{x}$, $\boldsymbol{\sigma}_t = \boldsymbol{I}_d$, and $\boldsymbol{v} = \upsilon\boldsymbol{I}_d$, under which (2.1) is an Ornstein-Uhlenbeck (OU) process converging exponentially to its stationary distribution, *i.e.* $p_T \approx \widehat{p}_T := \mathcal{N}(0, \boldsymbol{I}_d)$, and (2.1) and (2.2) reduce to the following form:

$$\mathrm{d}\boldsymbol{x}_s = -\frac{1}{2}\boldsymbol{x}_s\mathrm{d}s + \mathrm{d}\boldsymbol{w}_s, \quad \text{and} \quad \mathrm{d}\bar{\boldsymbol{x}}_t = \left[\frac{1}{2}\bar{\boldsymbol{x}}_t + \frac{1+\upsilon^2}{2}\nabla\log\bar{p}_t(\bar{\boldsymbol{x}}_t)\right]\mathrm{d}t + \upsilon\mathrm{d}\boldsymbol{w}_t. \tag{2.3}$$

In practice, the score function $\nabla\bar{p}_t(\bar{\boldsymbol{x}}_t)$ is often estimated by a neural network (NN) $\boldsymbol{s}_t^\theta(\boldsymbol{x}_t)$, where $\theta$ represents its parameters, by minimizing the denoising score-matching loss [117, 118]:

$$\begin{aligned}
\mathcal{L}(\theta) &:= \mathbb{E}_{\boldsymbol{x}_t\sim p_t}\left[\left\|\nabla\log p_t(\boldsymbol{x}_t) - \boldsymbol{s}_t^\theta(\boldsymbol{x}_t)\right\|^2\right] \\
&= \mathbb{E}_{\boldsymbol{x}_0\sim p_0}\left[\mathbb{E}_{\boldsymbol{x}_t\sim p_{t|0}(\boldsymbol{x}_t|\boldsymbol{x}_0)}\left[\left\|\frac{\boldsymbol{x}_t - \boldsymbol{x}_0 e^{-t/2}}{1 - e^{-t}} - \boldsymbol{s}_t^\theta(\boldsymbol{x}_t)\right\|^2\right]\right],
\end{aligned} \tag{2.4}$$

and the backward process in (2.3) is approximated by the following SDE thereafter:

$$\mathrm{d}\boldsymbol{y}_t = \left[\frac{1}{2}\boldsymbol{y}_t + \frac{1+\upsilon^2}{2}\boldsymbol{s}_t^\theta(\boldsymbol{y}_t)\right]\mathrm{d}t + \upsilon\mathrm{d}\boldsymbol{w}_t, \quad \text{with} \quad \boldsymbol{y}_0 \sim \mathcal{N}(0, \boldsymbol{I}_d). \tag{2.5}$$

**Implementations.** Diffusion models admit multiple *implementations* depending on the choice of the parameter $\upsilon$ in the backward process (2.2). The SDE implementation with $\upsilon = 1$ is widely used in the literature for its simplicity and efficiency [10], while recent studies [111] claim that the probability flow ODE implementation with $\upsilon = 0$ may exhibit better time complexity. We refer to [111, 119] for theoretical and [120, 121] for empirical comparisons of different implementations.

## 2.2 Parallel Sampling

Parallel sampling algorithms have been actively explored in the literature, including the parallel tempering method [122–124] and several recent studies [125–127]. For diffusion models, the idea of parallel sampling is based on the *Picard iteration* [128, 129] for solving nonlinear ODEs. Suppose we have an ODE $\mathrm{d}\boldsymbol{x}_t = \boldsymbol{f}_t(\boldsymbol{x}_t)\mathrm{d}t$ and we would like to solve it for $t \in [0, T]$, then the Picard iteration is defined as follows:

$$\boldsymbol{x}_t^{(0)} \equiv \boldsymbol{x}_0, \quad \text{and} \quad \boldsymbol{x}_t^{(k+1)} := \boldsymbol{x}_0 + \int_0^t \boldsymbol{f}_s(\boldsymbol{x}_s^{(k)})\mathrm{d}s, \quad \text{for } k \in [0:K-1]. \tag{2.6}$$

Under assumptions on the Lipschitz continuity of $\boldsymbol{f}_t$, the Picard iteration converges to the true solution exponentially fast, in the sense that $\||\boldsymbol{x}_t^{(k)} - \boldsymbol{x}_t|\|_{L^\infty([0,T])} \leq \delta$ with $K = \mathcal{O}(\log\delta^{-1})$ iterations. Unlike high-order ODE solvers, the Picard iteration is intrinsically parallelizable: for any $t \in [0, T]$, the computation of $\boldsymbol{x}_t^{(k+1)}$ relies merely on the values of the most recent iteration $\boldsymbol{x}_t^{(k)}$. With sufficient computational sources parallelizing the evaluations of $\boldsymbol{f}$, the computational cost of solving the ODE no longer scales with $T$ but with the number of iterations $K$.

Recently, this idea has been applied to both the Langevin Monte Carlo (LMC) and the underdamped Langevin Monte Carlo (UMLC) contexts [130]. Roughly speaking, it is proposed to simulate the Langevin diffusion process $\mathrm{d}\boldsymbol{x}_t = -\nabla V(\boldsymbol{x}_t)\mathrm{d}t + \mathrm{d}\boldsymbol{w}_t$ with the following iteration resembling (2.6):

$$\boldsymbol{x}_t^{(0)} \equiv \boldsymbol{x}_0, \quad \text{and} \quad \boldsymbol{x}_t^{(k+1)} := \boldsymbol{x}_0 - \int_0^t \nabla V(\boldsymbol{x}_t^{(k)})\mathrm{d}s + \boldsymbol{w}_t, \quad \text{for } k \in [0 : K-1], \qquad (2.7)$$

where all iterations share a common Wiener process $(\boldsymbol{w}_t)_{t \geq 0}$.

It is shown that for well-conditioned log-concave distributions, parallelized LMC would achieve an iteration depth of $K = \widetilde{\mathcal{O}}(\operatorname{poly}\log d)$ that matches the indispensable time horizon $T = \widetilde{\mathcal{O}}(\operatorname{poly}\log d)$ to achieve exponential ergodicity (*cf.* [130, Theorem 13]). This promises a significant speedup in sampling high-dimensional distributions from the standard LMC of $T = \widetilde{\mathcal{O}}(d)$ iterations, hindered by the $o(1/d)$ step size as imposed by the discretization error and now evaded by the parallelization.

### 2.3 Approximate Time Complexity

A similar situation is expected in diffusion models, where the application bottleneck is largely the inference process with sequential iterations and expensive evaluations of the learned score function $\boldsymbol{s}_t^\theta(\cdot)$, which is often parametrized by large-scale NNs. Despite several unavoidable costs involving pre- and post-processing, data storage and retrieval, and arithmetic operations, we define the following notion of the *approximate time complexity* of the inference process of diffusion models:

**Definition 2.1** (Approximate time complexity). *For a specific implementation of diffusion models (2.5), we define the* approximate time complexity *of the sampling process as the number of* unparallelizable *evaluations of the learned NN-based score function* $\boldsymbol{s}_t^\theta(\cdot)$.

This definition coincides with the notion of *the number of steps required to reach a certain accuracy* in [104, 100], *iteration complexity* in [107, 111], *etc.* in the previous theoretical studies of diffusion models. We have adopted this notion in Table 1 for a comparison of the current state-of-the-art results and our bounds in this work. We will use the notion of *space complexity* likewise to denote the approximate required storage during the inference. Trivially, the space complexity of the sequential implementation is $\mathcal{O}(d)$. Should no confusion occur, we omit the dependency of the complexities above on the accuracy threshold $\delta$, *etc.*, during our discussion, as we focus on applications of diffusion models to high-dimensional data distributions, following the standard practice in the literature.

## 3 Main Results

Inspired by the acceleration achieved by the parallel sampling technique in LMC and ULMC, we aim to accommodate parallel sampling into the theoretical analysis framework of diffusion models. The benefit of the parallel sampling technique in this scenario has been recently confirmed by up to $14\times$ acceleration achieved by the ParaDiGMS algorithm [1] and ParaTAA [92], where several practical compromises are made to mitigate GPU memory constraints and theoretical guarantees are still lacking.

In this section, we will propose **P**arallelized **I**nference **A**lgorithms for **D**iffusion **M**odels with both the **SDE** and probability flow **ODE** implementations, namely the **PIADM-SDE** (Algorithm 1) and **PIADM-ODE** (Algorithm 2), and present theoretical guarantees of our algorithms, including the approximate time complexity and space complexity, for both implementations in Section 3.1 and Section 3.2, respectively. Due to the large number of notations used in the presentation, we give an overview of notations in Appendix A.1 for readers' convenience.

### 3.1 SDE Implementation

We first focus on the approximation, parallelization strategies, and error analysis of diffusion models with the SDE implementation, *i.e.* the forward and backward process (2.3) and its approximatation (2.5) with $\upsilon = 1$. We will show that PIADM-SDE *achieves an* $\widetilde{\mathcal{O}}(\operatorname{poly}\log d)$ *approximate time complexity with* $\widetilde{\mathcal{O}}(d^2)$ *space complexity.*

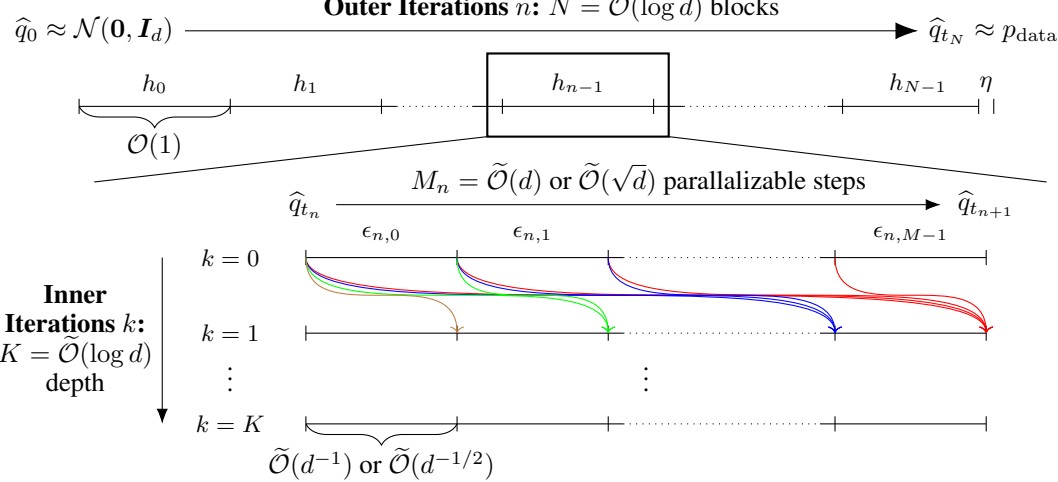

Figure 1: Illustration of PIADM-SDE/ODE. The outer iterations are divided into $\mathcal{O}(\log d)$ blocks of $\mathcal{O}(1)$ length. Within each block, the inner iterations are parallelized with $\widetilde{\mathcal{O}}(d)$ steps for SDE (*cf. Theorem 3.3*), or $\widetilde{\mathcal{O}}(\sqrt{d})$ for probability flow ODE implementation (*cf. Theorem 3.5*). The overall approximate time complexity is $KN = \widetilde{\mathcal{O}}(\text{poly}\log d)$. brown, green, blue, and red curves represent the computation graph at $t = t_n + \tau_{n,m}$ for $m = 1, 2, M_n - 1, M_n$.

### 3.1.1 Algorithm

PIADM-SDE is summarized in Algorithm 1 and illustrated in Figure 1. The main idea behind our algorithm is the fact that (2.5) can be efficiently solved by the Picard iteration within a period of $\mathcal{O}(1)$ length, transferring $\widetilde{\mathcal{O}}(d)$ sequential computations to a parallelizable iteration of depth $\widetilde{\mathcal{O}}(\log d)$. In the following, we introduce the numerical discretization scheme of our algorithm and the implementation of the Picard iteration in detail.

**Step Size Scheme.** In our algorithm, the time horizon $T$ is first segmented into $N$ blocks of length $(h_n)_{n=0}^{N-1}$, with each $h_n \leq h := T/N = \Omega(1)$, forming a grid $(t_n)_{n=0}^{N}$ with $t_n = \sum_{j=1}^{n} h_j$. For any $n \in [0 : N-1]$, the $n$-th block is further discretized into a grid $(\tau_{n,m})_{m=0}^{M_n}$ with $\tau_{n,0} = 0$ and $\tau_{n,M_n} = h_n$. We denote the step size of the $m$-th step in the $n$-th block as $\epsilon_{n,m} = \tau_{n,m+1} - \tau_{n,m}$, and the total number of steps in the $n$-th block as $M_n$.

For the first $N-1$ blocks, we simply use the unique discretization, *i.e.* $h_n = h$, $\epsilon_{n,m} = \epsilon$, and $M_n = M := h/\epsilon$, for $n \in [0 : N-2]$ and $m \in [0 : M-1]$. Following [104, 107], to curb the potential blow-up of the score function as $t \to T$, which is shown by [107] for $0 \leq s < t < T$ to be of the order

$$\mathbb{E}\left[\int_s^t \|\nabla \log \breve{p}_\tau(\breve{\boldsymbol{x}}_\tau) - \nabla \log \breve{p}_s(\breve{\boldsymbol{x}}_s)\|^2 \mathrm{d}\tau\right] \lesssim d\left(\frac{t-s}{T-t}\right)^2,$$

we apply early stopping at time $t_N = T - \eta$, where $\eta$ is chosen in a way such that the $\mathcal{O}(\sqrt{\eta})$ 2-Wasserstein distance between $\breve{p}_T$ and its smoothed version $\breve{p}_{T-\eta}$ that we aim to sample from alternatively, is tolerable for the downstream tasks. We also impose the exponential decay of the step size towards the data end in the last block. To be specific, we let $h_{N-1} = h - \delta$, and discretize the interval $[t_{N-1}, t_N] = [(N-1)h, T - \eta]$ into a grid $(\tau_{N-1,m})_{m=0}^{M_{N-1}}$ with step sizes $(\epsilon_{N-1,m})_{m=0}^{M_{N-1}-1}$ satisfying

$$\epsilon_{N-1,m} \leq \epsilon \wedge \epsilon\left(h - \tau_{N-1,m+1}\right). \tag{3.1}$$

As shown in Lemma B.7, this exponential decaying step size scheme towards the data end is crucial to bound the discretization error in the last block.

For the simplicity of notations, we introduce the following indexing function: for $\tau \in [t_n, t_{n+1}]$, we define $I_n(\tau)$ to be the unique integer such that $\sum_{j=1}^{I_n(\tau)} \epsilon_{n,j} \leq \tau < \sum_{j=1}^{I_n(\tau)+1} \epsilon_{n,j}$. We also define

**Algorithm 1:** PIADM-SDE

---

**Input:** $\widehat{y}_0 \sim \widehat{q}_0 = \mathcal{N}(0, \boldsymbol{I}_d)$, a discretization scheme $(T, (h_n)_{n=1}^N$ and $(\tau_{n,m})_{n\in[1:N], m\in[0:M]})$
satisfying (3.1), the depth of iteration $K$, the learned NN-based score function $\boldsymbol{s}_t^\theta(\cdot)$.
**Output:** A sample $\widehat{y}_{t_N} \sim \widehat{q}_{t_N} \approx \breve{p}_T$.

1 **for** $n = 0$ to $N - 1$ **do**
2     $\widehat{y}_{t_n, \tau_{n,m}}^{(0)} \leftarrow \widehat{y}_{t_n}$, $\boldsymbol{\xi}_m \sim \mathcal{N}(0, \boldsymbol{I}_d)$ for $m \in [0 : M_n]$ *in parallel*;
3     **for** $k = 0$ to $K - 1$ **do**
4         $\widehat{y}_{t_n, 0}^{(k)} \leftarrow \widehat{y}_{t_n}$;
5         **for** $m = 0$ to $M_n$ *in parallel* **do**
6             $\widehat{y}_{t_n, \tau_{n,m}}^{(k+1)} \leftarrow e^{\frac{\tau_{n,m}}{2}} \widehat{y}_{t_n,0}^{(k)}$
                $+ \sum_{j=0}^{m-1} e^{\frac{\tau_{n,m} - \tau_{n,j+1}}{2}} \left[ 2\left(e^{\epsilon_{n,j}} - 1\right) \boldsymbol{s}_{t_n + \tau_{n,j}}^\theta(\widehat{y}_{t_n, \tau_{n,j}}^{(k)}) + \sqrt{e^{\epsilon_{n,j}} - 1} \boldsymbol{\xi}_j \right]$;    (3.4)
7         **end**
8     **end**
9     $\widehat{y}_{t_{n+1}} \leftarrow \widehat{y}_{t_n, \tau_{n, M_n}}^{(K)}$;
10 **end**

---

a piecewise function $g$ such that $g_n(\tau) = \sum_{j=1}^{I_n(\tau)} \epsilon_{n,j}$. It is easy to check that under the uniform discretization for $n \in [1 : N - 1]$, we have $I_n(\tau) = \lfloor \tau/\epsilon \rfloor$ and $g_n(\tau) = \lfloor \tau/\epsilon \rfloor \epsilon$.

**Exponential Integrator.** For each step $\tau \in [t_n + \tau_{n,m}, t_n + \tau_{n,m+1}]$, we use the following exponential integrator scheme [77], as the numerical discretization of the SDE (2.5):

$$\widehat{y}_{t_n, \tau_{n,m+1}} = e^{\epsilon_{n,m}/2} \widehat{y}_{t_n, \tau_{n,m}} + 2\left(e^{\epsilon_{n,m}/2} - 1\right) \boldsymbol{s}_{t_n + \tau_{n,m}}^\theta(\widehat{y}_{t_n + \tau_{n,m}}) + \sqrt{e^{\epsilon_{n,m}} - 1}\boldsymbol{\xi},$$

where $\boldsymbol{\xi} \sim \mathcal{N}(0, \boldsymbol{I}_d)$. Lemma B.3 shows its equivalence to approximating (2.5) as

$$\mathrm{d}\widehat{y}_{t_n, \tau} = \left[\frac{1}{2}\widehat{y}_{t_n, \tau} + \boldsymbol{s}_{t_n + \tau_{n,m}}^\theta(\widehat{y}_{t_n, \tau_{n,m}})\right] \mathrm{d}\tau + \mathrm{d}\boldsymbol{w}_{t_n + \tau}, \quad \text{for } \tau \in [\tau_{n,m}, \tau_{n,m+1}]. \qquad (3.2)$$

**Remark 3.1.** *One could also implement a straightforward Euler-Maruyama scheme instead of the exponential integrator (3.4), where an additional high-order discretization error term would emerge [104, Theorem 1], which we believe would not affect the overall $\widetilde{\mathcal{O}}(\mathrm{poly} \log d)$ time complexity with parallel sampling.*

**Picard Iteration.** Within each block, we apply Picard iteration of depth $K$. As shown by Lemma B.3, the discretized scheme (3.4) implements the following iteration for $k \in [0 : K - 1]$:

$$\mathrm{d}\widehat{y}_{t_n, \tau}^{(k+1)} = \left[\frac{1}{2}\widehat{y}_{t_n, \tau}^{(k+1)} + \boldsymbol{s}_{t_n + g_n(\tau)}^\theta\left(\widehat{y}_{t_n, g_n(\tau)}^{(k)}\right)\right] \mathrm{d}\tau + \mathrm{d}\boldsymbol{w}_{t_n + \tau}, \quad \text{for } \tau \in [0, h_n]. \qquad (3.3)$$

We denote the distribution of $\widehat{y}_{t_n, \tau}^{(K)}$ by $\widehat{q}_{t_n + \tau}$. As proved in Lemma B.6, the iteration above would converge to (3.2) in each block exponentially fast, which given a sufficiently accurate learned score estimation $\boldsymbol{s}_t^\theta$ should be close to the true backward SDE (2.3). One should also notice that the Gaussians $\boldsymbol{\xi}_m$ are only sampled once and used for all iterations.

The parallelization for (3.4) in Algorithm 1 should be understood as that for any $k \in [0 : K - 1]$, each $\boldsymbol{s}_{t_n + \tau_{n,j}}^\theta(\widehat{y}_{t_n, \tau_{n,j}}^{(k)})$ for $j \in [0 : M_n]$ is evaluated in parallel, with subsequent floating-point operations comparably negligible, resulting in the overall $\mathcal{O}(NK)$ approximate time complexity.

### 3.1.2 Assumptions

Our theoretical analysis will be built on the following mild assumptions on the regularity of the data distribution and the numerical properties of the neural networks:

**Assumption 3.1** ($L^2([0, t_N])$ $\delta$-accurate learned score)**.** *The learned NN-based score $s_t^\theta$ is $\delta_2$-accurate in the sense of*

$$\mathbb{E}_{\breve{p}}\left[\sum_{n=0}^{N-1}\sum_{m=0}^{M_n-1}\epsilon_{n,m}\left\|s_{t_n+\tau_{n,m}}^\theta\left(\bar{\boldsymbol{x}}_{t_n+\tau_{n,m}}\right) - \nabla\log\breve{p}_{t_n+\tau_{n,m}}\left(\bar{\boldsymbol{x}}_{t_n+\tau_{n,m}}\right)\right\|^2\right] \le \delta_2^2. \qquad (3.5)$$

**Assumption 3.2** (Regular and normalized data distribution)**.** *The data distribution $p_0$ has finite second moments and is normalized such that $\mathrm{cov}_{p_0}(\boldsymbol{x}_0) = \boldsymbol{I}_d$.*

**Assumption 3.3** (Bounded and Lipschitz learned NN-based score)**.** *The learned NN-based score function $s_t^\theta$ has bounded $C^1$ norm, i.e. $\||\|s_t^\theta(\cdot)\|\|_{L^\infty([0,T])} \le M_{\boldsymbol{s}}$ with Lipschitz constant $L_{\boldsymbol{s}}$.*

**Remark 3.2.** *Assumption 3.1 and the finite moment assumption in Assumption 3.2 are standard assumptions across previous theoretical works on diffusion models [100, 104, 111], while we adopt the normalization Assumption 3.2 from [107] to simplify true score function-related computations (cf. Lemma A.8). Assumption 3.3 can be easily satisfied by truncation, ensuring computational stability. Notice that the exponential integrator, one actually applies Picard iteration to $e^{-t/2}s_t^\theta$, a relaxation of Assumption 3.1 might be possible, which is left for future work.*

### 3.1.3 Theoretical Guarantees

The following theorem summarizes our theoretical analysis for PIADM-SDE (Algorithm 1):

**Theorem 3.3** (Theoretical Guarantees for PIADM-SDE)**.** *Under Assumptions 3.1, 3.2, and 3.3, given the following choices of the order of the parameters*

$$T = \mathcal{O}(\log(d\delta^{-2})), \quad h = \Theta(1), \quad N = \mathcal{O}\left(\log(d\delta^{-2})\right),$$

$$\epsilon = \Theta\left(d^{-1}\delta^2\log^{-1}(d\delta^{-2})\right), \quad M = \mathcal{O}\left(d\delta^{-2}\log(d\delta^{-2})\right), \quad K = \widetilde{\mathcal{O}}(\log(d\delta^{-2})),$$

*and let $L_{\boldsymbol{s}}^2 h_n e^{\frac{7}{2}h_n} \ll 1$, $\delta_2 \lesssim \delta$, $T \lesssim \log\eta^{-1}$, the distribution $\widehat{q}_{t_N}$ that PIADM-SDE (Algorithm 1) generates samples from satisfies the following error bound:*

$$D_{\mathrm{KL}}(p_\eta\|\widehat{q}_{t_N}) \lesssim de^{-T} + d\epsilon T + \delta_2^2 + dTe^{-K} \lesssim \delta^2,$$

*with a total of $KN = \widetilde{\mathcal{O}}\left(\log^2(d\delta^{-2})\right)$ approximate time complexity and $dM = \widetilde{\mathcal{O}}\left(d^2\delta^{-2}\right)$ space complexity for parallelizable $\delta$-accurate score function computations.*

**Remark 3.4.** *We would like to make the following remarks on the result above:*

- *The acceleration from $\widetilde{\mathcal{O}}(d)$ to $\widetilde{\mathcal{O}}(\mathrm{poly}\log d)$ is at the cost of a trade-off with extra memory cost of $M = \widetilde{\mathcal{O}}(d)$ for computing and updating $\{s_{t_n+\tau_{n,j}}^\theta(\widehat{\boldsymbol{y}}_{t_n,\tau_{n,m}}^{(k)})\}_{m\in[0:M_n]}$ simultaneously during each Picard iteration;*

- *Compared with log-concave sampling [130], $M$ being of order $\widetilde{\mathcal{O}}(d)$ instead of $\widetilde{\mathcal{O}}(\sqrt{d})$ therein is partly due to the time independence of the score function $\nabla\log p(\cdot)$ in general sampling tasks. Besides, the scaling $M = \widetilde{\mathcal{O}}(d)$ agrees with the current state-of-the-art dependency [107] for the SDE implementation of diffusion models;*

- *As mentioned above, the scale of the step size $\epsilon$ within one block is still confined to $\Theta(1/M) = \widetilde{\Theta}(1/d)$. The block length $h$, despite being required to be small compared to $1/L_{\boldsymbol{s}}$, is of order $\Theta(1)$, resulting in only $\Theta(\log d)$ blocks and thus $\widetilde{\mathcal{O}}(\mathrm{poly}\log d)$ total iterations.*

### 3.1.4 Proof Sketch

The detailed proof of Theorem 3.3 is deferred to Section B. The pipeline of the proof is to (a) first decompose the error $D_{\mathrm{KL}}(\breve{p}_{t_N}\|\widehat{q}_{t_N})$ into blockwise errors using the chain rule of KL divergence; (b) bound the error in each block by invoking Girsanov's theorem; (c) sum up the errors in all blocks.

The key technical challenge lies in Step (b). Different from all previous theoretical works [100, 104, 111], the Picard iteration in our algorithm generates $K$ paths recursively in each block using the learned score $s_t^\theta$. And therefore the final path $(\widehat{\boldsymbol{y}}_{t_n,\tau}^{(K)})_{\tau\in[0,h_n]}$ depends on all previous paths $(\widehat{\boldsymbol{y}}_{t_n,\tau}^{(k)})_{\tau\in[0,h_n]}$ for $k \in [0:K-1]$, ruling out a direct change of measure argument from the

naïve application of Girsanov's theorem. To this end, we need a more sophisticated mathematical framework of stochastic processes, as given in Appendix A.2. We define the measurable space $(\Omega, \mathcal{F})$ with filtrations $(\mathcal{F}_t)_{t \geq 0}$ to specify the probability measures on $(\Omega, \mathcal{F})$ of each Wiener process, and resort to one of the most general forms of Girsanov's theorem ( [131, Theorem 8.6.6]). For example, in the $n$-th block, we apply the following change of measure procedure:

1. Let $q|_{\mathcal{F}_{t_n}}$ be the measure where $\boldsymbol{w}_t(\omega)$ is the shared Wiener process in the Picard iteration (3.3) for any $k \in [0 : K-1]$;

2. Define another process $\mathrm{d}\widetilde{\boldsymbol{w}}_{t_n+\tau}(\omega) = \mathrm{d}\boldsymbol{w}_{t_n+\tau}(\omega) + \boldsymbol{\delta}_{t_n}(\tau, \omega)\mathrm{d}\tau$, where

$$\boldsymbol{\delta}_{t_n}(\tau, \omega) := \boldsymbol{s}^\theta_{t_n+g_n(\tau)}(\widehat{\boldsymbol{y}}^{(K-1)}_{t_n,g_n(\tau)}(\omega)) - \nabla \log \breve{p}_{t_n+\tau}(\widehat{\boldsymbol{y}}^{(K)}_{t_n+\tau}(\omega));$$

3. Invoke Girsanov's theorem, which yields that the Radon-Nikodym derivative of the measure $\breve{p}|_{\mathcal{F}_{t_n}}$ with respect to $q|_{\mathcal{F}_{t_n}}$ satisfies

$$\log \frac{\mathrm{d}\breve{p}|_{\mathcal{F}_{t_n}}}{\mathrm{d}q|_{\mathcal{F}_{t_n}}}(\omega) = -\int_0^{h_n} \boldsymbol{\delta}_{t_n}(\tau, \omega)^\top \mathrm{d}\boldsymbol{w}_{t_n+\tau}(\omega) - \frac{1}{2}\int_0^{h_n} \|\boldsymbol{\delta}_{t_n}(\tau, \omega)\|^2 \mathrm{d}\tau;$$

4. Conclude that $(\widetilde{\boldsymbol{w}}_{t_n+\tau})_{\tau \geq 0}$ is a Wiener process under the measure $\breve{p}|_{\mathcal{F}_{t_n}}$ and thus (3.3) at iteration $K$ satisfies the following SDE:

$$\mathrm{d}\widehat{\boldsymbol{y}}^{(K)}_{t_n,\tau}(\omega) = \left[\frac{1}{2}\widehat{\boldsymbol{y}}^{(K)}_{t_n,\tau}(\omega) + \nabla \log \breve{p}_{t_n+\tau}\left(\widehat{\boldsymbol{y}}^{(K)}_{t_n,\tau}(\omega)\right)\right] \mathrm{d}\tau + \mathrm{d}\widetilde{\boldsymbol{w}}_{t_n+\tau}(\omega),$$

*i.e.* the true backward SDE (2.3) with the true score function for $\tau \in [t_n, t_{n+1}]$.

One should notice that this change of measure argument will cause an additional term in the bound of the discrepancy between the first iteration $\widehat{\boldsymbol{y}}^{(1)}_{t_n,\tau}$ and the initial condition $\widehat{\boldsymbol{y}}^{(0)}_{t_n,\tau}$ in Lemma B.5. However, due to the exponential convergence of the Picard iteration, this term does not affect the overall error bound.

## 3.2 Probability Flow ODE Implementation

In this section, we will show that our parallelization strategy is also compatible with the probability ODE implementation of diffusion models, *i.e.* the forward and backward process (2.3) and its approximatation (2.5) with $\upsilon = 0$. We will demonstrate that PIADM-ODE (Algorithm 2) further *improves the space complexity from $\widetilde{\mathcal{O}}(d^2)$ to $\widetilde{\mathcal{O}}(d^{3/2})$ while maintaining the same $\widetilde{\mathcal{O}}(\mathrm{poly}\log d)$ approximate time complexity*.

### 3.2.1 Algorithm

Due to the space limit, we refer the readers to Section C.1 and Algorithm 2 for the details of our parallelization of the probability flow ODE formulation of diffusion models. PIADM-ODE keeps the discretization scheme detailed in Section 3.1.1 that divides the time horizon $T$ into $N$ blocks and uses exponential integrators for all updating rules. Notably, PIADM-ODE has the following distinctions compared with PIADM-SDE (Algorithm 1):

- Instead of applying Picard iteration to the backward SDE as in (3.2), we apply Picard iteration to the probability flow ODE as in (C.3) within each block, which does not require sampling i.i.d. Gaussians to simulate a Wiener process;

- The most significant difference is the adoption of an additional *corrector step* [111] after running the probability flow ODE with Picard iteration within one block. During the corrector step, one augments the state space with a Gaussian that represents the initial momentum and then simulates an underdamped Langevin dynamics for $\mathcal{O}(1)$ time with the learned NN-based score function at the time of the block end;

- We then further parallelize the underdamped Langevin dynamics in the corrector step so that it can also be accomplished with $\mathcal{O}(\log d)$ approximate time complexity, as a naïve implementation would result in $\widetilde{\mathcal{O}}(\sqrt{d})$ [130], which is incompatible with our desired poly-logarithmic guarantee.

### 3.2.2 Assumptions

Due to technicalities specific to this implementation, we need first to modify Assumption 3.1 and add assumption on the Lipschitzness of the true score functions $\nabla \log p_t$, which is a common practice in related literature [104, 111]. Recent work on the probability flow ODE implementation [112, 114] also adopts stronger assumptions compared to the SDE implementation.

**Assumption 3.1'** ($L^\infty([0, t_N])$ $\delta$-accurate learned score). *For any $n \in [0 : N-1]$ and $m \in [0 : M_n - 1]$, the learned NN-based score $s_{t_n, \tau_{n,m}}^\theta$ is $\delta_\infty$-accurate in the sense of*

$$\mathbb{E}_{\bar{p}_{t_n+\tau_{n,m}}} \left[ \left\| s_{t_n+\tau_{n,m}}^\theta \left( \bar{\boldsymbol{x}}_{t_n+\tau_{n,m}} \right) - \nabla \log \bar{p}_{t_n+\tau_{n,m}} \left( \bar{\boldsymbol{x}}_{t_n+\tau_{n,m}} \right) \right\|^2 \right] \leq \delta_\infty^2.$$

**Assumption 3.4** (Bounded and Lipschitz true score). *The true score function $\nabla \log p_t$ has bounded $C^1$ norm, i.e. $\| \|\nabla \log p_t(\cdot)\| \|_{L^\infty([0,T])} \leq M_p$ with Lipschitz constant $L_p$.*

Further relaxations on Assumption 3.4 to time-dependent assumptions accommodating the blow-up to the data end (*e.g.* [103, Assumption 1.5]) are left for further work.

### 3.2.3 Theoretical Guarantees

Our results for PIADM-ODE are summarized in the following theorem:

**Theorem 3.5** (Theoretical Guarantees for PIADM-ODE). *Under Assumptions 3.1', 3.2, 3.3, and 3.4, given the following choices of the order of the parameters*

$$T = \mathcal{O}(\log(d\delta^{-2})), \quad h = \Theta(1), \quad N = \mathcal{O}(\log(d\delta^{-2})),$$

$$\epsilon = \Theta\left(d^{-1/2}\delta \log^{-1}(d^{-1/2}\delta^{-1})\right), \quad M = \mathcal{O}(d^{1/2}\delta^{-1}\log(d^{1/2}\delta^{-1})), \quad K = \widetilde{\mathcal{O}}(\log(d\delta^{-2})),$$

*for the outer iteration and*

$$T^\dagger = \mathcal{O}(1) \lesssim L_p^{-1/2} \wedge L_{\boldsymbol{s}}^{-1/2}, \quad h^\dagger = \Theta(1), \quad N^\dagger = \mathcal{O}(1),$$

$$\epsilon^\dagger = \Theta(d^{-1/2}\delta), \quad M^\dagger = \mathcal{O}(d^{1/2}\delta^{-1}), \quad K^\dagger = \mathcal{O}(\log(d\delta^{-2})),$$

*for the inner iteration during the corrector step, and let $L_{\boldsymbol{s}}^2 h^2 e^h \vee L_{\boldsymbol{s}}^2 h^{\dagger 2} e^{h^\dagger}/\gamma \ll 1$, $\delta_\infty \lesssim \delta \log^{-1}(d\delta^{-2})$, and $\gamma \gtrsim L_p^{1/2}$, then the distribution $\widehat{q}_{t_N}$ that PIADM-ODE (Algorithm 2) generates samples from satisfies the following error bound:*

$$\mathrm{TV}(p_\eta, \widehat{q}_{t_N})^2 \lesssim de^{-T} + d\epsilon^2 T^2 + (T^2 + N^2)\delta_\infty^2 + dN^2 e^{-K} \lesssim \delta^2,$$

*with a total of $(K + K^\dagger N^\dagger)N = \widetilde{\mathcal{O}}\left(\log^2(d\delta^{-2})\right)$ approximate time complexity and $d(M \vee M^\dagger) = \widetilde{\Theta}\left(d^{3/2}\delta^{-1}\right)$ space complexity for parallalizable $\delta$-accurate score function computations.*

The reduction of space complexity by the probability flow ODE implementation is intuitively owing to the fact that the probability flow ODE process is a deterministic process in time rather than a stochastic process as in the SDE implementation, getting rid of the $\mathcal{O}(\epsilon)$ term derived by Itô's symmetry. This allows the discretization error to be bounded with $\mathcal{O}(\epsilon^2)$ instead (*cf.* Lemma B.7 and C.5).

### 3.2.4 Proof Sketch

The details of the proof of Theorem 3.5 are provided in Section C. Along with the complexity benefits the deterministic nature of the probability flow ODE may bring, the analysis is technically more involved than that of Theorem 3.3 and requires an intricate interplay between statistical distances. Several major challenges and our corresponding solutions are summarized below:

- The error of the parallelized algorithm within each block may now only be bounded by 2-Wasserstein distance (*cf.* Theorem C.7) instead of any $f$-divergence that admits data processing inequality as in the SDE case by Girsanov's theorem. The additional corrector step exactly handles this issue and would intuitively translate 2-Wasserstein proximity to TV distance proximity (*cf.* Lemma C.18), allowing the decomposition of the overall error into each block;

- For the corrector step, the underdamped Langevin dynamics as a second-order dynamics requires only $\mathcal{O}(\sqrt{d})$ steps to converge, instead of $\mathcal{O}(d)$ steps in its overdamped counterpart. We then adapt the parallelization technique mentioned in Section 2.2 to conclude that it can be accomplished with $\mathcal{O}(\log d)$ approximate time complexity (*cf.* Theorem C.17). The error caused by the approximation to the true score and numerical discretization within this step is bounded in KL divergence by invoking Girsanov's theorem(Theorem A.4) as in the proof of Theorem 3.3;

- Different from the SDE case, where the chain rule of KL divergence can easily decouple the initial distribution and the subsequent dynamics, we need several interpolating processes between the implementation and the true backward process in this case. The final guarantee is in TV distance as it connects with the KL divergence via Pinsker's inequality and admits data processing inequality. We refer the readers to Figure 2 for an overview of the proof pipeline, as well as the notations and intuitions of the auxiliary and interpolating processes appearing in the proof.

## 4    Discussion and Conclusion

In this work, we have proposed novel parallelization strategies for the inference of diffusion models in both the SDE and probability flow ODE implementations. Our algorithms, namely PIADM-SDE and PIADM-ODE, are meticulously designed and rigorously proved to achieve $\widetilde{\mathcal{O}}(\operatorname{poly}\log d)$ approximate time complexity and $\widetilde{\mathcal{O}}(d^2)$ and $\widetilde{\mathcal{O}}(d^{3/2})$ space complexity, respectively, marking the first inference algorithm of diffusion and probability flow based models with sub-linear approximate time complexity. Our algorithm intuitively divides the time horizon into several $\mathcal{O}(1)$ blocks and applies Picard iteration within each block in parallel, transferring the time complexity into space complexity. Our analysis is built on a sophisticated mathematical framework of stochastic processes and provides deeper insights into the mathematical theory of diffusion models.

Our findings echo and corroborate the recent empirical work [1, 91–94] that parallel sampling techniques significantly accelerate the inference process of diffusion models. Theoretical exploration of the adaptive block window scheme therein presents an interesting future research potential. Possible future work also includes the investigation of how to apply our parallelization framework to other variants of diffusion models, such as the discrete [23, 132–142] and multi-marginal [143] formulations. Although we anticipate implementing diffusion models in parallel may introduce engineering challenges, *e.g.* scalability, hardware compatibility, memory bandwidth, *etc.*, we believe that our theoretical contributions lay a solid foundation that not only supports but also motivates the empirical development of parallel inference algorithms for diffusion models since advancements continue in GPU power and memory efficiency.

## Acknowledgments and Disclosure of Funding

LY acknowledges the support of the National Science Foundation under Grant No. DMS-2011699 and DMS-2208163. GMR is supported by a Google Research Scholar Award.

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

# A   Mathematical Background

In this section, we will summarize used notations and rigorous mathematical framework of Itô processes as necessary in the proofs. We will also present several technical lemmas for later reference.

## A.1   Notations

We adopt the following notations throughout the paper:

| Notation | Description |
|---|---|
| $[a:b]$ | The set $\{a, a+1, \ldots, b\}$ |
| $\boldsymbol{I}_d$ | Identity matrix in $\mathbb{R}^{d \times d}$ |
| $\overleftarrow{*}_t$ | $*_{T-t}$ |
| $\widehat{*}$ | Used to denote quantities produced by the algorithm |
| $\widetilde{*}$ | Used to denote quantities related to the auxiliary processes |
| $*^\dagger$ | Used to denote quantities related to the corrector step |
| $\|\cdot\|$ | The Euclidean norm of a vector |
| $\lesssim$ or $\gtrsim$ | The inequality holds up to a constant factor |
| $\ll$ | Absolute continuity (for measures)/ much less than (for quantities) |
| $(\boldsymbol{x}_t)_{t \geq 0}$ | The forward process of the diffusion model (2.3) |
| $(\overleftarrow{\boldsymbol{x}}_t)_{t \in [0,T]}$ | The backward process of the diffusion model (2.3) |
| $(\boldsymbol{y}_t)_{t \in [0,T]}$ | The approximate backward process of the diffusion model (2.5) |
| $\widehat{\boldsymbol{y}}_{t_n, \tau_{n,m}}^{(k)}$ | The approximate value of the approximate process $\boldsymbol{y}_t$ at time $t_n + \tau_{n,m}$ after $k$ iterations in the $(n+1)$-th block |
| $\widehat{\boldsymbol{y}}_{t_n}$ | The value of the approximate process $\boldsymbol{y}_t$ at time $t_n$ |
| $\widehat{q}_{t_n}$ | The distribution of $\widehat{\boldsymbol{y}}_{t_n}$ |
| $\boldsymbol{z}_{t_1:t_2}$ | The path $(\boldsymbol{z}_t)_{t \in [t_1, t_2]}$ of the process $\boldsymbol{z}_t$ |
| $D_f(\cdot \,\|\, \cdot)$ | The $f$-divergence between two distributions |
| $D_{\mathrm{KL}}(\cdot \,\|\, \cdot)$ | The KL divergence between two distributions |
| $TV(\cdot, \cdot)$ | The total variation distance between two distributions |
| $W_2(\cdot, \cdot)$ | The 2-Wasserstein distance between two distributions |

Table 2: Summary of notations

## A.2   Preliminaries

**Theorem A.1** (Properties of $f$-divergence)**.** *Suppose $p$ and $q$ are two probability measures on a common measurable space $(\Omega, \mathcal{F})$ with $p \ll q$. The $f$-divergence between $p$ and $q$ is defined as*

$$D_f(p\|q) = \mathbb{E}_{X \sim q}\left[ f\left( \frac{\mathrm{d}p}{\mathrm{d}q} \right) \right], \tag{A.1}$$

*where $\frac{\mathrm{d}p}{\mathrm{d}q}$ is the Radon-Nikodym derivative of $p$ with respect to $q$, and $f : \mathbb{R}^+ \to \mathbb{R}$ is a convex function. In particular, $D_f(\cdot \,\|\, \cdot)$ coincides with the KullbackLeibler (KL) divergence when $f(x) = x \log x$ and $D_f(\cdot \,\|\, \cdot) = \mathrm{TV}$ coincides with the total variation (TV) distance when $f(x) = \frac{1}{2}|x - 1|$.*

*For the $f$-divergence defined above, we have the following properties:*

1. *(Data-processing inequality). Suppose $\mathcal{H}$ is a sub-$\sigma$-algebra of $\mathcal{F}$, the following inequality holds*
$$D_f\left( p|_{\mathcal{H}} \,\|\, q|_{\mathcal{H}} \right) \leq D_f(p \,\|\, q);$$
*for any $f$-divergence $D_f(\cdot\|\cdot)$.*

2. *(Chain rule). Suppose $X$ is a random variable generating a sub-$\sigma$-algebra $\mathcal{F}_X$ of $\mathcal{F}$, and $p(\cdot|X) \ll q(\cdot|X)$ holds for any value of $X$, then*
$$D_{\mathrm{KL}}(p\|q) = D_{\mathrm{KL}}(p|_{\mathcal{F}_X}\|q|_{\mathcal{F}_X}) + \mathbb{E}_{\mathcal{F}_X}[D_{\mathrm{KL}}(p(\cdot|X)\|q(\cdot|X))].$$

In this paper, we consider a probability space $(\Omega, \mathcal{F}, p)$ on which $(\boldsymbol{w}_t(\omega))_{t \geq 0}$ is a Wiener process in $\mathbb{R}^d$. The Wiener process $(\boldsymbol{w}_t(\omega))_{t \geq 0}$ generates the filtration $\{\mathcal{F}_t\}_{t \geq 0}$ on the measurable space $(\Omega, \mathcal{F})$. For an Itô process $\boldsymbol{z}_t(\omega)$ with the following governing SDE:

$$\mathrm{d}\boldsymbol{z}_t(\omega) = \boldsymbol{\alpha}(t, \omega)\mathrm{d}t + \boldsymbol{\Sigma}(t, \omega)\mathrm{d}\boldsymbol{w}_t(\omega),$$

for any time $t$, we denote the marginal distribution of $\boldsymbol{z}_t$ by $p_t$, *i.e.*

$$p_t := p\left(\boldsymbol{z}_t^{-1}(\cdot)\right), \text{ where } \boldsymbol{z}_t : \Omega \to \mathbb{R}^m, \ \omega \mapsto \boldsymbol{z}_t(\omega),$$

as well as the *path measure* of the process $\boldsymbol{z}_t$ in the sense of

$$p_{t_1:t_2} := p\left(\boldsymbol{z}_{t_1:t_2}^{-1}(\cdot)\right), \text{ where } \boldsymbol{z}_{t_1:t_2} : \Omega \to \mathcal{C}([t_1, t_2], \mathbb{R}^m), \ \omega \mapsto (\boldsymbol{z}_t(\omega))_{t \in [t_1, t_2]}.$$

For the sake of simplicity, we define the following class of functions:

**Definition A.2.** *For any $0 \leq t_1 < t_2$, we define $\mathcal{V}(t_1, t_2)$ as the class of functions $f(t, \omega)$ : $[0, +\infty) \times \Omega \to \mathbb{R}$ such that*

1. *$f(t, \omega)$ is $\mathcal{B} \times \mathcal{F}$-measurable, where $\mathcal{B}$ is the Borel $\sigma$-algebra on $\mathbb{R}^d$;*

2. *$f(t, \omega)$ is $\mathcal{F}_t$-adapted for all $t \geq 0$;*

3. *The following* Novikov condition *holds*

$$\mathbb{E}\left[\exp \int_{t_1}^{t_2} f^2(t, \omega)\mathrm{d}t\right] < +\infty,$$

*and $\mathcal{V} = \cap_{t>0} \mathcal{V}(0, t)$. For vectors and matrices, we say it belongs to $\mathcal{V}^n(t, \omega)$ or $\mathcal{V}^{m \times n}(t, \omega)$ if each component of the vector or each entry of the matrix belongs to $\mathcal{V}(t, \omega)$.*

**Remark A.3.** *Novikov's condition appeared in the third requirement is often relaxed to the* squared integrability *condition in the general definition of Itô processes, which requires*

$$\mathbb{E}\left[\int_{t_1}^{t_2} f^2(t, \omega)\mathrm{d}t\right] < +\infty.$$

*Here, we adopt the more restricted condition in the spirit of its necessity for Girsanov's theorem to hold, as we shall see later.*

Similar to previous work [111], here we can avoid checking Novikov's condition throughout our proofs below by using the approximation argument presented in [100]. A review of Girsanov can be found in textbooks like in [131, 144]. We will present the following generalized version of Girsanov's theorem:

**Theorem A.4** (Girsanov's Theorem [131, Theorem 8.6.6]). *Let $\boldsymbol{\alpha}(t, \omega) \in \mathcal{V}^m$, $\boldsymbol{\Sigma}(t, \omega) \in \mathcal{V}^{m \times n}$, and $(\boldsymbol{w}_t(\omega))_{t \geq 0}$ be a Wiener process on the probability space $(\Omega, \mathcal{F}, q)$. For $t \in [0, T]$, suppose $\boldsymbol{z}_t(\omega)$ is an Itô process with the following SDE:*

$$\mathrm{d}\boldsymbol{z}_t(\omega) = \boldsymbol{\alpha}(t, \omega)\mathrm{d}t + \boldsymbol{\Sigma}(t, \omega)\mathrm{d}\boldsymbol{w}_t(\omega), \tag{A.2}$$

*and there exist processes $\boldsymbol{\delta}(t, \omega) \in \mathcal{V}^n$ and $\boldsymbol{\beta}(t, \omega) \in \mathcal{V}^m$ such that*

1. *$\boldsymbol{\Sigma}(t, \omega)\boldsymbol{\delta}(t, \omega) = \boldsymbol{\alpha}(t, \omega) - \boldsymbol{\beta}(t, \omega)$;*

2. *The process $M_t(\omega)$ as defined below is a martingale with respect to the filtration $\{\mathcal{F}_t\}_{t \geq 0}$ and probability measure $q$:*

$$M_t(\omega) = \exp\left(-\int_0^t \boldsymbol{\delta}(s, \omega)^\top \mathrm{d}\boldsymbol{w}_s(\omega) - \frac{1}{2}\int_0^t \|\boldsymbol{\delta}(s, \omega)\|^2 \mathrm{d}s\right),$$

*then there exists another probability measure $p$ on $(\Omega, \mathcal{F})$ such that*

1. *$p \ll q$ with the Radon-Nikodym derivative $\dfrac{\mathrm{d}p}{\mathrm{d}q}(\omega) = M_T(\omega)$,*

2. *The process $\widetilde{\boldsymbol{w}}_t(\omega)$ as defined below is a Wiener process on $(\Omega, \mathcal{F}, p)$:*

$$\widetilde{\boldsymbol{w}}_t(\omega) = \boldsymbol{w}_t(\omega) + \int_0^t \boldsymbol{\delta}(s, \omega) \mathrm{d}s,$$

3. *Any continuous path in $\mathcal{C}([t_1, t_2], \mathbb{R}^m)$ generated by the process $\boldsymbol{z}_t$ satisfies the following SDE under the probability measure $p$:*

$$\mathrm{d}\widetilde{\boldsymbol{z}}_t(\omega) = \boldsymbol{\beta}(t, \omega)\mathrm{d}t + \boldsymbol{\Sigma}(t, \omega)\mathrm{d}\widetilde{\boldsymbol{w}}_t(\omega). \tag{A.3}$$

**Corollary A.5.** *Suppose the conditions in Theorem A.4 hold, then for any $t_1, t_2 \in [0, T]$ with $t_1 < t_2$, the path measure of the SDE (A.3) under the probability measure $p$ in the sense of $p_{t_1:t_2} = p\left(\boldsymbol{z}_{t_1:t_2}^{-1}(\cdot)\right)$ is absolutely continuous with respect to the path measure of the SDE (A.2) in the sense of $q_{t_1:t_2} = q\left(\boldsymbol{z}_{t_1:t_2}^{-1}(\cdot)\right)$. Moreover, the KL divergence between the two path measures is given by*

$$D_{\mathrm{KL}}(p_{t_1:t_2} \| q_{t_1:t_2}) = D_{\mathrm{KL}}(p_{t_1} \| q_{t_1}) + \mathbb{E}_{\omega \sim p|_{\mathcal{F}_{t_1}}}\left[\frac{1}{2}\int_{t_1}^{t_2} \|\boldsymbol{\delta}(t, \omega)\|^2 \mathrm{d}t\right] \tag{A.4}$$

*Proof.* First, by Theorem A.1, we have

$$D_{\mathrm{KL}}(p_{t_1:t_2} \| q_{t_1:t_2}) = D_{\mathrm{KL}}(p|_{\mathcal{F}_{t_1}} \| q|_{\mathcal{F}_{t_1}}) + \mathbb{E}_{\boldsymbol{z} \sim p|_{\mathcal{F}_{t_1}}}\left[D_{\mathrm{KL}}\left(p(\widetilde{\boldsymbol{z}}_{t_1:t_2}^{-1}(\cdot))|\widetilde{\boldsymbol{z}}_{t_1} = \widetilde{\boldsymbol{z}})\|q(\widetilde{\boldsymbol{z}}_{t_1:t_2}^{-1}(\cdot))|\widetilde{\boldsymbol{z}}_{t_1} = \widetilde{\boldsymbol{z}})\right)\right].$$

From Girsanov's theorem (Theorem A.4), we have that the measure $p|_{\mathcal{F}_{t_1}}$ is absolutely continuous with respect to $q|_{\mathcal{F}_{t_1}}$, which allows us to compute the second term above as follows:

$$D_{\mathrm{KL}}\left(p(\widetilde{\boldsymbol{z}}_{t_1:t_2}^{-1}(\cdot)|\widetilde{\boldsymbol{z}}_{t_1} = \widetilde{\boldsymbol{z}})\|q(\widetilde{\boldsymbol{z}}_{t_1:t_2}^{-1}(\cdot)|\widetilde{\boldsymbol{z}}_{t_1} = \widetilde{\boldsymbol{z}})\right)$$

$$= \mathbb{E}_{\widetilde{\boldsymbol{z}}_{t_1:t_2}}\left[\log \frac{\mathrm{d}p(\widetilde{\boldsymbol{z}}_{t_1:t_2}^{-1}(\cdot)|\widetilde{\boldsymbol{z}}_{t_1} = \boldsymbol{z})}{\mathrm{d}q(\widetilde{\boldsymbol{z}}_{t_1:t_2}^{-1}(\cdot)|\widetilde{\boldsymbol{z}}_{t_1} = \boldsymbol{z})}\right] = \mathbb{E}_{\omega \sim p|_{\mathcal{F}_{t_1}}}\left[\log \frac{\mathrm{d}p|_{\mathcal{F}_{t_1}}}{\mathrm{d}q|_{\mathcal{F}_{t_1}}}\right]$$

$$= \mathbb{E}_{\omega \sim p|_{\mathcal{F}_{t_1}}}\left[-\int_{t_1}^{t_2} \boldsymbol{\delta}(t, \omega)^\top \mathrm{d}\boldsymbol{w}_t(\omega) - \frac{1}{2}\int_{t_1}^{t_2} \|\boldsymbol{\delta}(t, \omega)\|^2 \mathrm{d}t\right]$$

$$= \mathbb{E}_{\omega \sim p|_{\mathcal{F}_{t_1}}}\left[-\int_{t_1}^{t_2} \boldsymbol{\delta}(t, \omega)^\top (\mathrm{d}\widetilde{\boldsymbol{w}}_t(\omega) - \boldsymbol{\delta}(t, \omega)\mathrm{d}t) - \frac{1}{2}\int_{t_1}^{t_2} \|\boldsymbol{\delta}(t, \omega)\|^2 \mathrm{d}t\right]$$

$$= \mathbb{E}_{\omega \sim p|_{\mathcal{F}_{t_1}}}\left[\frac{1}{2}\int_{t_1}^{t_2} \|\boldsymbol{\delta}(t, \omega)\|^2 \mathrm{d}t\right],$$

and therefore

$$D_{\mathrm{KL}}(p_{t_1:t_2} \| q_{t_1:t_2}) = D_{\mathrm{KL}}(p_{t_1} \| q_{t_1}) + \mathbb{E}_{\omega \sim p|_{\mathcal{F}_{t_1}}}\left[\frac{1}{2}\int_{t_1}^{t_2} \|\boldsymbol{\delta}(t, \omega)\|^2 \mathrm{d}t\right],$$

which completes the proof. $\qquad\square$

### A.3 Helper Lemmas

**Lemma A.6** ([107, Lemma 2])**.** *For the backward process (2.3), we have for $0 \leq s < t < T$,*

$$\frac{\mathrm{d}}{\mathrm{d}t}\left(\mathbb{E}\left[\|\nabla \log \bar{p}_t(\bar{\boldsymbol{x}}_t) - \nabla \log \bar{p}_s(\bar{\boldsymbol{x}}_s)\|^2\right]\right) \leq \frac{1}{2}\mathbb{E}\left[\|\nabla \log \bar{p}_s(\bar{\boldsymbol{x}}_s)\|^2\right] + \mathbb{E}\left[\|\nabla^2 \log \bar{p}_t(\bar{\boldsymbol{x}}_t)\|_F^2\right].$$

**Lemma A.7** ([107, Lemma 3])**.** *For the forward process (2.3), we have for $0 \leq t < T$,*

$$\mathbb{E}\left[\nabla \log p_t(\boldsymbol{x}_t)\right] \leq d\sigma_t^{-2}, \text{ and } \mathbb{E}\left[\|\nabla^2 \log p_t(\boldsymbol{x}_t)\|_F^2\right] \leq d\sigma_t^{-4} + 2\frac{\mathrm{d}}{\mathrm{d}t}\left(\sigma_t^{-4}\mathbb{E}\left[\operatorname{tr}\boldsymbol{\Sigma}_t\right]\right),$$

*where the posterior covariance matrix $\boldsymbol{\Sigma}_t := \operatorname{cov}_{p_{0|t}}(\boldsymbol{x}_0)$ and $\sigma_t^2 = 1 - e^{-t}$. Moreover, the posterior covariance matrix $\boldsymbol{\Sigma}_t$ satisfies*

$$\mathbb{E}\left[\operatorname{tr}\boldsymbol{\Sigma}_t\right] \lesssim d \wedge d\sigma_t^2.$$

**Lemma A.8.** *For any $n \in [0 : N-1]$ and $\tau \in [0, h_n]$, under the assumption $\mathrm{cov}_{p_0}(\boldsymbol{x}_0) = \boldsymbol{I}_d$, we have*

$$\mathbb{E}\left[\|\bar{\boldsymbol{x}}_{t_n}\|^2\right] \leq 2d, \tag{A.5}$$

*and*

$$\mathbb{E}\left[\|\bar{\boldsymbol{x}}_{t_n} - \bar{\boldsymbol{x}}_{t_n+\tau}\|^2\right] \leq 3d. \tag{A.6}$$

*Proof.* Conditioned on $\boldsymbol{x}_0$, we have that

$$\bar{\boldsymbol{x}}_{t_n} = \boldsymbol{x}_{T-t_n} \sim \mathcal{N}\left(e^{-\frac{1}{2}(T-t_n)}\boldsymbol{x}_0, (1 - e^{-(T-t_n)})\boldsymbol{I}_d\right),$$

and

$$\bar{\boldsymbol{x}}_{t_n+\tau} = \boldsymbol{x}_{T-t_n-\tau} \sim \mathcal{N}\left(e^{-\frac{1}{2}(T-t_n-\tau)}\boldsymbol{x}_0, (1 - e^{-(T-t_n-\tau)})\boldsymbol{I}_d\right)$$

for any $\tau \in [0, h_n]$. Therefore, we have

$$\begin{aligned}
\mathbb{E}\left[\|\bar{\boldsymbol{x}}_{t_n}\|^2\right] &= \mathbb{E}\left[\mathbb{E}\left[\|\boldsymbol{x}_{T-t_n}\|^2 \big| \boldsymbol{x}_0\right]\right] \\
&\leq \mathbb{E}\left[\mathbb{E}\left[\|\boldsymbol{x}_{T-t_n} - e^{-\frac{1}{2}(T-t_n)}\boldsymbol{x}_0\|^2 \big| \boldsymbol{x}_0\right] + \|e^{-\frac{1}{2}(T-t_n)}\boldsymbol{x}_0\|^2\right] \\
&\leq d(1 - e^{-(T-t_n)}) + e^{-(T-t_n)}\mathbb{E}[\|\boldsymbol{x}_0\|^2] \leq 2d.
\end{aligned}$$

Taking the difference between them then implies that for any $\tau \in [0, h_n]$,

$$\begin{aligned}
\mathbb{E}\left[\|\bar{\boldsymbol{x}}_{t_n} - \bar{\boldsymbol{x}}_{t_n+\tau}\|^2\right] &= \mathbb{E}\left[\mathbb{E}\left[\|\boldsymbol{x}_{T-t_n} - \boldsymbol{x}_{T-t_n-\tau}\|^2 \big| \boldsymbol{x}_0\right]\right] \\
&\leq d(2 - e^{-(T-t_n)} - e^{-(T-t_n-\tau)}) \\
&\quad + \left(e^{-\frac{1}{2}(T-t_n)} - e^{-\frac{1}{2}(T-t_n-\tau)}\right)^2 \mathbb{E}[\|\boldsymbol{x}_0\|^2] \\
&\leq 2d + e^{-(T-t_n-\tau)}(1 - e^{-\frac{1}{2}\tau})^2 \mathbb{E}[\|\boldsymbol{x}_0\|^2] \leq 3d.
\end{aligned}$$

$\square$

**Lemma A.9** (Lemma 9 in [104]). *For $\widehat{q}_0 \sim \mathcal{N}(0, \boldsymbol{I}_d)$ and $\bar{p}_0 = p_T$ is the distribution of the solution to the forward process (2.3), we have*

$$\mathrm{TV}(\bar{p}_0, \widehat{q}_0)^2 \leq D_{\mathrm{KL}}(\bar{p}_0 \| \widehat{q}_0) \lesssim de^{-T}.$$

# B  Details of SDE Implementation

In this section, we will present the missing proofs for Theorem 3.3. For readers' convenience, we reiterate the backward process (2.3)

$$\mathrm{d}\bar{\boldsymbol{x}}_t = \left[\frac{1}{2}\bar{\boldsymbol{x}}_t + \nabla \log \bar{p}_t(\bar{\boldsymbol{x}}_t)\right]\mathrm{d}t + \mathrm{d}\boldsymbol{w}_t, \quad \text{with} \quad \bar{\boldsymbol{x}}_0 \sim p_T, \tag{B.1}$$

and its approximate version (2.5) with the learned score function

$$\mathrm{d}\boldsymbol{y}_t = \left[\frac{1}{2}\boldsymbol{y}_t + \boldsymbol{s}_t^\theta(\boldsymbol{y}_t)\right]\mathrm{d}t + \mathrm{d}\boldsymbol{w}_t, \quad \text{with} \quad \boldsymbol{y}_0 \sim \mathcal{N}(0, \boldsymbol{I}_d).$$

The filtration $\mathcal{F}_t$ refers to the filtration of the SDE (B.1) up to time $t$.

## B.1 Auxiliary Process

We would like first to consider the errors that Algorithm 1 may cause within one block of update. To this end, we consider the following auxiliary process for $\tau \in [0, h_n]$ conditioned on the filtration $\mathcal{F}_{t_n}$ at time $t_n$:

**Definition B.1** (Auxiliary Process). *For any $n \in [0 : N-1]$, we define the auxiliary process $(\widehat{\boldsymbol{y}}_{t_n,\tau}^{(k)})_{\tau \in [0,h_n]}$ as the solution to the following SDE recursively for $k \in [0 : K-1]$:*

$$\mathrm{d}\widehat{\boldsymbol{y}}_{t_n,\tau}^{(k+1)}(\omega) = \left[ \frac{1}{2}\widehat{\boldsymbol{y}}_{t_n,\tau}^{(k+1)}(\omega) + \boldsymbol{s}_{t_n+g_n(\tau)}^{\theta}\left( \widehat{\boldsymbol{y}}_{t_n,g_n(\tau)}^{(k)}(\omega) \right) \right] \mathrm{d}\tau + \mathrm{d}\boldsymbol{w}_{t_n+\tau}(\omega), \qquad \text{(B.2)}$$

*with the initial condition*

$$\widehat{\boldsymbol{y}}_{t_n,\tau}^{(0)}(\omega) \equiv \widehat{\boldsymbol{y}}_{t_n}(\omega) \, \text{for } \tau \in [0, h_n], \quad \text{and} \quad \widehat{\boldsymbol{y}}_{t_n,0}^{(k)}(\omega) \equiv \widehat{\boldsymbol{y}}_{t_n}(\omega) \, \text{for } k \in [1 : K] \qquad \text{(B.3)}$$

*where $\widehat{\boldsymbol{y}}_{t_n}(\omega) = \widehat{\boldsymbol{y}}_{t_{n-1},\tau_{n-1},M_{n-1}}^{(K)}(\omega)$ if $n \in [1 : N-1]$ and $\widehat{\boldsymbol{y}}_{t_0}(\omega) \sim \mathcal{N}(0, \boldsymbol{I}_d)$.*

The iteration should be perceived as a deterministic procedure to each event $\omega \in \Omega$, *i.e.* each realization of the Wiener process $(\boldsymbol{w}_t)_{t \geq 0}$. The following lemma clarifies this fact and proves the well-definedness and parallelability of the iteration in (B.2).

**Lemma B.2.** *The auxiliary process $(\widehat{\boldsymbol{y}}_{t_n,\tau}^{(k)}(\omega))_{\tau \in [0,h_n]}$ is $\mathcal{F}_{t_n+\tau}$-adapted for any $k \in [0 : K]$ and $n \in [0 : N-1]$.*

*Proof.* Since the initialization $\widehat{\boldsymbol{y}}_{t_n,\tau}^{(0)}(\omega) \equiv \widehat{\boldsymbol{y}}_{t_n}(\omega)$ for $\tau \in [0, h_n]$, where $\widehat{\boldsymbol{y}}_{t_n}(\omega)$ is $\mathcal{F}_{t_n}$-adapted, it is obvious that $\widehat{\boldsymbol{y}}_{t_n,\tau}^{(0)}(\omega)$ is $\mathcal{F}_{t_n+\tau}$-adapted. Now suppose that $(\widehat{\boldsymbol{y}}_{t_n,\tau}^{(k)}(\omega))_{\tau \in [0,h_n]}$ is $\mathcal{F}_{t_n+\tau}$-adapted, since $g_n(\tau) \leq \tau$, we have the following Itô integral well-defined and $\mathcal{F}_{t_n+\tau}$-adapted:

$$\int_0^{\tau} \boldsymbol{s}_{t_n+g_n(\tau')}^{\theta}\left( \widehat{\boldsymbol{y}}_{t_n,g_n(\tau')}^{(k)}(\omega) \right) \mathrm{d}\tau',$$

and therefore (B.2) has a unique strong solution $(\widehat{\boldsymbol{y}}_{t_n,\tau}^{(k+1)}(\omega))_{\tau \in [0,h_n]}$ that is also $\mathcal{F}_{t_n+\tau}$-adapted. The lemma follows by induction. $\qquad \square$

**Lemma B.3** (Equivalence between (3.4) and (B.2)). *For any $n \in [0 : N-1]$, the update rule (3.4) in Algorithm 1 is equivalent to the exact solution of the auxiliary process (B.2) for any $k \in [0 : K-1]$ and $\tau \in [0, h_n]$.*

*Proof.* The dependency on $\omega$ will be omitted in the proof below.

Rewriting (B.2) and multiplying $e^{-\frac{\tau}{2}}$ on both sides yield

$$\mathrm{d}\left[ e^{-\frac{\tau}{2}}\widehat{\boldsymbol{y}}_{t_n,\tau}^{(k+1)} \right] = e^{-\frac{\tau}{2}}\left[ \mathrm{d}\widehat{\boldsymbol{y}}_{t_n,\tau}^{(k+1)} - \frac{1}{2}\widehat{\boldsymbol{y}}_{t_n,\tau}^{(k+1)}\mathrm{d}\tau \right] = e^{-\frac{\tau}{2}}\left[ \boldsymbol{s}_{t_n+g_n(\tau)}^{\theta}\left( \widehat{\boldsymbol{y}}_{t_n,g_n(\tau)}^{(k)} \right) \mathrm{d}\tau + \mathrm{d}\boldsymbol{w}_{t_n+\tau} \right].$$

Integrating on both sides from 0 to $\tau$ implies

$$e^{-\frac{\tau}{2}}\widehat{\boldsymbol{y}}_{t_n,\tau}^{(k+1)} - \widehat{\boldsymbol{y}}_{t_n,0}^{(k+1)} = \int_0^{\tau} e^{-\frac{\tau'}{2}}\left( \boldsymbol{s}_{t_n+g_n(\tau')}^{\theta}\left( \widehat{\boldsymbol{y}}_{t_n,g_n(\tau')}^{(k)} \right) \mathrm{d}\tau' + \mathrm{d}\boldsymbol{w}_{t_n+\tau'} \right)$$

$$= \sum_{m=0}^{M_n} \int_{\tau \wedge t_{n,m}}^{\tau \wedge \tau_{n,m+1}} e^{-\frac{\tau'}{2}}\boldsymbol{s}_{t_n+\tau_{n,m}}^{\theta}\left( \widehat{\boldsymbol{y}}_{t_n,\tau_{n,m}}^{(k)} \right) \mathrm{d}\tau' + \int_0^{\tau} e^{-\frac{\tau'}{2}}\mathrm{d}\boldsymbol{w}_{t_n+\tau'}$$

$$= \sum_{m=0}^{M_n} 2\left( e^{-\frac{\tau \wedge \tau_{n,m}}{2}} - e^{-\frac{\tau \wedge \tau_{n,m+1}}{2}} \right)\boldsymbol{s}_{t_n+\tau_{n,j}}^{\theta}\left( \widehat{\boldsymbol{y}}_{t_n,\tau_{n,m}}^{(k)} \right) + \int_0^{\tau} e^{-\frac{\tau'}{2}}\mathrm{d}\boldsymbol{w}_{t_n+\tau'},$$

and then multiplying $e^{\frac{\tau}{2}}$ on both sides above yields

$$\widehat{\boldsymbol{y}}_{t_n,\tau}^{(k+1)} = e^{\frac{\tau}{2}}\widehat{\boldsymbol{y}}_{t_n,0}^{(k+1)} + \sum_{m=0}^{M_n} 2\left( e^{\frac{\tau \wedge \tau_{n,m+1} - \tau \wedge \tau_{n,m}}{2}} - 1 \right)e^{\frac{0 \vee (\tau - \tau_{n,m+1})}{2}}\boldsymbol{s}_{t_n+\tau_{n,m}}^{\theta}\left( \widehat{\boldsymbol{y}}_{t_n,\tau_{n,m}}^{(k)} \right)$$

$$+ \sum_{m=0}^{M_n} \int_{\tau \wedge \tau_{n,m}}^{\tau \wedge \tau_{n,m+1}} e^{\frac{\tau - \tau'}{2}}\mathrm{d}\boldsymbol{w}_{t_n+\tau'},$$

where, by Itô isometry, we have

$$\int_{\tau \wedge \tau_{n,m}}^{\tau \wedge \tau_{n,m+1}} e^{\frac{\tau-\tau'}{2}} \mathrm{d}\boldsymbol{w}_{t_n+\tau'} \sim \mathcal{N}\left(\boldsymbol{0}, \left(e^{\tau \wedge \tau_{n,m+1} - \tau \wedge \tau_{n,m}} - 1\right) e^{0 \vee (\tau - \tau_{n,m+1})} \boldsymbol{I}_d\right)$$

for $\tau > \tau_{n,m}$ and equals to $\boldsymbol{0}$ otherwise. Plugging in $\tau = \tau_{j,m}$ gives us (3.4), as desired. $\qquad\square$

## B.2 Errors within Block

We shall invoke Girsanov's theorem (Theorem A.4) in the procedure as detailed below:

1. Setting (A.2) in Theorem A.4 as the auxiliary process (B.2) at iteration $K$, where $\boldsymbol{w}_t(\omega)$ is a Wiener process under the measure $q|_{\mathcal{F}_{t_n}}$;

2. Defining another process $\widetilde{\boldsymbol{w}}_{t_n+\tau}(\omega)$ governed by the following SDE:

$$\mathrm{d}\widetilde{\boldsymbol{w}}_{t_n+\tau}(\omega) = \mathrm{d}\boldsymbol{w}_{t_n+\tau}(\omega) + \boldsymbol{\delta}_{t_n}(\tau, \omega)d\tau,$$

    where

$$\boldsymbol{\delta}_{t_n}(\tau, \omega) := \boldsymbol{s}_{t_n+g_n(\tau)}^{\theta}(\widehat{\boldsymbol{y}}_{t_n,g_n(\tau)}^{(K-1)}(\omega)) - \nabla \log \breve{p}_{t_n+\tau}(\widehat{\boldsymbol{y}}_{t_n+\tau}^{(K)}(\omega)), \tag{B.4}$$

    and computing the Radon-Nikodym derivative of the measure $\breve{p}|_{\mathcal{F}_{t_n}}$ with respect to $q|_{\mathcal{F}_{t_n}}$ as

$$\frac{\mathrm{d}\breve{p}|_{\mathcal{F}_{t_n}}}{\mathrm{d}q|_{\mathcal{F}_{t_n}}}(\omega) := \exp\left(-\int_0^{h_n} \boldsymbol{\delta}_{t_n}(\tau, \omega)^\top \mathrm{d}\boldsymbol{w}_{t_n+\tau}(\omega) - \frac{1}{2}\int_0^{h_n} \|\boldsymbol{\delta}_{t_n}(\tau, \omega)\|^2 \mathrm{d}\tau\right),$$

3. Concluding that (B.2) at iteration $K$ under the measure $q|_{\mathcal{F}_{t_n}}$ satisfies the following SDE:

$$\mathrm{d}\widehat{\boldsymbol{y}}_{t_n,\tau}^{(K)}(\omega) = \left[\frac{1}{2}\widehat{\boldsymbol{y}}_{t_n,\tau}^{(K)}(\omega) + \nabla \log \breve{p}_{t_n+\tau}\left(\widehat{\boldsymbol{y}}_{t_n,\tau}^{(K)}(\omega)\right)\right] \mathrm{d}\tau + \mathrm{d}\widetilde{\boldsymbol{w}}_{t_n+\tau}(\omega),$$

    with $(\widetilde{\boldsymbol{w}}_{t_n+\tau})_{\tau \geq 0}$ being a Wiener process under the measure $\breve{p}|_{\mathcal{F}_{t_n}}$. If we replace $\widehat{\boldsymbol{y}}_{t_n,g_n(\tau)}^{(K)}(\omega)$ by $\breve{\boldsymbol{x}}_{t_n+\tau}(\omega)$, one should notice (B.5) is immediately the original backward SDE (2.3) with the true score function on $t \in [t_n, t_{n+1}]$:

$$\mathrm{d}\breve{\boldsymbol{x}}_{t_n+\tau}(\omega) = \left[\frac{1}{2}\breve{\boldsymbol{x}}_{t_n+\tau}(\omega) + \nabla \log \breve{p}_{t_n+\tau}(\breve{\boldsymbol{x}}_{t_n+\tau}(\omega))\right] \mathrm{d}\tau + \mathrm{d}\widetilde{\boldsymbol{w}}_{t_n+\tau}(\omega). \tag{B.5}$$

**Remark B.4.** *The applicability of Girsanov's theorem here relies on the $\mathcal{F}_\tau$-adaptivity of $\boldsymbol{s}_{t_n+g_n(\tau)}^{\theta}\left(\widehat{\boldsymbol{y}}_{t_n,g_n(\tau)}^{(K-1)}(\omega)\right)$ established by Lemma B.2. One should notice the change of measure procedure above depends on the number of iterations $K$, and different $K$ would lead to different transform (B.4).*

Then Corollary A.5 provides the following computation

$$D_{\mathrm{KL}}(\breve{p}_{t_{n+1}} \| \widehat{q}_{t_{n+1}}) \leq D_{\mathrm{KL}}(\breve{p}_{t_n:t_{n+1}} \| \widehat{q}_{t_n:t_{n+1}})$$

$$= D_{\mathrm{KL}}(\breve{p}_{t_n} \| \widehat{q}_{t_n}) + \mathbb{E}_{\omega \sim q|_{\mathcal{F}_{t_n}}}\left[\frac{1}{2}\int_0^{h_n} \|\boldsymbol{\delta}_{t_n}(\tau, \omega)\|^2 \mathrm{d}\tau\right], \tag{B.6}$$

where the first inequality is by the data-processing inequality (Theorem A.1). Now, the problem remaining is to bound the discrepancy quantified by

$$\int_0^{h_n} \|\boldsymbol{\delta}_{t_n}(\tau,\omega)\|^2 \mathrm{d}\tau$$

$$= \int_0^{h_n} \left\| \boldsymbol{s}_{t_n+g_n(\tau)}^{\theta}(\widehat{\boldsymbol{y}}_{t_n,g_n(\tau)}^{(K-1)}(\omega)) - \nabla \log \breve{p}_{t_n+\tau}(\widehat{\boldsymbol{y}}_{t_n,\tau}^{(K)}(\omega)) \right\|^2 \mathrm{d}\tau$$

$$\leq 3 \left( \underbrace{\int_0^{h_n} \left\| \nabla \log \breve{p}_{t_n+g_n(\tau)}\big(\widehat{\boldsymbol{y}}_{t_n,g_n(\tau)}^{(K)}(\omega)\big) - \nabla \log \breve{p}_{t_n+\tau}\big(\widehat{\boldsymbol{y}}_{t_n,\tau}^{(K)}(\omega)\big) \right\|^2 \mathrm{d}\tau}_{:=A_{t_n}(\omega)} \right.$$

$$+ \underbrace{\int_0^{h_n} \left\| \boldsymbol{s}_{t_n+g_n(\tau)}^{\theta}\big(\widehat{\boldsymbol{y}}_{t_n,g_n(\tau)}^{(K)}(\omega)\big) - \nabla \log \breve{p}_{t_n+g_n(\tau)}\big(\widehat{\boldsymbol{y}}_{t_n,g_n(\tau)}^{(K)}(\omega)\big) \right\|^2 \mathrm{d}\tau}_{:=B_{t_n}(\omega)}$$

$$+ \left. \int_0^{h_n} \left\| \boldsymbol{s}_{t_n+g_n(\tau)}^{\theta}\big(\widehat{\boldsymbol{y}}_{t_n,g_n(\tau)}^{(K)}(\omega)\big) - \boldsymbol{s}_{t_n+g_n(\tau)}^{\theta}\big(\widehat{\boldsymbol{y}}_{t_n,g_n(\tau)}^{(K-1)}(\omega)\big) \right\|^2 \mathrm{d}\tau \right). \tag{B.7}$$

Before we continue our proof, we would like first to provide the following lemma bounding the behavior of the auxiliary process (B.2) when $k = 0$ for $\tau \in [0, h_n]$.

**Lemma B.5.** *For any $n \in [0 : N-1]$, suppose the initialization $\widehat{\boldsymbol{y}}_{t_n}$ in (B.3) of the auxiliary process (B.2) follows the distribution of $\breve{\boldsymbol{x}}_{t_n} \sim \breve{p}_{t_n}$, then the following estimate holds*

$$\sup_{\tau \in [0,h_n]} \mathbb{E}_{\omega \sim \breve{p}|_{\mathcal{F}_{t_n}}} \left[ \|\widehat{\boldsymbol{y}}_{t_n,\tau}^{(1)}(\omega) - \widehat{\boldsymbol{y}}_{t_n,\tau}^{(0)}(\omega)\|^2 \right]$$

$$\leq h_n e^{\frac{7}{2}h_n}\left( M_{\boldsymbol{s}}^2 + 2d \right) + 3e^{\frac{7}{2}h_n} \mathbb{E}_{\omega \sim \breve{p}|_{\mathcal{F}_{t_n}}}\left[ A_{t_n}(\omega) + B_{t_n}(\omega) \right] \tag{B.8}$$

$$+ 3e^{\frac{7}{2}h_n} h_n L_{\boldsymbol{s}}^2 \sup_{\tau \in [0,h_n]} \mathbb{E}_{\omega \sim \breve{p}|_{\mathcal{F}_{t_n}}}\left[ \left\| \widehat{\boldsymbol{y}}_{t_n,\tau}^{(K)}(\omega) - \widehat{\boldsymbol{y}}_{t_n,\tau}^{(K-1)}(\omega) \right\|^2 \right].$$

*Proof.* Let $\boldsymbol{z}_{t_n,\tau} = \widehat{\boldsymbol{y}}_{t_n,\tau}^{(1)} - \widehat{\boldsymbol{y}}_{t_n,\tau}^{(0)}$. For $k = 0$, we can rewrite (B.2) as

$$\mathrm{d}\boldsymbol{z}_{t_n,\tau} = \left[ \frac{1}{2}\left( \boldsymbol{z}_{t_n,\tau} + \widehat{\boldsymbol{y}}_{t_n,\tau}^{(0)} \right) + \boldsymbol{s}_{t_n+g_n(\tau)}^{\theta}\big(\widehat{\boldsymbol{y}}_{t_n,g_n(\tau)}^{(0)}\big) \right] \mathrm{d}\tau + \mathrm{d}\boldsymbol{w}_{t_n+\tau},$$

By applying Itô's lemma and plugging in the expression of $\boldsymbol{w}_{t_n+\tau}$ given by Theorem A.4, we have

$$\mathrm{d}\|\boldsymbol{z}_{t_n,\tau}\|^2 = \left[ \|\boldsymbol{z}_{t_n,\tau}\|^2 + \boldsymbol{z}_{t_n,\tau}^{\top}\widehat{\boldsymbol{y}}_{t_n,\tau}^{(0)} + 2\boldsymbol{z}_{t_n,\tau}^{\top}\boldsymbol{s}_{t_n+g_n(\tau)}^{\theta}\big(\widehat{\boldsymbol{y}}_{t_n,g_n(\tau)}^{(0)}\big) + d \right] \mathrm{d}\tau$$

$$+ 2\boldsymbol{z}_{t_n,\tau}^{\top}\big(\mathrm{d}\widetilde{\boldsymbol{w}}_{t_n+\tau}(\omega) - \boldsymbol{\delta}_{t_n}(\tau,\omega)d\tau\big), \tag{B.9}$$

By integrating from 0 to $\tau$ and taking the expectation on both sides of (B.9), we obtain that

$$\mathbb{E}_{\omega \sim \breve{p}|_{\mathcal{F}_{t_n}}}\left[ \|\boldsymbol{z}_{t_n,\tau}\|^2 \right]$$

$$= \mathbb{E}_{\omega \sim \breve{p}|_{\mathcal{F}_{t_n}}}\left[ \int_0^{\tau}\left( \|\boldsymbol{z}_{t_n,\tau'}\|^2 + \boldsymbol{z}_{t_n,\tau'}^{\top}\widehat{\boldsymbol{y}}_{t_n,\tau'}^{(0)} + 2\boldsymbol{z}_{t_n,\tau'}^{\top}\boldsymbol{s}_{t_n+g_n(\tau')}^{\theta}\big(\widehat{\boldsymbol{y}}_{t_n,g_n(\tau')}^{(0)}\big) + d \right)\mathrm{d}\tau' \right]$$

$$+ 2\mathbb{E}_{\omega \sim \breve{p}|_{\mathcal{F}_{t_n}}}\left[ \int_0^{\tau}\boldsymbol{z}_{t_n,\tau'}^{\top}\big(\mathrm{d}\widetilde{\boldsymbol{w}}_{t_n+\tau'}(\omega) - \boldsymbol{\delta}_{t_n}(\tau',\omega)\mathrm{d}\tau'\big) \right],$$

and by AM-GM, we further have

$$\mathbb{E}_{\omega \sim \breve{p}|_{\mathcal{F}_{t_n}}}\left[ \|\boldsymbol{z}_{t_n,\tau}\|^2 \right]$$

$$\leq \mathbb{E}_{\omega \sim \breve{p}|_{\mathcal{F}_{t_n}}}\left[ \int_0^{\tau}\left[ \frac{7}{2}\|\boldsymbol{z}_{t_n,\tau'}\|^2 + \frac{1}{2}\left\|\widehat{\boldsymbol{y}}_{t_n,\tau'}^{(0)}\right\|^2 + \left\|\boldsymbol{s}_{t_n+g_n(\tau')}^{\theta}\big(\widehat{\boldsymbol{y}}_{t_n,g_n(\tau')}^{(0)}\big)\right\|^2 + d + \|\boldsymbol{\delta}_{t_n}(\tau,\omega)\|^2 \right]\mathrm{d}\tau' \right]$$

$$\leq \int_0^{\tau}\mathbb{E}_{\omega \sim \breve{p}|_{\mathcal{F}_{t_n}}}\left[ \frac{7}{2}\|\boldsymbol{z}_{t_n,\tau'}\|^2 + \|\boldsymbol{\delta}_{t_n}(\tau,\omega)\|^2 \right]\mathrm{d}\tau' + \left( \frac{1}{2}\mathbb{E}\left[\left\|\widehat{\boldsymbol{y}}_{t_n,\tau}^{(0)}\right\|^2\right] + M_{\boldsymbol{s}}^2 + d \right)\tau,$$

where $\boldsymbol{\delta}_{t_n}(\tau, \omega)$ is defined in (B.4). Similar to (B.7), we may use triangle inequality to upper bound $\|\boldsymbol{\delta}_{t_n}(\tau, \omega)\|^2$, which implies that for any $\tau \in [0, h_n]$

$$
\mathbb{E}_{\omega \sim \breve{p}|_{\mathcal{F}_{t_n}}} \left[ \|\boldsymbol{z}_{t_n, \tau}\|^2 \right]
$$

$$
\leq \frac{7}{2} \int_0^\tau \mathbb{E}_{\omega \sim \breve{p}|_{\mathcal{F}_{t_n}}} \left[ \|\boldsymbol{z}_{t_n, \tau'}\|^2 \right] \mathrm{d}\tau' + \left( \frac{1}{2} \mathbb{E} \left[ \left\| \widehat{\boldsymbol{y}}_{t_n, \tau}^{(0)} \right\|^2 \right] + M_{\boldsymbol{s}}^2 + d \right) \tau
$$

$$
+ 3\mathbb{E}_{\omega \sim \breve{p}|_{\mathcal{F}_{t_n}}} \left[ \int_0^\tau \left\| \boldsymbol{s}_{t_n + g_n(\tau)}^\theta (\widehat{\boldsymbol{y}}_{t_n, g_n(\tau)}^{(K-1)}(\omega)) - \boldsymbol{s}_{t_n + g_n(\tau)}^\theta (\widehat{\boldsymbol{y}}_{t_n, g_n(\tau)}^{(K)}(\omega)) \right\|^2 \mathrm{d}\tau' \right]
$$

$$
+ 3\mathbb{E}_{\omega \sim \breve{p}|_{\mathcal{F}_{t_n}}} \left[ \int_0^\tau \left\| \boldsymbol{s}_{t_n + g_n(\tau)}^\theta (\widehat{\boldsymbol{y}}_{t_n, g_n(\tau)}^{(K)}(\omega)) - \nabla \log \breve{p}_{t_n, g_n(\tau)}(\widehat{\boldsymbol{y}}_{t_n, g_n(\tau)}^{(K)}(\omega)) \right\|^2 \mathrm{d}\tau' \right]
$$

$$
+ 3\mathbb{E}_{\omega \sim \breve{p}|_{\mathcal{F}_{t_n}}} \left[ \int_0^\tau \left\| \nabla \log \breve{p}_{t_n + g_n(\tau)}(\widehat{\boldsymbol{y}}_{t_n, g_n(\tau)}^{(K)}(\omega)) - \nabla \log \breve{p}_{t_n + \tau}(\widehat{\boldsymbol{y}}_{t_n + \tau}^{(K)}(\omega)) \right\|^2 \mathrm{d}\tau' \right]
$$

$$
\leq \frac{7}{2} \int_0^\tau \mathbb{E}_{\omega \sim \breve{p}|_{\mathcal{F}_{t_n}}} \left[ \|\boldsymbol{z}_{t_n, \tau'}\|^2 \right] \mathrm{d}\tau' + \left( \frac{1}{2} \mathbb{E} \left[ \left\| \widehat{\boldsymbol{y}}_{t_n, \tau}^{(0)} \right\|^2 \right] + M_{\boldsymbol{s}}^2 + d \right) \tau
$$

$$
+ 3L_{\boldsymbol{s}}^2 \int_0^\tau \mathbb{E}_{\omega \sim \breve{p}|_{\mathcal{F}_{t_n}}} \left[ \left\| \widehat{\boldsymbol{y}}_{t_n, g_n(\tau')}^{(K)}(\omega) - \widehat{\boldsymbol{y}}_{t_n, g_n(\tau')}^{(K-1)}(\omega) \right\|^2 \right] \mathrm{d}\tau' + 3\mathbb{E}_{\omega \sim \breve{p}|_{\mathcal{F}_{t_n}}} \left[ A_{t_n}(\omega) + B_{t_n}(\omega) \right],
$$

where in the second inequality above, we have used the fact that $s_t^\theta(\cdot)$ is $L_{\boldsymbol{s}}$-Lipschitz for any $t$. By Grönwall's inequality, we have that for any $\tau \in [0, h_n]$

$$
\mathbb{E}_{\omega \sim \breve{p}|_{\mathcal{F}_{t_n}}} \left[ \|\boldsymbol{z}_{t_n, \tau}\|^2 \right] \leq e^{\frac{7}{2}\tau} \left[ \left( \frac{1}{2} \mathbb{E} \left[ \left\| \widehat{\boldsymbol{y}}_{t_n, \tau}^{(0)} \right\|^2 \right] + M_{\boldsymbol{s}}^2 + d \right) \tau \right]
$$

$$
+ 3e^{\frac{7}{2}\tau} \mathbb{E}_{\omega \sim \breve{p}|_{\mathcal{F}_{t_n}}} \left[ A_{t_n}(\omega) + B_{t_n}(\omega) \right] \qquad \text{(B.10)}
$$

$$
+ 3e^{\frac{7}{2}\tau} L_{\boldsymbol{s}}^2 \int_0^\tau \mathbb{E}_{\omega \sim \breve{p}|_{\mathcal{F}_{t_n}}} \left[ \left\| \widehat{\boldsymbol{y}}_{t_n, g_n(\tau')}^{(K)}(\omega) - \widehat{\boldsymbol{y}}_{t_n, g_n(\tau')}^{(K-1)}(\omega) \right\|^2 \right] \mathrm{d}\tau'.
$$

By assumption, $\widehat{\boldsymbol{y}}_{t_n, \tau}^{(0)} = \widehat{\boldsymbol{y}}_{t_n}$ follows the distribution of $\breve{\boldsymbol{x}}_{t_n} \sim \breve{p}_{t_n}$, which allows us to bound the second moment of $\widehat{\boldsymbol{y}}_{t_n}$ for any $n \in [0 : N]$ by Lemma A.8:

$$
\mathbb{E} \left[ \|\widehat{\boldsymbol{y}}_{t_n}\|^2 \right] = \mathbb{E} \left[ \|\breve{\boldsymbol{x}}_{t_n}\|^2 \right] \leq 2d.
$$

Substituting (A.5) into (B.10) then yields that for any $\tau \in [0, h_n]$

$$
\mathbb{E}_{\omega \sim q|_{\mathcal{F}_{t_n}}} \left[ \|\boldsymbol{z}_{t_n, \tau}\|^2 \right] \leq \tau e^{\frac{7}{2}\tau} \left( M_{\boldsymbol{s}}^2 + 2d \right) + 3e^{\frac{7}{2}\tau} \mathbb{E}_{\omega \sim \breve{p}|_{\mathcal{F}_{t_n}}} \left[ A_{t_n}(\omega) + B_{t_n}(\omega) \right]
$$

$$
+ 3\tau e^{\frac{7}{2}\tau} L_{\boldsymbol{s}}^2 \sup_{\tau' \in [0, h_n]} \mathbb{E}_{\omega \sim \breve{p}|_{\mathcal{F}_{t_n}}} \left[ \left\| \widehat{\boldsymbol{y}}_{t_n, \tau'}^{(K)}(\omega) - \widehat{\boldsymbol{y}}_{t_n, \tau'}^{(K-1)}(\omega) \right\|^2 \right].
$$

Taking supremum with respect to $\tau \in [0, h_n]$ on both sides above completes our proof. $\qquad \square$

As utilized in the proof of the existence of solutions of SDEs, the following lemma demonstrates the exponential convergence of the iteration defined in (B.2).

**Lemma B.6** (Exponential convergence of Picard iteration in PIADM-SDE). *For any $n \in [0, N]$, suppose the initialization $\widehat{\boldsymbol{y}}_{t_n}$ in (B.3) of the auxiliary process (B.2) follows the distribution of $\breve{\boldsymbol{x}}_{t_n} \sim \breve{p}_{t_n}$, then the two ending terms $\widehat{\boldsymbol{y}}_{t_n, \tau}^{(K)}$ and $\widehat{\boldsymbol{y}}_{t_n, \tau}^{(K-1)}$ of the sequence $\{\widehat{\boldsymbol{y}}_{t_n, \tau}^{(k)}\}_{k \in [0:K-1]}$ satisfy the following exponential convergence rate*

$$
\sup_{\tau \in [0, h_n]} \mathbb{E}_{\omega \sim \breve{p}|_{\mathcal{F}_{t_n}}} \left[ \left\| \widehat{\boldsymbol{y}}_{t_n, \tau}^{(K)}(\omega) - \widehat{\boldsymbol{y}}_{t_n, \tau}^{(K-1)}(\omega) \right\|_2^2 \right]
$$

$$
\leq \frac{\left( L_{\boldsymbol{s}}^2 h_n e^{2h_n} \right)^{K-1} h e^{\frac{7}{2}h_n} \left( M_{\boldsymbol{s}}^2 + 2d \right)}{1 - 3 \left( L_{\boldsymbol{s}}^2 h_n e^{2h_n} \right)^{K-1} e^{\frac{7}{2}h_n} h_n L_{\boldsymbol{s}}^2} + \frac{3 \left( L_{\boldsymbol{s}}^2 h_n e^{2h_n} \right)^{K-1} e^{\frac{7}{2}h_n} \mathbb{E}_{\omega \sim \breve{p}|_{\mathcal{F}_{t_n}}} \left[ A_{t_n}(\omega) + B_{t_n}(\omega) \right]}{1 - 3 \left( L_{\boldsymbol{s}}^2 h_n e^{2h_n} \right)^{K-1} e^{\frac{7}{2}h_n} h_n L_{\boldsymbol{s}}^2}.
$$

*Proof.* For each $\omega \in \Omega$ conditioned on the filtration $\mathcal{F}_{t_n}$, subtracting (B.2) from the process as defined by

$$
\mathrm{d}\widehat{\boldsymbol{y}}_{t_n, \tau}^{(k)}(\omega) = \left[ \frac{1}{2} \widehat{\boldsymbol{y}}_{t_n, \tau}^{(k)}(\omega) + \boldsymbol{s}_{t_n + g_n(\tau)}^\theta \left( \widehat{\boldsymbol{y}}_{t_n, g_n(\tau)}^{(k-1)}(\omega) \right) \right] \mathrm{d}\tau + \mathrm{d}\boldsymbol{w}_{t_n + \tau}(\omega), \qquad \text{(B.11)}
$$

we have

$$\mathrm{d}\left(\widehat{\boldsymbol{y}}_{t_n,\tau}^{(k+1)}(\omega) - \widehat{\boldsymbol{y}}_{t_n,\tau}^{(k)}(\omega)\right)$$

$$= \left[\frac{1}{2}\left(\widehat{\boldsymbol{y}}_{t_n,\tau}^{(k+1)}(\omega) - \widehat{\boldsymbol{y}}_{t_n,\tau}^{(k)}(\omega)\right) + \boldsymbol{s}_{t_n+g_n(\tau)}^{\theta}\left(\widehat{\boldsymbol{y}}_{t_n,g_n(\tau)}^{(k)}(\omega)\right) - \boldsymbol{s}_{t_n+g_n(\tau)}^{\theta}\left(\widehat{\boldsymbol{y}}_{t_n,g_n(\tau)}^{(k-1)}(\omega)\right)\right]\mathrm{d}\tau,$$

where the diffusion term $\mathrm{d}\boldsymbol{w}_{t_n+\tau}$ cancels each other out. Now we may use the formula above to compute derivative $\frac{\mathrm{d}}{\mathrm{d}\tau'}\left\|\widehat{\boldsymbol{y}}_{t_n,\tau'}^{(k+1)}(\omega) - \widehat{\boldsymbol{y}}_{t_n,\tau'}^{(k)}(\omega)\right\|^2$ explicitly, integrate it from $\tau' = 0$ to $\tau$, and obtain the following inequality

$$\left\|\widehat{\boldsymbol{y}}_{t_n,\tau}^{(k+1)}(\omega) - \widehat{\boldsymbol{y}}_{t_n,\tau}^{(k)}(\omega)\right\|^2$$

$$= \int_0^\tau 2\left(\widehat{\boldsymbol{y}}_{t_n,\tau'}^{(k+1)}(\omega) - \widehat{\boldsymbol{y}}_{t_n,\tau'}^{(k)}(\omega)\right)^\top\left(\boldsymbol{s}_{t_n+g_n(\tau')}^{\theta}\left(\widehat{\boldsymbol{y}}_{t_n,g_n(\tau')}^{(k)}(\omega)\right) - \boldsymbol{s}_{t_n+g_n(\tau')}^{\theta}\left(\widehat{\boldsymbol{y}}_{t_n,g_n(\tau')}^{(k-1)}(\omega)\right)\right)\mathrm{d}\tau'$$

$$+ \int_0^\tau\left\|\widehat{\boldsymbol{y}}_{t_n,\tau'}^{(k+1)}(\omega) - \widehat{\boldsymbol{y}}_{t_n,\tau'}^{(k)}(\omega)\right\|^2\mathrm{d}\tau'$$

$$\leq 2\int_0^\tau\left\|\widehat{\boldsymbol{y}}_{t_n,\tau'}^{(k+1)}(\omega) - \widehat{\boldsymbol{y}}_{t_n,\tau'}^{(k)}(\omega)\right\|^2\mathrm{d}\tau'$$

$$+ \int_0^\tau\left\|\boldsymbol{s}_{t_n+g_n(\tau')}^{\theta}\left(\widehat{\boldsymbol{y}}_{t_n,g_n(\tau')}^{(k)}(\omega)\right) - \boldsymbol{s}_{t_n+g_n(\tau')}^{\theta}\left(\widehat{\boldsymbol{y}}_{t_n,g_n(\tau')}^{(k-1)}(\omega)\right)\right\|^2\mathrm{d}\tau'$$

$$\leq 2\int_0^\tau\left\|\widehat{\boldsymbol{y}}_{t_n,\tau'}^{(k+1)}(\omega) - \widehat{\boldsymbol{y}}_{t_n,\tau'}^{(k)}(\omega)\right\|^2\mathrm{d}\tau' + L_{\boldsymbol{s}}^2\int_0^\tau\left\|\widehat{\boldsymbol{y}}_{t_n,g_n(\tau')}^{(k)}(\omega) - \widehat{\boldsymbol{y}}_{t_n,g_n(\tau')}^{(k-1)}(\omega)\right\|^2\mathrm{d}\tau'.$$

By Grönwall's inequality, we have

$$\left\|\widehat{\boldsymbol{y}}_{t_n,\tau}^{(k+1)}(\omega) - \widehat{\boldsymbol{y}}_{t_n,\tau}^{(k)}(\omega)\right\|^2 \leq L_{\boldsymbol{s}}^2 e^{2\tau}\int_0^\tau\left\|\widehat{\boldsymbol{y}}_{t_n,g_n(\tau')}^{(k)}(\omega) - \widehat{\boldsymbol{y}}_{t_n,g_n(\tau')}^{(k-1)}(\omega)\right\|^2\mathrm{d}\tau'. \tag{B.12}$$

Taking expectation on both sides above further implies that for any $\tau \in [0, h_n]$,

$$\mathbb{E}_{\omega\sim\bar{p}|_{\mathcal{F}_{t_n}}}\left[\left\|\widehat{\boldsymbol{y}}_{t_n,\tau}^{(k+1)}(\omega) - \widehat{\boldsymbol{y}}_{t_n,\tau}^{(k)}(\omega)\right\|^2\right]$$

$$\leq L_{\boldsymbol{s}}^2 e^{2\tau}\int_0^\tau\mathbb{E}_{\omega\sim\bar{p}|_{\mathcal{F}_{t_n}}}\left[\left\|\widehat{\boldsymbol{y}}_{t_n,g_n(\tau')}^{(k)}(\omega) - \widehat{\boldsymbol{y}}_{t_n,g_n(\tau')}^{(k-1)}(\omega)\right\|^2\right]\mathrm{d}\tau' \tag{B.13}$$

$$\leq L_{\boldsymbol{s}}^2\tau e^{2\tau}\sup_{\tau'\in[0,\tau]}\mathbb{E}_{\omega\sim\bar{p}|_{\mathcal{F}_{t_n}}}\left[\left\|\widehat{\boldsymbol{y}}_{t_n,\tau'}^{(k)}(\omega) - \widehat{\boldsymbol{y}}_{t_n,\tau'}^{(k-1)}(\omega)\right\|^2\right].$$

Furthermore, we take supremum over $\tau \in [0, h_n]$ on both sides above and iterate (B.12) over $k \in \mathbb{N}$, which indicates

$$\sup_{\tau\in[0,h_n]}\mathbb{E}_{\omega\sim\bar{p}|_{\mathcal{F}_{t_n}}}\left[\left\|\widehat{\boldsymbol{y}}_{t_n,\tau}^{(k+1)}(\omega) - \widehat{\boldsymbol{y}}_{t_n,\tau}^{(k)}(\omega)\right\|^2\right]$$

$$\leq L_{\boldsymbol{s}}^2 h_n e^{2h_n}\sup_{\tau\in[0,h_n]}\mathbb{E}_{\omega\sim\bar{p}|_{\mathcal{F}_{t_n}}}\left[\left\|\widehat{\boldsymbol{y}}_{t_n,\tau}^{(k)}(\omega) - \widehat{\boldsymbol{y}}_{t_n,\tau'}^{(k-1)}(\omega)\right\|^2\right]$$

$$\leq \left(L_{\boldsymbol{s}}^2 h_n e^{2h_n}\right)^k\sup_{\tau\in[0,h_n]}\mathbb{E}\left[\left\|\widehat{\boldsymbol{y}}_{t_n,\tau}^{(1)}(\omega) - \widehat{\boldsymbol{y}}_{t_n,\tau}^{(0)}(\omega)\right\|^2\right] \tag{B.14}$$

$$\leq \left(L_{\boldsymbol{s}}^2 h_n e^{2h_n}\right)^k h e^{\frac{7}{2}h_n}\left(M_{\boldsymbol{s}}^2 + 2d\right) + 3\left(L_{\boldsymbol{s}}^2 h_n e^{2h_n}\right)^k e^{\frac{7}{2}h_n}\mathbb{E}_{\omega\sim\bar{p}|_{\mathcal{F}_{t_n}}}\left[A_{t_n}(\omega) + B_{t_n}(\omega)\right]$$

$$+ 3\left(L_{\boldsymbol{s}}^2 h_n e^{2h_n}\right)^k e^{\frac{7}{2}h_n} h_n L_{\boldsymbol{s}}^2\sup_{\tau\in[0,h_n]}\mathbb{E}_{\omega\sim\bar{p}|_{\mathcal{F}_{t_n}}}\left[\left\|\widehat{\boldsymbol{y}}_{t_n,\tau}^{(K)}(\omega) - \widehat{\boldsymbol{y}}_{t_n,\tau}^{(K-1)}(\omega)\right\|^2\right],$$

where the last inequality follows from Lemma B.5. By rearranging the inequality above, setting $k = K - 1$ and using the assumption that $L_{\boldsymbol{s}}^2 h_n e^{2h_n} \ll 1$, we obtain

$$\sup_{\tau \in [0, h_n]} \mathbb{E}_{\omega \sim \breve{p}|_{\mathcal{F}_{t_n}}} \left[ \left\| \widehat{\boldsymbol{y}}_{t_n, \tau}^{(K)}(\omega) - \widehat{\boldsymbol{y}}_{t_n, \tau}^{(K-1)}(\omega) \right\|^2 \right]$$

$$\leq \frac{\left( L_{\boldsymbol{s}}^2 h_n e^{2h_n} \right)^{K-1} h e^{\frac{7}{2} h_n} \left( M_{\boldsymbol{s}}^2 + 2d \right) + 3 \left( L_{\boldsymbol{s}}^2 h_n e^{2h_n} \right)^{K-1} e^{\frac{7}{2} h_n} \mathbb{E}_{\omega \sim \breve{p}|_{\mathcal{F}_{t_n}}} \left[ A_{t_n}(\omega) + B_{t_n}(\omega) \right]}{1 - 3 \left( L_{\boldsymbol{s}}^2 h_n e^{2h_n} \right)^{K-1} e^{\frac{7}{2} h_n} h_n L_{\boldsymbol{s}}^2},$$
(B.15)

as desired. $\qquad\square$

The following lemma from [107] bounds the expectation of the term $A_{t_n}(\omega)$ in (B.7):

**Lemma B.7** ([107, Section 3.1]). *We have*

$$\mathbb{E}_{\omega \sim \breve{p}|_{\mathcal{F}_{t_n}}} \left[ A_{t_n}(\omega) \right] \lesssim \epsilon d h_n, \quad \text{for } n \in [0 : N - 2], \text{ and } \mathbb{E}_{\omega \sim \breve{p}|_{\mathcal{F}_{t_n}}} \left[ A_{t_{N-1}}(\omega) \right] \lesssim \epsilon d \log \eta^{-1},$$

*where $\eta$ is the parameter for early stopping.*

*Proof.* Notice that

$$\mathbb{E}_{\omega \sim \breve{p}|_{\mathcal{F}_{t_n}}} \left[ A_{t_n}(\omega) \right]$$

$$= \mathbb{E}_{\omega \sim \breve{p}|_{\mathcal{F}_{t_n}}} \left[ \int_0^{h_n} \left\| \nabla \log \breve{p}_{t_n + g_n(\tau)} \big( \widehat{\boldsymbol{y}}_{t_n, g_n(\tau)}^{(K)}(\omega) \big) - \nabla \log \breve{p}_{t_n + \tau} \big( \widehat{\boldsymbol{y}}_{t_n, g_n(\tau)}^{(K)}(\omega) \big) \right\|^2 \mathrm{d}\tau \right]$$

$$= \mathbb{E}_{\omega \sim \breve{p}|_{\mathcal{F}_{t_n}}} \left[ \sum_{m=0}^{M_n} \int_{\tau_{n,m}}^{\tau_{n,m+1}} \left\| \nabla \log \breve{p}_{t_n + \tau_{n,m}} \big( \widehat{\boldsymbol{y}}_{t_n, \tau_{n,m}}^{(K)}(\omega) \big) - \nabla \log \breve{p}_{t_n + \tau} \big( \widehat{\boldsymbol{y}}_{t_n, \tau}^{(K)}(\omega) \big) \right\|^2 \mathrm{d}\tau \right],$$

$$= \sum_{m=0}^{M_n} \int_{\tau_{n,m}}^{\tau_{n,m+1}} \mathbb{E}_{\omega \sim \breve{p}|_{\mathcal{F}_{t_n}}} \left[ \left\| \nabla \log \breve{p}_{t_n + \tau_{n,m}} \big( \bar{\boldsymbol{x}}_{t_n + \tau}(\omega) \big) - \nabla \log \breve{p}_{t_n + \tau} \big( \bar{\boldsymbol{x}}_{t_n + \tau}(\omega) \big) \right\|^2 \right] \mathrm{d}\tau,$$

where for the last equality, we use the fact that the process $\widehat{\boldsymbol{y}}_{t_n, \tau}^{(K)}(\omega)$ follows the backward SDE with the true score function under the measure $\breve{p}$. In the following, we drop the superscript $\omega \sim \breve{p}|_{\mathcal{F}_{t_n}}$ of the expectation for simplicity.

By Lemma A.6 and A.7, we have

$$\mathbb{E} \left[ \left\| \nabla \log \breve{p}_{t_n + \tau_{n,m}} \big( \bar{\boldsymbol{x}}_{t_n + \tau}(\omega) \big) - \nabla \log \breve{p}_{t_n + \tau} \big( \bar{\boldsymbol{x}}_{t_n + \tau}(\omega) \big) \right\|^2 \right]$$

$$\leq \int_0^\tau \left( \frac{1}{2} \mathbb{E} \left[ \| \nabla \log \breve{p}_{t_n + \tau_{n,m}} \big( \bar{\boldsymbol{x}}_{t_n + \tau_{n,m}}(\omega) \big) \|^2 \right] + \mathbb{E} \left[ \| \nabla^2 \log \breve{p}_{t_n + \tau'} \big( \bar{\boldsymbol{x}}_{t_n + \tau'}(\omega) \big) \|_F^2 \right] \right) \mathrm{d}\tau'$$

$$\leq \int_0^\tau \left( \frac{1}{2} d \bar{\sigma}_{\tau'}^{-2} + d \bar{\sigma}_{\tau'}^{-4} \right) \mathrm{d}\tau' + \left( \bar{\sigma}_{t_n + \tau_{n,m}}^{-4} \mathbb{E} \left[ \mathrm{tr} \, \overleftarrow{\boldsymbol{\Sigma}}_{t_n + \tau_{n,m}} \right] - \bar{\sigma}_{t_n + \tau}^{-4} \mathbb{E} \left[ \mathrm{tr} \, \overleftarrow{\boldsymbol{\Sigma}}_{t_n + \tau} \right] \right),$$

Now noticing that

$$\bar{\sigma}_t^2 = \sigma_{T-t}^2 \lesssim T - t,$$

we further have

$$\int_{\tau_{n,m}}^{\tau_{n,m+1}} \mathbb{E} \left[ \left\| \nabla \log \breve{p}_{t_n + \tau_{n,m}} \big( \bar{\boldsymbol{x}}_{t_n + \tau}(\omega) \big) - \nabla \log \breve{p}_{t_n + \tau} \big( \bar{\boldsymbol{x}}_{t_n + \tau}(\omega) \big) \right\|^2 \right] \mathrm{d}\tau$$

$$\lesssim \int_{\tau_{n,m}}^{\tau_{n,m+1}} \int_0^{\tau'} \frac{d}{(T - t_n - \tau_{n,m+1})^2} \mathrm{d}\tau' \mathrm{d}\tau + \frac{\epsilon_{n,m} \left( \mathbb{E} \left[ \mathrm{tr} \, \overleftarrow{\boldsymbol{\Sigma}}_{t_n + \tau_{n,m}} \right] - \mathbb{E} \left[ \mathrm{tr} \, \overleftarrow{\boldsymbol{\Sigma}}_{t_n + \tau_{n,m+1}} \right] \right)}{(T - t_n - \tau_{n,m})^2}$$

$$\lesssim d \frac{\epsilon_{n,m}^2}{(T - t_n - \tau_{n,m+1})^2} + \frac{\epsilon \left( \mathbb{E} \left[ \mathrm{tr} \, \overleftarrow{\boldsymbol{\Sigma}}_{t_n + \tau_{n,m}} \right] - \mathbb{E} \left[ \mathrm{tr} \, \overleftarrow{\boldsymbol{\Sigma}}_{t_n + \tau_{n,m+1}} \right] \right)}{T - t_n - \tau_{n,m}},$$

and thus

$$\sum_{m=0}^{M_n} \int_{\tau_{n,m}}^{\tau_{n,m+1}} \mathbb{E}\left[\left\|\nabla \log \breve{p}_{t_n+\tau_{n,m}}\left(\breve{\boldsymbol{x}}_{t_n+\tau}(\omega)\right) - \nabla \log \breve{p}_{t_n+\tau}\left(\breve{\boldsymbol{x}}_{t_n+\tau}(\omega)\right)\right\|^2\right] \mathrm{d}\tau$$

$$\lesssim d\sum_{m=0}^{M_n} \frac{\epsilon_{n,m}^2}{(T-t_n-\tau_{n,m+1})^2} + \sum_{m=0}^{M_n} \frac{\epsilon}{T-t_n-\tau_{n,m}}\left(\mathbb{E}\left[\mathrm{tr}\,\breve{\boldsymbol{\Sigma}}_{t_n+\tau_{n,m}}\right] - \mathbb{E}\left[\mathrm{tr}\,\breve{\boldsymbol{\Sigma}}_{t_n+\tau_{n,m+1}}\right]\right)$$

$$\leq d\epsilon^2 M_n + \frac{\epsilon\mathbb{E}\left[\mathrm{tr}\,\breve{\boldsymbol{\Sigma}}_{t_n+\tau_{n,0}}\right]}{T-t_n-\tau_{n,0}} + \sum_{m=0}^{M_n} \frac{\epsilon\epsilon_{n,m}\mathbb{E}\left[\mathrm{tr}\,\breve{\boldsymbol{\Sigma}}_{t_n+\tau_{n,m}}\right]}{(T-t_n-\tau_{n,m+1})(T-t_n-\tau_{n,m})}$$

$$\leq d\epsilon^2 M_n + \epsilon d + d\epsilon^2 M_n \lesssim d\epsilon^2 M_n.$$

For $n \in [0, N-2]$, we have $M_n\epsilon = h_n$ and thus $\mathbb{E}_{\omega\sim\breve{p}|_{\mathcal{F}_{t_n}}}\left[A_{t_n}(\omega)\right] \lesssim \epsilon d h_n$, and for $n = N-1$, we have

$$M_N \lesssim \int_\eta^h \frac{1}{\epsilon\tau}\mathrm{d}\tau = \log\eta^{-1}\epsilon^{-1}$$

and thus $\mathbb{E}_{\omega\sim\breve{p}|_{\mathcal{F}_{t_n}}}\left[A_{t_{N-1}}(\omega)\right] \lesssim \epsilon^2 dM_n \lesssim \epsilon d \log\eta^{-1}.$ $\qquad\square$

## B.3 Overall Error Bound

*Proof of Theorem 3.3.* We first continue the computation in (B.6) and (B.7):

$$D_{\mathrm{KL}}(\breve{p}_{t_{n+1}}\|\widehat{q}_{t_{n+1}}) \leq D_{\mathrm{KL}}(\breve{p}_{t_n}\|\widehat{q}_{t_n}) + \mathbb{E}_{\omega\sim\breve{p}|_{\mathcal{F}_{t_n}}}\left[\frac{1}{2}\int_0^{h_n}\|\boldsymbol{\delta}_{t_n}(\tau,\omega)\|^2\mathrm{d}\tau\right]$$

$$\leq D_{\mathrm{KL}}(\breve{p}_{t_n}\|\widehat{q}_{t_n}) + 3\mathbb{E}_{\omega\sim\breve{p}|_{\mathcal{F}_{t_n}}}\left[A_{t_n}(\omega) + B_{t_n}(\omega)\right]$$

$$+3\mathbb{E}_{\omega\sim\breve{p}|_{\mathcal{F}_{t_n}}}\left[\int_0^{h_n}\left\|\boldsymbol{s}_{t_n+g_n(\tau)}^\theta\left(\widehat{\boldsymbol{y}}_{t_n,g_n(\tau)}^{(K)}(\omega)\right) - \boldsymbol{s}_{t_n+g_n(\tau)}^\theta\left(\widehat{\boldsymbol{y}}_{t_n,g_n(\tau)}^{(K-1)}(\omega)\right)\right\|^2\mathrm{d}\tau\right]$$

$$\leq D_{\mathrm{KL}}(\breve{p}_{t_n}\|\widehat{q}_{t_n}) + 3\mathbb{E}_{\omega\sim\breve{p}|_{\mathcal{F}_{t_n}}}\left[A_{t_n}(\omega) + B_{t_n}(\omega) + L_{\boldsymbol{s}}^2\int_0^{h_n}\left\|\widehat{\boldsymbol{y}}_{t_n,g_n(\tau)}^{(K)}(\omega) - \widehat{\boldsymbol{y}}_{t_n,g_n(\tau)}^{(K-1)}(\omega)\right\|^2\mathrm{d}\tau\right]$$

$$\leq D_{\mathrm{KL}}(\breve{p}_{t_n}\|\widehat{q}_{t_n}) + 3\mathbb{E}_{\omega\sim\breve{p}|_{\mathcal{F}_{t_n}}}\left[A_{t_n}(\omega) + B_{t_n}(\omega) + h_nL_{\boldsymbol{s}}^2\sup_{\tau\in[0,h_n]}\left\|\widehat{\boldsymbol{y}}_{t_n,\tau}^{(K)}(\omega) - \widehat{\boldsymbol{y}}_{t_n,\tau}^{(K-1)}(\omega)\right\|^2\right].$$

Then plugging in the result of Lemma B.6, we have

$$D_{\mathrm{KL}}(\breve{p}_{t_{n+1}}\|\widehat{q}_{t_{n+1}})$$

$$\leq D_{\mathrm{KL}}(\breve{p}_{t_n}\|\widehat{q}_{t_n}) + 3\mathbb{E}_{\omega\sim\breve{p}|_{\mathcal{F}_{t_n}}}\left[A_{t_n}(\omega) + B_{t_n}(\omega)\right] + 3h_nL_{\boldsymbol{s}}^2\frac{\left(L_{\boldsymbol{s}}^2 h_n e^{2h_n}\right)^{K-1}he^{\frac{7}{2}h_n}\left(M_{\boldsymbol{s}}^2 + 2d\right)}{1 - 3\left(L_{\boldsymbol{s}}^2 h_n e^{2h_n}\right)^{K-1}e^{\frac{7}{2}h_n}h_nL_{\boldsymbol{s}}^2}$$

$$+h_nL_{\boldsymbol{s}}^2\frac{9\left(L_{\boldsymbol{s}}^2 h_n e^{2h_n}\right)^{K-1}e^{\frac{7}{2}h_n}\mathbb{E}_{\omega\sim\breve{p}|_{\mathcal{F}_{t_n}}}\left[A_{t_n}(\omega) + B_{t_n}(\omega)\right]}{1 - 3\left(L_{\boldsymbol{s}}^2 h_n e^{2h_n}\right)^{K-1}e^{\frac{7}{2}h_n}h_nL_{\boldsymbol{s}}^2}$$

$$\lesssim D_{\mathrm{KL}}(\breve{p}_{t_n}\|\widehat{q}_{t_n}) + \frac{1 + e^{-K}h_n e^{h_n}}{1 - e^{-K}h_n e^{h_n}}\mathbb{E}_{\omega\sim\breve{p}|_{\mathcal{F}_{t_n}}}\left[A_{t_n}(\omega) + B_{t_n}(\omega)\right] + e^{-K}h_n^2 e^{h_n}d$$

$$\lesssim D_{\mathrm{KL}}(\breve{p}_{t_n}\|\widehat{q}_{t_n}) + \mathbb{E}_{\omega\sim\breve{p}|_{\mathcal{F}_{t_n}}}\left[A_{t_n}(\omega) + B_{t_n}(\omega)\right] + e^{-K}h_n^2 e^{h_n}d,$$

where we used the assumption that $L_{\boldsymbol{s}}^2 h_n e^{\frac{7}{2}h_n} \ll 1$.

The term $\sum_{n=0}^{N-1} \mathbb{E}_{\omega \sim \breve{p}|_{\mathcal{F}_{t_n}}}[B_{t_n}(\omega)]$ is bounded by Assumption 3.1 as

$$\sum_{n=0}^{N-1} \mathbb{E}_{\omega \sim \breve{p}|_{\mathcal{F}_{t_n}}}[B_{t_n}(\omega)]$$

$$\leq \mathbb{E}_{\omega \sim \breve{p}|_{\mathcal{F}_{t_n}}} \left[ \sum_{n=0}^{N-1} \int_0^{h_n} \left\| s_{t_n+g_n(\tau)}^\theta \big(\widehat{\boldsymbol{y}}_{t_n,g_n(\tau)}^{(K)}(\omega)\big) - \nabla \log \breve{p}_{t_n+g_n(\tau)}\big(\widehat{\boldsymbol{y}}_{t_n,g_n(\tau)}^{(K)}(\omega)\big) \right\|^2 \mathrm{d}\tau \right]$$

$$= \mathbb{E}_{\omega \sim \breve{p}|_{\mathcal{F}_{t_n}}} \left[ \sum_{n=0}^{N-1} \sum_{m=0}^{M_n-1} \epsilon_{n,m} \left\| s_{t_n+\tau_{n,m}}^\theta \big(\widehat{\boldsymbol{y}}_{t_n,\tau_{n,m}}^{(K)}(\omega)\big) - \nabla \log \breve{p}_{t_n+\tau_{n,m}}\big(\widehat{\boldsymbol{y}}_{t_n,\tau_{n,m}}^{(K)}(\omega)\big) \right\|^2 \right]$$

$$= \mathbb{E}_{\omega \sim \breve{p}|_{\mathcal{F}_{t_n}}} \left[ \sum_{n=0}^{N-1} \sum_{m=0}^{M_n-1} \epsilon_{n,m} \left\| s_{t_n+\tau_{n,m}}^\theta \big(\breve{\bar{\boldsymbol{x}}}_{t_n+\tau}(\omega)\big) - \nabla \log \breve{p}_{t_n+\tau_{n,m}}\big(\breve{\bar{\boldsymbol{x}}}_{t_n+\tau}(\omega)\big) \right\|^2 \right] \leq \delta_2^2,$$

where the last equality is because the process $\widehat{\boldsymbol{y}}_{t_n,\tau}^{(K)}(\omega)$ under measure $\breve{p}$ follows the backward SDE (B.5).

Thus, by Theorem A.1 and plugging in the iteration relations above

$$D_{\mathrm{KL}}(p_\eta \| \widehat{q}_{t_N}) = D_{\mathrm{KL}}(\breve{p}_{t_N} \| \widehat{q}_{t_N})$$

$$\leq D_{\mathrm{KL}}(\breve{p}_0 \| \widehat{q}_0) + \sum_{n=0}^{N-1} \left( \mathbb{E}_{\omega \sim \breve{p}|_{\mathcal{F}_{t_n}}}[A_{t_n}(\omega) + B_{t_n}(\omega)] + e^{-K} h_n^2 e^{h_n} d \right)$$

$$\leq D_{\mathrm{KL}}(\breve{p}_0 \| \widehat{q}_0) + \sum_{n=0}^{N-2} \epsilon d h_n + \epsilon d \log \eta^{-1} + \sum_{n=0}^{N-1} \mathbb{E}_{\omega \sim \breve{p}|_{\mathcal{F}_{t_n}}}[B_{t_n}(\omega)] + e^{-K} h_n^2 e^{h_n} d N$$

$$\leq d e^{-T} + \epsilon d (T + \log \eta^{-1}) + \delta_2^2 + e^{-K} d T \leq d e^{-T} + \epsilon d T + \delta^2 + e^{-K} d T,$$

as $T \gtrsim \log \eta^{-1}$, $h_n \lesssim 1$, and $\delta_2 \lesssim \delta$, and then it is straightforward to see that the following choices of parameters

$$T = \mathcal{O}(\log(d\delta^{-2})), \quad h = \Theta(1), \quad N = \mathcal{O}\left(\log(d\delta^{-2})\right),$$

$$\epsilon = \Theta\left(d^{-1}\delta^2 \log^{-1}(d\delta^{-2})\right), \quad M = \mathcal{O}\left(d\delta^{-2}\log(d\delta^{-2})\right),$$

$$K = \widetilde{\mathcal{O}}(\log(d\delta^{-2})),$$

would yield an overall error of $\mathcal{O}(\delta^2)$. $\qquad\square$

## C  Details of Probability Flow ODE Implementation

In this section, we provide the details of the parallelized algorithm for the probability flow ODE formulation of diffusion models. We first introduce the algorithm and define the necessary notations, then discuss the error analysis during the predictor and corrector steps, respectively, and finally provide the proof of Theorem 3.5.

### C.1  Algorithm

In the parallelized inference algorithm for diffusion models in the probability flow ODE formulation, we adopt the same discretization scheme as in Section 3.1.1 and the exponential integrator for all updating rules. For each block, we first run a *predictor step*, which consists of running the probability flow ODE in parallel. Then we run a *corrector step*, which runs an underdamped Langevin dynamics in parallel to correct the distribution of the samples. The algorithm is summarized In Algorithm 2.

**Parallelized Predictor Step**  The parallelization strategies in the predictor step are similar to those in the SDE algorithm (Algorithm 1). The only difference here is that instead of applying Picard iteration to the backward SDE as in (3.2), we apply Picard iteration to the probability flow ODE as in (C.3), which does not require i.i.d. samples from standard Gaussian distribution. As shown in Lemma C.3, the update rule in the predictor step (C.1) in Algorithm 2 is equivalent to running

**Algorithm 2:** PIADM-ODE

**Input:** $\widehat{\boldsymbol{y}}_0 \sim \widehat{q}_0 = \mathcal{N}(0, \boldsymbol{I}_d)$, a discretization scheme $(T, (h_n)_{n=1}^N$ and $(\tau_{n,m})_{n\in[1:N],m\in[0:M_n]})$ satisfying (3.1), parameters for the corrector step $(T^\dagger, N^\dagger, h^\dagger, M^\dagger, \epsilon^\dagger)$, the depth of iteration $K$ and $K^\dagger$, the learned NN-based score $\boldsymbol{s}_t^\theta(\cdot)$.

**Output:** A sample $\widehat{\boldsymbol{y}}_T \sim \widehat{q}_T \approx \breve{p}_T$.

1 **for** $n = 0$ **to** $N - 1$ **do**
2    $\triangleright$ Predictor Step (Section C.2)
3    $\widehat{\boldsymbol{y}}_{t_n,\tau_{n,m}}^{(0)} \leftarrow \widehat{\boldsymbol{y}}_{t_n}$ for $m \in [0 : M_n]$;
4    **for** $k = 1$ **to** $K$ **do**
5      $\widehat{\boldsymbol{y}}_{t_n,0}^{(k)} \leftarrow \widehat{\boldsymbol{y}}_{t_n}$;
6      **for** $m = 1$ **to** $M_n$ *in parallel* **do**
7

$$\begin{aligned} \widehat{\boldsymbol{y}}_{t_n,\tau_{n,m}}^{(k)} \leftarrow &\frac{1}{2} e^{\frac{\tau_{n,m}}{2}} \widehat{\boldsymbol{y}}_{t_n,0}^{(k-1)} \\ &+ \frac{1}{2} \sum_{j=0}^{m-1} e^{\frac{\tau_{n,m}-\tau_{n,j+1}}{2}} \left(e^{\epsilon_{n,j}} - 1\right) \boldsymbol{s}_{t_n+\tau_{n,j}}^{\theta}(\widehat{\boldsymbol{y}}_{t_n,\tau_{n,j}}^{(k-1)}) \end{aligned} \qquad \text{(C.1)}$$

8      **end**
9    **end**
10    $\triangleright$ Corrector Step (Section C.3)
11    $\widehat{\boldsymbol{u}}_{t_n,0}^{(0)} \leftarrow \widehat{\boldsymbol{y}}_{t_n,h_n}^{(K)}$ and $\widehat{\boldsymbol{v}}_{t_n,0}^{(0)} \sim \mathcal{N}(0, \boldsymbol{I}_d)$;
12    **for** $n^\dagger = 0$ **to** $N^\dagger - 1$ **do**
13      $(\widehat{\boldsymbol{u}}_{t_n,n^\dagger h^\dagger,m^\dagger \epsilon^\dagger}^{(0)}, \widehat{\boldsymbol{v}}_{t_n,n^\dagger h^\dagger,m^\dagger \epsilon^\dagger}^{(0)}) \leftarrow (\widehat{\boldsymbol{u}}_{t_n,n^\dagger h^\dagger}, \widehat{\boldsymbol{v}}_{t_n,n^\dagger h^\dagger})$ for $m^\dagger \in [0 : M^\dagger]$;
14      **for** $k^\dagger = 1$ **to** $K^\dagger$ **do**
15        $(\widehat{\boldsymbol{u}}_{t_n,n^\dagger h^\dagger,0}^{(k^\dagger)}, \widehat{\boldsymbol{v}}_{t_n,n^\dagger h^\dagger,0}^{(k^\dagger)}) \leftarrow (\widehat{\boldsymbol{u}}_{t_n,n^\dagger h^\dagger}, \widehat{\boldsymbol{v}}_{t_n,n^\dagger h^\dagger})$;
16        $\boldsymbol{\xi}_{j^\dagger} \sim \mathcal{N}\left(\boldsymbol{0}, 2\gamma(1 + \gamma^{-2})(1 - e^{-\gamma\epsilon^\dagger})^2 e^{-2\gamma((M^\dagger - j^\dagger + 1)\epsilon^\dagger)} \boldsymbol{I}_d\right)$ for $j^\dagger \in [0 : M^\dagger]$;
17        **for** $m^\dagger = 1$ **to** $M^\dagger$ *in parallel* **do**
18

$$\begin{aligned} \begin{bmatrix} \widehat{\boldsymbol{u}}_{t_n,n^\dagger h^\dagger,m^\dagger \epsilon^\dagger}^{(k^\dagger)} \\ \widehat{\boldsymbol{v}}_{t_n,n^\dagger h^\dagger,m^\dagger \epsilon^\dagger}^{(k^\dagger)} \end{bmatrix} \leftarrow &\boldsymbol{G}(m^\dagger \epsilon^\dagger) \begin{bmatrix} \widehat{\boldsymbol{u}}_{t_n,n^\dagger h^\dagger,0}^{(k^\dagger-1)} \\ \widehat{\boldsymbol{v}}_{t_n,n^\dagger h^\dagger,0}^{(k^\dagger-1)} \end{bmatrix} \\ &+ \sum_{j^\dagger=0}^{m-1} \boldsymbol{G}((m^\dagger - j^\dagger - 1)\epsilon^\dagger) \left(\boldsymbol{I}_d - \boldsymbol{G}(\epsilon^\dagger)\right) \begin{bmatrix} \boldsymbol{0} \\ \boldsymbol{s}_{t_{n+1}}^{\theta}(\widehat{\boldsymbol{u}}_{t_n,n^\dagger h^\dagger,j^\dagger \epsilon^\dagger}^{(k^\dagger-1)}) \end{bmatrix} \\ &+ \sum_{j^\dagger=0}^{m-1} \boldsymbol{G}((m^\dagger - j^\dagger - 1)\epsilon^\dagger) \begin{bmatrix} \boldsymbol{0} \\ \boldsymbol{\xi}_{j^\dagger} \end{bmatrix}; \end{aligned} \qquad \text{(C.2)}$$

19        **end**
20      **end**
21      $(\widehat{\boldsymbol{u}}_{t_n,(n^\dagger+1)h^\dagger}, \widehat{\boldsymbol{v}}_{t_n,(n^\dagger+1)h^\dagger}) \leftarrow (\widehat{\boldsymbol{u}}_{t_n,n^\dagger h^\dagger,h^\dagger}^{(K^\dagger)}, \widehat{\boldsymbol{v}}_{t_n,n^\dagger h^\dagger,h^\dagger}^{(K^\dagger)})$;
22    **end**
23    $\widehat{\boldsymbol{y}}_{t_{n+1}} \leftarrow \widehat{\boldsymbol{u}}_{t_n,T^\dagger}$;
24 **end**

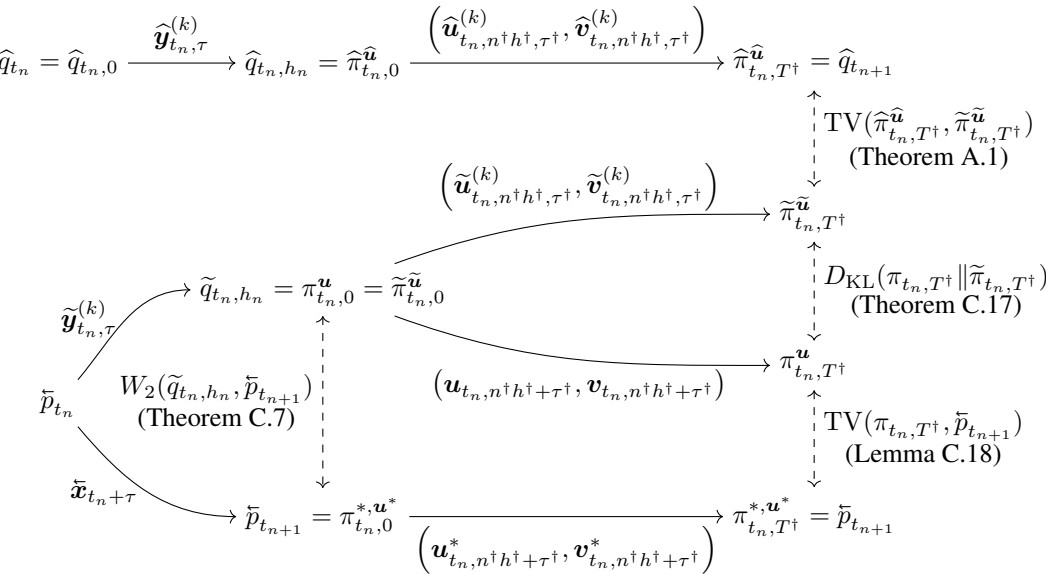

Figure 2: Illustration of the proof pipeline of Theorem 3.5 for PIADM-ODE within the $n$-th block.

the auxiliary predictor process (C.3). The auxiliary predictor process takes in the result from the previous corrector step (or the initialization if $n = 0$) and outputs $\widehat{\boldsymbol{y}}_{t_n,h_n}^{(K)}$ as the initialization for the next corrector step.

**Parallelized Corrector Step**   The parallelization of the underdamped Langevin dynamics is similar to that mentioned in Section 2.2. Given a sample resulting from the predictor step, we initialize the auxiliary corrector process (Definition C.8) which is an underdamped Langevin dynamics with the initialization $\widehat{\boldsymbol{u}}_{t_n,0} = \boldsymbol{y}_{t_n,h_n}^{(K)}$ and the augmented variable $\widehat{\boldsymbol{v}}_{t_n,0} \sim \mathcal{N}(0, \boldsymbol{I}_d)$ representing the momentum.

We run the underdamped Langevin dynamics for time $T^\dagger$, which is set to be of order $\Omega(1)$ so that it is large enough to correct the distribution of the samples (*cf.* Lemma C.18) while being comparably short to ensure numerical stability (*cf.* Theorem C.17). Following a similar strategy as in Section 2.2 and in Algorithm 1, we further divide the time horizon $T^\dagger$ into $N^\dagger$ blocks with step size $h^\dagger$, and for each block the block length $h^\dagger$ into $M^\dagger$ steps with step size $\epsilon^\dagger$. Within each block, we run the underdamped Langevin dynamics in parallel for $K^\dagger$ iterations. As shown in Lemma C.9, the update rule in the corrector step (C.2) in Algorithm 2 is equivalent to running the auxiliary corrector process (C.11).

In the following subsections, we proceed to provide theoretical guarantees for the algorithm.

## C.2   Parallelized Predictor Step

**Definition C.1** (Auxiliary Predictor Process). *For any $n \in [0 : N - 1]$, we define the auxiliary predictor process $(\widehat{\boldsymbol{y}}_{t_n,\tau}^{(k)})_{\tau \in [0,h_n]}$ as the solution to the following ODE recursively for $k \in [0 : K - 1]$:*

$$\mathrm{d}\widehat{\boldsymbol{y}}_{t_n,\tau}^{(k+1)} = \left[ \frac{1}{2}\widehat{\boldsymbol{y}}_{t_n,\tau}^{(k+1)} + \frac{1}{2}\boldsymbol{s}_{t_n+g_n(\tau)}^\theta \left( \widehat{\boldsymbol{y}}_{t_n,g_n(\tau)}^{(k)} \right) \right] \mathrm{d}\tau, \tag{C.3}$$

*with the initial condition*

$$\widehat{\boldsymbol{y}}_{t_n,\tau}^{(0)} \equiv \widehat{\boldsymbol{y}}_{t_n} \text{ for } \tau \in [0, h_n], \quad \text{and} \quad \boldsymbol{y}_{t_n,0}^{(k)} \equiv \widehat{\boldsymbol{y}}_{t_n} \text{ for } k \in [1 : K] \tag{C.4}$$

*where $\widehat{\boldsymbol{y}}_{t_n} = \widehat{\boldsymbol{u}}_{t_{n-1},N^\dagger h^\dagger}$ if $n \in [1 : N - 1]$ and $\widehat{\boldsymbol{y}}_{t_0} \sim \mathcal{N}(0, \boldsymbol{I}_d)$. We will also denote the probability distribution of $\widehat{\boldsymbol{y}}_{t_n,\tau}^{(K)}$ as $\widehat{q}_{t_n,\tau}$.*

**Definition C.2** (Interpolating Process). *For any $n \in [0 : N-1]$, we define the interpolating process $(\widetilde{\boldsymbol{y}}_{t_n,\tau}^{(k)})_{\tau \in [0,h_n]}$ as the solution to the following ODE recursively for $k \in [0 : K-1]$:*

$$\mathrm{d}\widetilde{\boldsymbol{y}}_{t_n,\tau}^{(k+1)} = \left[ \frac{1}{2}\widetilde{\boldsymbol{y}}_{t_n,\tau}^{(k+1)} + \frac{1}{2}\boldsymbol{s}_{t_n+g_n(\tau)}^{\theta}\left(\widetilde{\boldsymbol{y}}_{t_n,g_n(\tau)}^{(k)}\right) \right]\mathrm{d}\tau, \tag{C.5}$$

*with initial condition*

$$\widetilde{\boldsymbol{y}}_{t_n,\tau}^{(0)} \equiv \widetilde{\boldsymbol{y}}_{t_n,0}^{(0)} \quad \text{for } \tau \in [0,h_n], \text{ and } \widetilde{\boldsymbol{y}}_{t_n,0}^{(k)} \equiv \widetilde{\boldsymbol{y}}_{t_n,0}^{(0)} \quad \text{for } k \in [1:K],$$

*where $\widetilde{\boldsymbol{y}}_{t_n,0}^{(0)} \sim \breve{p}_{t_n}$. We will also denote the probability distribution of $\widetilde{\boldsymbol{y}}_{t_n,\tau}^{(K)}$ as $\widetilde{q}_{t_n,\tau}$.*

Similar to the equivalence between (3.4) and (B.2), we have the following lemma:

**Lemma C.3** (Equivalence between (C.1) and (C.3)). *For any $n \in [0 : N-1]$, the update rule (C.1) in Algorithm 2 is equivalent to the exact solution of (C.3) for any $k \in [0 : K-1]$ and $\tau \in [0, h_n]$.*

*Proof.* Rewriting (C.3) and multiplying $e^{-\frac{\tau}{2}}$ on both sides yield

$$\mathrm{d}\left[ e^{-\frac{\tau}{2}}\widehat{\boldsymbol{y}}_{t_n,\tau}^{(k+1)} \right] = e^{-\frac{\tau}{2}}\left[ \mathrm{d}\widehat{\boldsymbol{y}}_{t_n,\tau}^{(k+1)} - \frac{1}{2}\widehat{\boldsymbol{y}}_{t_n,\tau}^{(k+1)}\mathrm{d}\tau \right] = \frac{e^{-\frac{\tau}{2}}}{2}\boldsymbol{s}_{t_n+g_n(\tau)}^{\theta}\left(\widehat{\boldsymbol{y}}_{t_n,g_n(\tau)}^{(k)}\right)\mathrm{d}\tau$$

Integrating on both sides from $0$ to $\tau$ implies

$$e^{-\frac{\tau}{2}}\widehat{\boldsymbol{y}}_{t_n,\tau}^{(k+1)} - \widehat{\boldsymbol{y}}_{t_n,0}^{(k+1)} = \int_0^{\tau} \frac{e^{-\frac{\tau}{2}}}{2}\boldsymbol{s}_{t_n+g_n(\tau')}^{\theta}\left(\widehat{\boldsymbol{y}}_{t_n,g_n(\tau')}^{(k)}\right)\mathrm{d}\tau'$$

$$=\frac{1}{2}\sum_{m=0}^{M_n} \int_{\tau \wedge t_{n,m}}^{\tau \wedge \tau_{n,m+1}} e^{-\frac{\tau'}{2}}\boldsymbol{s}_{t_n+\tau_{n,m}}^{\theta}\left(\boldsymbol{y}_{t_n,\tau_{n,m}}^{(k)}\right)\mathrm{d}\tau'$$

$$=\sum_{m=0}^{M_n} \left( e^{-\frac{\tau \wedge \tau_{n,m}}{2}} - e^{-\frac{\tau \wedge \tau_{n,m+1}}{2}} \right)\boldsymbol{s}_{t_n+\tau_{n,j}}^{\theta}\left(\widehat{\boldsymbol{y}}_{t_n,\tau_{n,m}}^{(k)}\right),$$

and then multiplying $e^{\frac{\tau}{2}}$ on both sides above yields

$$\widehat{\boldsymbol{y}}_{t_n,\tau}^{(k+1)} = e^{\frac{\tau}{2}}\widehat{\boldsymbol{y}}_{t_n,0}^{(k+1)} + \sum_{m=0}^{M_n} \left( e^{\frac{\tau \wedge \tau_{n,m+1} - \tau \wedge \tau_{n,m}}{2}} - 1 \right) e^{\frac{0 \vee (\tau - \tau_{n,m+1})}{2}}\boldsymbol{s}_{t_n+\tau_{n,m}}^{\theta}\left(\widehat{\boldsymbol{y}}_{t_n,\tau_{n,m}}^{(k)}\right).$$

Plugging in $\tau = \tau_{n,m}$ gives us (C.1), as desired. $\qquad\square$

**Lemma C.4** (Error between the interpolating process and the true process). *Under the Picard iteration, we have that the ending process $\{\widehat{\boldsymbol{y}}_{t_n,\tau}^{(K)}\}_{\tau \in [0,h_n]}$ satisfies the following exponential convergence rate*

$$\sup_{\tau \in [0,h_n]} \mathbb{E}\left[ \left\| \widetilde{\boldsymbol{y}}_{t_n,\tau}^{(K)} - \bar{\boldsymbol{x}}_{t_n+\tau} \right\|^2 \right] \leq 3d \left( \frac{h_n^2 e^{h_n+\frac{3}{2}}L_{\boldsymbol{s}}^2}{2} \right)^K + \frac{e^{h_n+\frac{3}{2}}h_n/2}{1 - h_n^2 e^{h_n+\frac{3}{2}}L_{\boldsymbol{s}}^2/2}\left( h_n\delta_{\infty}^2 + \mathbb{E}[D_{t_n}] \right),$$

*where*

$$D_{t_n} := \int_0^{h_n} \left\| \boldsymbol{s}_{t_n+g_n(\tau')}^{\theta}\left(\bar{\boldsymbol{x}}_{t_n+g_n(\tau')}\right) - \boldsymbol{s}_{t_n+\tau'}^{\theta}\left(\bar{\boldsymbol{x}}_{t_n+\tau'}\right) \right\|^2 \mathrm{d}\tau'.$$

*Proof.* Recall that the backward true process $\{\bar{\boldsymbol{x}}_{t_n+\tau}\}_{\tau \in [0,h_n]}$ satisfies the following backward SDE within one block

$$\mathrm{d}\bar{\boldsymbol{x}}_{t_n+\tau} = \left[ \frac{1}{2}\bar{\boldsymbol{x}}_{t_n+\tau} + \frac{1}{2}\nabla\log\breve{p}_{t_n+\tau}(\bar{\boldsymbol{x}}_{t_n+\tau}) \right]\mathrm{d}\tau. \tag{C.6}$$

By subtracting (C.6) from (C.5), we obtain that

$$\begin{aligned}
\frac{\mathrm{d}}{\mathrm{d}\tau}\left( \widetilde{\boldsymbol{y}}_{t_n,\tau}^{(k+1)} - \bar{\boldsymbol{x}}_{t_n+\tau} \right) = {} & \frac{1}{2}\left[ \widetilde{\boldsymbol{y}}_{t_n,\tau}^{(k+1)} - \bar{\boldsymbol{x}}_{t_n+\tau} \right] \\
& + \frac{1}{2}\left[ \boldsymbol{s}_{t_n+g_n(\tau)}^{\theta}\left(\widetilde{\boldsymbol{y}}_{t_n,g_n(\tau)}^{(k)}\right) - \boldsymbol{s}_{t_n+g_n(\tau)}^{\theta}(\bar{\boldsymbol{x}}_{t_n+g_n(\tau)}) \right] \\
& + \frac{1}{2}\left[ \boldsymbol{s}_{t_n+g_n(\tau)}^{\theta}(\bar{\boldsymbol{x}}_{t_n+g_n(\tau)}) - \nabla\log\breve{p}_{t_n+g_n(\tau)}(\bar{\boldsymbol{x}}_{t_n+g_n(\tau)}) \right] \\
& + \frac{1}{2}\left[ \nabla\log\breve{p}_{t_n+g_n(\tau)}(\bar{\boldsymbol{x}}_{t_n+g_n(\tau)}) - \nabla\log\breve{p}_{t_n+\tau}(\bar{\boldsymbol{x}}_{t_n+\tau}) \right].
\end{aligned} \tag{C.7}$$

Then by
$$\mathrm{d}\left\|\widetilde{\boldsymbol{y}}_{t_n,\tau'}^{(k+1)} - \bar{\boldsymbol{x}}_{t_n+\tau'}\right\|^2 = 2\left(\widetilde{\boldsymbol{y}}_{t_n,\tau'}^{(k+1)} - \bar{\boldsymbol{x}}_{t_n+\tau'}\right)^\top \mathrm{d}\left(\widetilde{\boldsymbol{y}}_{t_n,\tau'}^{(k+1)} - \bar{\boldsymbol{x}}_{t_n+\tau'}\right),$$

and integrating for $\tau' \in [0, h_n]$, we have

$$\left\|\widetilde{\boldsymbol{y}}_{t_n,\tau}^{(k+1)} - \bar{\boldsymbol{x}}_{t_n+\tau}\right\|^2$$
$$= \int_0^\tau \left(\widetilde{\boldsymbol{y}}_{t_n,\tau'}^{(k+1)} - \bar{\boldsymbol{x}}_{t_n+\tau'}\right)^\top \left(\boldsymbol{s}_{t_n+g_n(\tau')}^\theta\big(\widetilde{\boldsymbol{y}}_{t_n,g_n(\tau')}^{(k)}\big) - \boldsymbol{s}_{t_n+g_n(\tau')}^\theta\big(\bar{\boldsymbol{x}}_{t_n+g_n(\tau')}\big)\right)\mathrm{d}\tau'$$
$$+ \int_0^\tau \left(\widetilde{\boldsymbol{y}}_{t_n,\tau'}^{(k+1)} - \bar{\boldsymbol{x}}_{t_n+\tau'}\right)^\top \left(\boldsymbol{s}_{t_n+g_n(\tau')}^\theta\big(\bar{\boldsymbol{x}}_{t_n+g_n(\tau')}\big) - \nabla\log\breve{p}_{t_n+g_n(\tau')}\big(\bar{\boldsymbol{x}}_{t_n+g_n(\tau')}\big)\right)\mathrm{d}\tau'$$
$$+ \int_0^\tau \left(\widetilde{\boldsymbol{y}}_{t_n,\tau'}^{(k+1)} - \bar{\boldsymbol{x}}_{t_n+\tau'}\right)^\top \left(\nabla\log\breve{p}_{t_n+g_n(\tau')}\big(\bar{\boldsymbol{x}}_{t_n+g_n(\tau')}\big) - \nabla\log\breve{p}_{t_n+\tau'}\big(\bar{\boldsymbol{x}}_{t_n+\tau'}\big)\right)\mathrm{d}\tau'$$
$$+ \int_0^\tau \left\|\widetilde{\boldsymbol{y}}_{t_n,\tau'}^{(k+1)} - \bar{\boldsymbol{x}}_{t_n+\tau'}\right\|^2\mathrm{d}\tau'.$$

Using AM-GM inequality and taking expectations on both sides, we further upper bound the summation above as

$$\mathbb{E}\left[\left\|\widetilde{\boldsymbol{y}}_{t_n,\tau}^{(k+1)} - \bar{\boldsymbol{x}}_{t_n+\tau}\right\|^2\right]$$
$$\leq \left(1 + \frac{3}{2h_n}\right)\int_0^\tau \mathbb{E}\left[\left\|\widetilde{\boldsymbol{y}}_{t_n,\tau'}^{(k+1)} - \bar{\boldsymbol{x}}_{t_n+\tau'}\right\|^2\right]\mathrm{d}\tau'$$
$$+ \frac{h_n}{2}\int_0^\tau \mathbb{E}\left[\left\|\boldsymbol{s}_{t_n+g_n(\tau')}^\theta\big(\widetilde{\boldsymbol{y}}_{t_n,g_n(\tau')}^{(k)}\big) - \boldsymbol{s}_{t_n+g_n(\tau')}^\theta\big(\bar{\boldsymbol{x}}_{t_n+g_n(\tau')}\big)\right\|^2\right]\mathrm{d}\tau'$$
$$+ \frac{h_n}{2}\int_0^\tau \mathbb{E}\left[\left\|\boldsymbol{s}_{t_n+g_n(\tau')}^\theta\big(\bar{\boldsymbol{x}}_{t_n+g_n(\tau')}\big) - \nabla\log\breve{p}_{t_n+g_n(\tau')}\big(\bar{\boldsymbol{x}}_{t_n+g_n(\tau')}\big)\right\|^2\right]\mathrm{d}\tau'$$
$$+ \frac{h_n}{2}\mathbb{E}\Bigg[\underbrace{\int_0^\tau \left\|\nabla\log\breve{p}_{t_n+g_n(\tau')}\big(\bar{\boldsymbol{x}}_{t_n+g_n(\tau')}\big) - \nabla\log\breve{p}_{t_n+\tau'}\big(\bar{\boldsymbol{x}}_{t_n+\tau'}\big)\right\|^2\mathrm{d}\tau'}_{\leq D_{t_n}}\Bigg]$$
$$\leq \left(1 + \frac{3}{2h_n}\right)\int_0^\tau \mathbb{E}\left[\left\|\widetilde{\boldsymbol{y}}_{t_n,\tau'}^{(k+1)} - \bar{\boldsymbol{x}}_{t_n+\tau'}\right\|^2\right]\mathrm{d}\tau' + \frac{h_n}{2}\left(\tau\delta_\infty^2 + \mathbb{E}\left[D_{t_n}\right]\right)$$
$$+ \frac{L_{\boldsymbol{s}}^2 h_n}{2}\int_0^\tau \mathbb{E}\left[\left\|\widetilde{\boldsymbol{y}}_{t_n,g_n(\tau')}^{(k)} - \bar{\boldsymbol{x}}_{t_n+g_n(\tau')}\right\|^2\right]\mathrm{d}\tau',$$

where the last equality is by Assumption 3.1'.

Applying Grönwall's inequality, we have

$$\mathbb{E}\left[\left\|\widetilde{\boldsymbol{y}}_{t_n,\tau}^{(k+1)} - \bar{\boldsymbol{x}}_{t_n+\tau}\right\|^2\right]$$
$$\leq \frac{e^{\left(1+\frac{3}{2h_n}\right)\tau}L_{\boldsymbol{s}}^2 h_n}{2}\int_0^\tau \mathbb{E}\left[\left\|\widetilde{\boldsymbol{y}}_{t_n,g_n(\tau')}^{(k)} - \bar{\boldsymbol{x}}_{t_n+g_n(\tau')}\right\|^2\right]\mathrm{d}\tau' + \frac{e^{\left(1+\frac{3}{2h_n}\right)\tau}h_n}{2}\left(\tau\delta_\infty^2 + \mathbb{E}[D_{t_n}]\right)$$
$$\leq \frac{\tau e^{\left(1+\frac{3}{2h_n}\right)\tau}L_{\boldsymbol{s}}^2 h_n}{2}\sup_{\tau'\in[0,\tau]}\mathbb{E}\left[\left\|\widetilde{\boldsymbol{y}}_{t_n,\tau'}^{(k)} - \bar{\boldsymbol{x}}_{t_n+\tau'}\right\|^2\right] + \frac{e^{\left(1+\frac{3}{2h_n}\right)\tau}h_n}{2}\left(\tau\delta_\infty^2 + \mathbb{E}[D_{t_n}]\right),$$
$$\tag{C.8}$$

and by taking supremum

$$\sup_{\tau\in[0,h_n]}\mathbb{E}\left[\left\|\widetilde{\boldsymbol{y}}_{t_n,\tau}^{(k+1)} - \bar{\boldsymbol{x}}_{t_n+\tau}\right\|^2\right]$$
$$\leq \frac{h_n^2 e^{h_n+\frac{3}{2}}L_{\boldsymbol{s}}^2}{2}\sup_{\tau'\in[0,\tau]}\mathbb{E}\left[\left\|\widetilde{\boldsymbol{y}}_{t_n,\tau'}^{(k)} - \bar{\boldsymbol{x}}_{t_n+\tau'}\right\|^2\right] + \frac{e^{h_n+\frac{3}{2}}h_n}{2}\left(h_n\delta_\infty^2 + \mathbb{E}[D_{t_n}]\right)$$
$$\tag{C.9}$$

Given that constant $h_n$ is sufficiently small, which ensures $L_{\boldsymbol{s}}^2 h_n e^{\frac{5}{2}h_n} \ll 1$, iterating the above inequality for $k \in [0:K-1]$ gives us that

$$\sup_{\tau \in [0,h_n]} \mathbb{E}\left[\left\|\widetilde{\boldsymbol{y}}_{t_n,\tau}^{(K)} - \bar{\bar{\boldsymbol{x}}}_{t_n+\tau}\right\|^2\right]$$

$$\leq \left(\frac{h_n^2 e^{h_n + \frac{3}{2}} L_{\boldsymbol{s}}^2}{2}\right)^K \sup_{\tau \in [0,h_n]} \mathbb{E}\left[\left\|\widetilde{\boldsymbol{y}}_{t_n,\tau}^{(0)} - \bar{\bar{\boldsymbol{x}}}_{t_n+\tau}\right\|^2\right] + \frac{e^{h_n + \frac{3}{2}} h_n/2}{1 - h_n^2 e^{h_n + \frac{3}{2}} L_{\boldsymbol{s}}^2/2}\left(h_n \delta_\infty^2 + \mathbb{E}[D_{t_n}]\right),$$

Notice that by Lemma A.8, we have

$$\mathbb{E}\left[\left\|\widehat{\boldsymbol{y}}_{t_n,\tau}^{(0)} - \bar{\bar{\boldsymbol{x}}}_{t_n+\tau}\right\|^2\right] = \mathbb{E}\left[\left\|\bar{\bar{\boldsymbol{x}}}_{t_n} - \bar{\bar{\boldsymbol{x}}}_{t_n+\tau}\right\|^2\right] \leq 3d,$$

substituting which into (C.9) then gives us that

$$\sup_{\tau \in [0,h_n]} \mathbb{E}\left[\left\|\widetilde{\boldsymbol{y}}_{t_n,\tau}^{(K)} - \bar{\bar{\boldsymbol{x}}}_{t_n+\tau}\right\|^2\right] \leq 3d\left(\frac{h_n^2 e^{h_n + \frac{3}{2}} L_{\boldsymbol{s}}^2}{2}\right)^K + \frac{e^{h_n + \frac{3}{2}} h_n/2}{1 - h_n^2 e^{h_n + \frac{3}{2}} L_{\boldsymbol{s}}^2/2}\left(h_n \delta_\infty^2 + \mathbb{E}[D_{t_n}]\right),$$

as desired. $\qquad\square$

Now it remains to bound $C_{t_n}$ and $D_{t_n}$ in Lemma C.4. We first bound $D_{t_n}$ using the following lemma:

**Lemma C.5.** *For any $n \in [0:N-1]$, we have that*

$$\mathbb{E}\left[D_{t_n}\right] \lesssim d\epsilon^2 h_n.$$

*Proof.* For any $n \in [0:N-2]$, we have $T - t_{n+1} \gtrsim \mathcal{O}(1)$ and thus by [111, Corollary 1] that

$$\mathbb{E}\left[\left\|\nabla \log \breve{p}_{t_{N-1}+\tau_{n,m}}(\bar{\bar{\boldsymbol{x}}}_{t_{N-1}+\tau_{n,m}}) - \nabla \log \breve{p}_{t_{N-1}+\tau'}(\bar{\bar{\boldsymbol{x}}}_{t_{N-1}+\tau'})\right\|^2\right] \lesssim d\epsilon_{n,m}^2,$$

for any $\tau' \in [\tau_{n,m}, \tau_{n,m+1}]$, and thus

$$\mathbb{E}\left[D_{t_n}\right] = \int_0^{h_n} \mathbb{E}\left[\left\|\nabla \log \breve{p}_{t_n+g_n(\tau')}(\bar{\bar{\boldsymbol{x}}}_{t_n+g_n(\tau')}) - \nabla \log \breve{p}_{t_n+\tau'}(\bar{\bar{\boldsymbol{x}}}_{t_n+\tau'})\right\|^2\right] d\tau'$$

$$= \sum_{m=0}^{M_n} \int_{\tau_{n,m}}^{\tau_{n,m+1}} \mathbb{E}\left[\left\|\nabla \log \breve{p}_{t_n+\tau_{n,m}}(\bar{\bar{\boldsymbol{x}}}_{t_n+\tau_{n,m}}) - \nabla \log \breve{p}_{t_n+\tau'}(\bar{\bar{\boldsymbol{x}}}_{t_n+\tau'})\right\|^2\right] d\tau'$$

$$\lesssim \sum_{m=0}^{M_n} d\epsilon_{n,m}^2 \epsilon_{n,m} \leq d\epsilon^2 h_n.$$

For $n = N-1$, notice that by the step size schedule (*cf.* Section 3.1.1) and suppose $\epsilon \leq 1/2$, we have

$$\frac{T-\tau}{2} \leq T - g_n(\tau) \leq T - \tau,$$

and then again [111, Corollary 1] states

$$\mathbb{E}\left[\left\|\nabla \log \breve{p}_{t_n+\epsilon_{n,m}}(\bar{\bar{\boldsymbol{x}}}_{t_n+\epsilon_{n,m}}) - \nabla \log \breve{p}_{t_n+\tau'}(\bar{\bar{\boldsymbol{x}}}_{t_n+\tau'})\right\|^2\right] \lesssim \frac{d\epsilon_{n,m}^2}{T - \tau_{n,m}},$$

and thus

$$\mathbb{E}\left[D_{t_{N-1}}\right] = \int_0^{h_{N-1}} \mathbb{E}\left[\left\|\nabla \log \breve{p}_{t_{N-1}+g_n(\tau')}(\bar{\bar{\boldsymbol{x}}}_{t_{N-1}+g_n(\tau')}) - \nabla \log \breve{p}_{t_{N-1}+\tau'}(\bar{\bar{\boldsymbol{x}}}_{t_{N-1}+\tau'})\right\|^2\right] d\tau$$

$$= \sum_{m=0}^{M_{N-1}} \int_{\tau_{n,m}}^{\tau_{n,m+1}} \mathbb{E}\left[\left\|\nabla \log \breve{p}_{t_{N-1}+\tau_{n,m}}(\bar{\bar{\boldsymbol{x}}}_{t_{N-1}+g_n(\tau')}) - \nabla \log \breve{p}_{t_{N-1}+\tau'}(\bar{\bar{\boldsymbol{x}}}_{t_{N-1}+\tau'})\right\|^2\right] d\tau'$$

$$\lesssim \sum_{m=0}^{M_{N-1}} \frac{d\epsilon_{n,m}^2}{T - \tau_{n,m}} \epsilon_{n,m} \leq \sum_{m=0}^{M_{N-1}} d\epsilon_{n,m}^2 \epsilon \lesssim \int_{\delta_\infty}^{T-t_{N-1}} d\tau d\tau \lesssim d\epsilon^2 h_{N-1}.$$

$\qquad\square$

**Remark C.6.** *The above lemma is able to achieve a better dependency on $\epsilon$ compared to Lemma B.7, because the backward process $(\breve{\bar{x}}_t)_{t\in[0,T]}$ is now a deterministic process in the probability flow ODE formulation, instead of a stochastic process as in the SDE formulation as in Lemma B.7. Thus, intuitively applying Cauchy-Schwarz rather than Itô symmetry gives us a $\mathcal{O}(\epsilon^2)$-dependency rather than $\mathcal{O}(\epsilon)$-dependency.*

**Theorem C.7.** *Under Assumptions 3.1', 3.2, 3.3, and 3.4, then the distribution $\widetilde{q}_{t_n,h_n}$ that the parallelized predictor step generates samples from satisfies the following error bound:*

$$W_2(\widetilde{q}_{t_n,h_n},\breve{p}_{t_{n+1}})^2 \lesssim de^{-K} + h_n^2\delta_\infty^2 + d\epsilon^2 h_n^2,$$

*for $n \in [0:N-1]$.*

*Proof.* By the definition of 2-Wasserstein distance, we have for any coupling of $\widetilde{y}_{t_n,h_n}^{(K)}$ and $\breve{\bar{x}}_{t_n+h_n}$,

$$W_2(\widetilde{q}_{t_n,h_n},\breve{p}_{t_{n+1}})^2 \leq \mathbb{E}\left[\left\|\widetilde{y}_{t_n,h_n}^{(K)} - \breve{\bar{x}}_{t_n+h_n}\right\|^2\right],$$

and therefore

$$W_2(\widetilde{q}_{t_n,h_n},\breve{p}_{t_{n+1}})^2 \leq \mathbb{E}\left[\left\|\widetilde{y}_{t_n,h_n}^{(K)} - \breve{\bar{x}}_{t_n+h_n}\right\|^2\right] \leq \sup_{\tau\in[0,h_n]}\mathbb{E}\left[\left\|\widetilde{y}_{t_n,\tau}^{(K)} - \breve{\bar{x}}_{t_n+\tau}\right\|^2\right]$$

$$\leq 3d\left(\frac{h_n^2 e^{h_n+\frac{3}{2}}L_{\boldsymbol{s}}^2}{2}\right)^K + \frac{e^{h_n+\frac{3}{2}}h_n/2}{1-h_n^2 e^{h_n+\frac{3}{2}}L_{\boldsymbol{s}}^2/2}\left(h_n\delta_\infty^2 + \mathbb{E}[D_{t_n}]\right)$$

$$\lesssim de^{-K} + h_n^2\delta_\infty^2 + d\epsilon^2 h_n^2,$$

where for the second to last inequality we used Lemma C.4, the last inequality is due to Lemma C.5 and the assumption $h_n^2 e^{h_n}L_{\boldsymbol{s}}^2 \ll 1$. $\qquad\square$

## C.3 Parallelized Corrector Step

After each predictor step, we run the corrector step for $\mathcal{O}(1)$ time to reduce the error. Particularly, we apply the Parallelized underdamped Langevin dynamics algorithm [130] to the corrector step, which yields $\mathcal{O}(1)$ approximate time complexity compared to the ordinary implementation of the ULMC dynamics as in [111]. In the following, we will drop the dependency on $\omega$ for notational simplicity, and we refer readers to Appendix A.2 and B.2 to review the change of measure arguments and the application of Girsanov's theorem A.4. We will also use a general notation $*^\dagger$ to distinguish the time in the backward process and the inner time in the corrector step of the $n$-th block.

We first define the true underdamped Langevin dynamics $(\boldsymbol{u}_{t_n,t^\dagger},\boldsymbol{v}_{t_n,t^\dagger})_{t\geq 0}$:

$$\begin{cases}\mathrm{d}\boldsymbol{u}_{t_n,t^\dagger} = \boldsymbol{v}_{t_n,t^\dagger}\mathrm{d}t^\dagger \\ \mathrm{d}\boldsymbol{v}_{t_n,t^\dagger} = -\gamma\boldsymbol{v}_{t_n,t^\dagger}\mathrm{d}t^\dagger - \nabla\log\breve{p}_{t_{n+1}}(\boldsymbol{u}_{t_n,t^\dagger})\mathrm{d}t^\dagger + \sqrt{2\gamma}\mathrm{d}\boldsymbol{b}_{t_n,t^\dagger},\end{cases} \tag{C.10}$$

with initial condition $\boldsymbol{u}_{t_n,0} \equiv \widetilde{\boldsymbol{y}}_{t_n,h_n}^{(K^\dagger)}$ from the predictor step and $\boldsymbol{v}_{t_n,0} \sim \mathcal{N}(0,\boldsymbol{I}_d)$, where $(\boldsymbol{b}_{t_n,t^\dagger})_{t\geq 0}$ is a Wiener process. We may also write the system of SDEs above in the following matrix form:

$$\mathrm{d}\begin{bmatrix}\boldsymbol{u}_{t_n,t^\dagger} \\ \boldsymbol{v}_{t_n,t^\dagger}\end{bmatrix} = \left[\begin{bmatrix}\boldsymbol{0} & \boldsymbol{I}_d \\ \boldsymbol{0} & -\gamma\boldsymbol{I}_d\end{bmatrix}\begin{bmatrix}\boldsymbol{u}_{t_n,t^\dagger} \\ \boldsymbol{v}_{t_n,t^\dagger}\end{bmatrix} - \begin{bmatrix}\boldsymbol{0} \\ \nabla\log\breve{p}_{t_{n+1}}(\boldsymbol{u}_{t_n,t^\dagger})\end{bmatrix}\right]\mathrm{d}t^\dagger + \begin{bmatrix}\boldsymbol{0} & \boldsymbol{0} \\ \boldsymbol{0} & \sqrt{2\gamma}\boldsymbol{I}_d\end{bmatrix}\mathrm{d}\begin{bmatrix}\boldsymbol{b}'_{t_n,t^\dagger} \\ \boldsymbol{b}_{t_n,t^\dagger}\end{bmatrix}.$$

We run this underdamped Langevin dynamics until the pre-determined time horizon $T^\dagger$. We also define the joint probability distribution of $(\boldsymbol{u}_{t_n,t^\dagger},\boldsymbol{v}_{t_n,t^\dagger})$ at time $t$ as $\pi_{t_n,t^\dagger}(\boldsymbol{u}_{t_n,t^\dagger},\boldsymbol{v}_{t_n,t^\dagger})$ and its marginal on $\boldsymbol{u}_{t_n,t^\dagger}$ as $\pi_{t_n,t^\dagger}^{\boldsymbol{u}}(\boldsymbol{u}_{t_n,t^\dagger})$.

Similar to the parallelizing strategy in Section 3.1.1, we discretize the time interval $[0,T^\dagger]$ into $N^\dagger$ blocks with length $h^\dagger = T^\dagger/N^\dagger$. Within the $n$-th block, we further divide the block $[n^\dagger h^\dagger, (n+1)h^\dagger]$ into $M^\dagger$ steps, each with step size $\epsilon^\dagger = h^\dagger/M^\dagger$.

**Definition C.8** (Auxiliary corrector process). *For any $n^\dagger \in [0 : N^\dagger - 1]$, we define the auxiliary corrector process $(\widehat{\boldsymbol{u}}^{(k^\dagger)}_{t_n, n^\dagger h^\dagger, \tau^\dagger})_{\tau^\dagger \in [0, h^\dagger]}$ as the solution to the following SDE recursively for $k^\dagger \in [0 : K^\dagger - 1]$:*

$$
\begin{cases}
\mathrm{d}\widehat{\boldsymbol{u}}^{(k+1)}_{t_n, n^\dagger h^\dagger, \tau^\dagger} = \widehat{\boldsymbol{v}}^{(k+1)}_{t_n, n^\dagger h^\dagger, \tau^\dagger} \mathrm{d}\tau^\dagger, \\
\mathrm{d}\widehat{\boldsymbol{v}}^{(k+1)}_{t_n, n^\dagger h^\dagger, \tau^\dagger} = -\gamma \widehat{\boldsymbol{v}}^{(k+1)}_{t_n, n^\dagger h^\dagger, \tau^\dagger} \mathrm{d}\tau^\dagger - \boldsymbol{s}_{t_{n+1}}\!\left(\widehat{\boldsymbol{u}}^{(k^\dagger)}_{t_n, n^\dagger h^\dagger, g_n(\tau^\dagger)}\right)\mathrm{d}\tau^\dagger + \sqrt{2\gamma}\mathrm{d}\boldsymbol{b}_{t_n, n^\dagger h^\dagger + \tau^\dagger}
\end{cases}
\tag{C.11}
$$

*with the initial condition*

$$
\begin{cases}
\widehat{\boldsymbol{u}}^{(0)}_{t_n, n^\dagger h^\dagger, \tau^\dagger} \equiv \widehat{\boldsymbol{u}}_{t_n, n^\dagger h^\dagger} \\
\widehat{\boldsymbol{v}}^{(0)}_{t_n, n^\dagger h^\dagger, \tau^\dagger} = \widehat{\boldsymbol{v}}_{t_n, n^\dagger h^\dagger}
\end{cases}
\text{for } \tau^\dagger \in [0, h^\dagger], \text{ and }
\begin{cases}
\widehat{\boldsymbol{u}}^{(k^\dagger)}_{t_n, n^\dagger h^\dagger, \tau^\dagger} \equiv \widehat{\boldsymbol{u}}_{t_n, n^\dagger h^\dagger} \\
\widehat{\boldsymbol{v}}^{(k^\dagger)}_{t_n, n^\dagger h^\dagger, 0} \equiv \widehat{\boldsymbol{v}}_{t_n, n^\dagger h^\dagger}
\end{cases}
\text{for } k \in [1 : K^\dagger],
\tag{C.12}
$$

*where*

$$
\widehat{\boldsymbol{u}}_{t_n, n^\dagger h^\dagger} := \widehat{\boldsymbol{u}}^{(K^\dagger)}_{t_n, (n^\dagger - 1)h^\dagger, h^\dagger}, \quad \widehat{\boldsymbol{v}}_{t_n, n^\dagger h^\dagger} := \widehat{\boldsymbol{v}}^{(K^\dagger)}_{t_n, (n^\dagger - 1)h^\dagger, h^\dagger}
$$

*for $n^\dagger \in [1 : N^\dagger - 1]$, and*

$$
\widehat{\boldsymbol{u}}_{t_n, 0} = \boldsymbol{y}^{(K)}_{t_n, h_n}, \quad \widehat{\boldsymbol{v}}_{t_n, 0} \sim \mathcal{N}(0, \boldsymbol{I}_d).
$$

*We define the joint probability distribution of $(\widehat{\boldsymbol{u}}_{t_n, t^\dagger}, \widehat{\boldsymbol{v}}_{t_n, t^\dagger})$ at time $t$ as $\widehat{\pi}_{t_n, t^\dagger}(\widehat{\boldsymbol{u}}_{t_n, t^\dagger}, \widehat{\boldsymbol{v}}_{t_n, t^\dagger})$ and its marginal on $\widehat{\boldsymbol{u}}_{t_n, t^\dagger}$ as $\widehat{\pi}^{\widehat{\boldsymbol{u}}}_{t_n, t^\dagger}(\widehat{\boldsymbol{u}}_{t_n, t^\dagger})$. We will also denote the resulting probability distribution of $\widehat{\pi}^{\widehat{\boldsymbol{u}}}_{t_n, T^\dagger}$ as $\widehat{q}_{t_{n+1}}$.*

**Lemma C.9** (Equivalence between (C.2) and (C.11)). *For any $n^\dagger \in [0 : N^\dagger - 1]$, the update rule in Algorithm 2 is equivalent to the exact solution of the auxiliary process (C.11) for any $k^\dagger \in [0 : K^\dagger - 1]$ and $\tau^\dagger \in [0, h^\dagger]$.*

*Proof.* Without loss of generality, we will prove the lemma for $m^\dagger = M^\dagger$. The proof for $m^\dagger \in [0 : M^\dagger - 1]$ can be done similarly.

We first rewrite (C.2) into the matrix form:

$$
\mathrm{d}\begin{bmatrix} \widetilde{\boldsymbol{u}}^{(k^\dagger)}_{t_n, n^\dagger h^\dagger, \tau^\dagger} \\ \widetilde{\boldsymbol{v}}^{(k^\dagger)}_{t_n, n^\dagger h^\dagger, \tau^\dagger} \end{bmatrix} = \left(\begin{bmatrix} \boldsymbol{0} & \boldsymbol{I}_d \\ \boldsymbol{0} & -\gamma\boldsymbol{I}_d \end{bmatrix} \begin{bmatrix} \widetilde{\boldsymbol{u}}^{(k^\dagger)}_{t_n, n^\dagger h^\dagger, \tau^\dagger} \\ \widetilde{\boldsymbol{v}}^{(k^\dagger)}_{t_n, n^\dagger h^\dagger, \tau^\dagger} \end{bmatrix} - \begin{bmatrix} \boldsymbol{0} \\ \boldsymbol{s}_{t_{n+1}}\!\left(\widetilde{\boldsymbol{u}}^{(k^\dagger)}_{t_n, n^\dagger h^\dagger, g_n(\tau^\dagger)}\right) \end{bmatrix}\right)\mathrm{d}\tau^\dagger
$$

$$
+ \begin{bmatrix} \boldsymbol{0} & \boldsymbol{0} \\ \boldsymbol{0} & \sqrt{2\gamma}\boldsymbol{I}_d \end{bmatrix} \mathrm{d}\begin{bmatrix} \boldsymbol{b}'_{t_n, n^\dagger h^\dagger + \tau^\dagger} \\ \boldsymbol{b}_{t_n, n^\dagger h^\dagger + \tau^\dagger} \end{bmatrix}.
\tag{C.13}
$$

Define the time-dependent matrix $\boldsymbol{G}(\cdot)$ as

$$
\boldsymbol{G}(t^\dagger) := \begin{bmatrix} \boldsymbol{I}_d & \frac{1 - e^{-\gamma t^\dagger}}{\gamma}\boldsymbol{I}_d \\ \boldsymbol{0} & e^{-\gamma t^\dagger}\boldsymbol{I}_d \end{bmatrix} = \exp\left(\begin{pmatrix} \boldsymbol{0} & \boldsymbol{I}_d \\ \boldsymbol{0} & -\gamma\boldsymbol{I}_d \end{pmatrix} t^\dagger\right),
\tag{C.14}
$$

satisfying that

$$
\frac{\mathrm{d}}{\mathrm{d}t^\dagger}\boldsymbol{G}(t^\dagger) = \begin{bmatrix} \boldsymbol{0} & \boldsymbol{I}_d \\ \boldsymbol{0} & -\gamma\boldsymbol{I}_d \end{bmatrix} \boldsymbol{G}(t^\dagger) = \boldsymbol{G}(t^\dagger)\begin{bmatrix} \boldsymbol{0} & \boldsymbol{I}_d \\ \boldsymbol{0} & -\gamma\boldsymbol{I}_d \end{bmatrix}.
$$

Now we multiply $\boldsymbol{G}(-\tau^\dagger)$ on both sides of (C.13) to obtain:

$$
\mathrm{d}\left(\boldsymbol{G}(-\tau^\dagger)\begin{bmatrix} \widetilde{\boldsymbol{u}}^{(k^\dagger)}_{t_n, n^\dagger h^\dagger, \tau^\dagger} \\ \widetilde{\boldsymbol{v}}^{(k^\dagger)}_{t_n, n^\dagger h^\dagger, \tau^\dagger} \end{bmatrix}\right) = -\boldsymbol{G}(-\tau^\dagger)\begin{bmatrix} \boldsymbol{0} \\ \boldsymbol{s}_{t_{n+1}}\!\left(\widetilde{\boldsymbol{u}}^{(k^\dagger)}_{t_n, n^\dagger h^\dagger, g_n(\tau^\dagger)}\right) \end{bmatrix}\mathrm{d}\tau^\dagger
$$

$$
+ \boldsymbol{G}(-\tau^\dagger)\begin{bmatrix} \boldsymbol{0} & \boldsymbol{0} \\ \boldsymbol{0} & \sqrt{2\gamma}\boldsymbol{I}_d \end{bmatrix} \mathrm{d}\begin{bmatrix} \boldsymbol{b}'_{t_n, n^\dagger h^\dagger + \tau^\dagger} \\ \boldsymbol{b}_{t_n, n^\dagger h^\dagger + \tau^\dagger} \end{bmatrix}.
$$

Integrating on both sides from 0 to $h^\dagger$ and multiplying $\boldsymbol{G}(h^\dagger)$ on both sides, we have

$$
\begin{bmatrix} \widetilde{\boldsymbol{u}}^{(k^\dagger)}_{t_n,n^\dagger h^\dagger,\tau} \\ \widetilde{\boldsymbol{v}}^{(k^\dagger)}_{t_n,n^\dagger h^\dagger,\tau} \end{bmatrix} - \boldsymbol{G}(h^\dagger) \begin{bmatrix} \widetilde{\boldsymbol{u}}^{(k^\dagger)}_{t_n,n^\dagger h^\dagger,0} \\ \widetilde{\boldsymbol{v}}^{(k^\dagger)}_{t_n,n^\dagger h^\dagger,0} \end{bmatrix}
$$

$$
= -\int_0^{h^\dagger} \boldsymbol{G}(h^\dagger - \tau^{\dagger\prime}) \begin{bmatrix} \boldsymbol{0} \\ \boldsymbol{s}_{t_{n+1}}\big(\widetilde{\boldsymbol{u}}^{(k^\dagger)}_{t_n,n^\dagger h^\dagger,g(\tau^{\dagger\prime})}\big) \end{bmatrix} \mathrm{d}\tau^{\dagger\prime}
$$

$$
+ \int_0^{h^\dagger} \boldsymbol{G}(h^\dagger - \tau^{\dagger\prime}) \begin{bmatrix} \boldsymbol{0} & \boldsymbol{0} \\ \boldsymbol{0} & \sqrt{2\gamma}\boldsymbol{I}_d \end{bmatrix} \mathrm{d} \begin{bmatrix} \boldsymbol{b}'_{t_n,n^\dagger h^\dagger + \tau^{\dagger\prime}} \\ \boldsymbol{b}_{t_n,n^\dagger h^\dagger + \tau^{\dagger\prime}} \end{bmatrix}
$$

$$
= -\sum_{m^\dagger=0}^{M^\dagger-1} \int_{m^\dagger\epsilon^\dagger}^{(m^\dagger+1)\epsilon^\dagger} \boldsymbol{G}(h^\dagger - \tau^{\dagger\prime})\mathrm{d}\tau^{\dagger\prime} \begin{bmatrix} \boldsymbol{0} \\ \boldsymbol{s}_{t_{n+1}}\big(\widetilde{\boldsymbol{u}}^{(k^\dagger)}_{t_n,n^\dagger h^\dagger,m^\dagger\epsilon^\dagger}\big) \end{bmatrix}
$$

$$
+ \sum_{m^\dagger=0}^{M^\dagger-1} \int_{m^\dagger\epsilon^\dagger}^{(m^\dagger+1)\epsilon^\dagger} \boldsymbol{G}(h^\dagger - \tau^{\dagger\prime}) \begin{bmatrix} \boldsymbol{0} & \boldsymbol{0} \\ \boldsymbol{0} & \sqrt{2\gamma}\boldsymbol{I}_d \end{bmatrix} \mathrm{d} \begin{bmatrix} \boldsymbol{b}'_{t_n,n^\dagger h^\dagger + \tau^{\dagger\prime}} \\ \boldsymbol{b}_{t_n,n^\dagger h^\dagger + \tau^{\dagger\prime}} \end{bmatrix}
$$

$$
= -\sum_{m^\dagger=0}^{M^\dagger-1} \big(\boldsymbol{G}(\epsilon^\dagger) - \boldsymbol{I}_d\big)\boldsymbol{G}((M^\dagger - m^\dagger - 1)\epsilon^\dagger) \begin{bmatrix} \boldsymbol{0} \\ \boldsymbol{s}_{t_{n+1}}\big(\widetilde{\boldsymbol{u}}^{(k^\dagger)}_{t_n,n^\dagger h^\dagger,m^\dagger\epsilon^\dagger}\big) \end{bmatrix}
$$

$$
+ \sum_{m^\dagger=0}^{M^\dagger-1} \int_{m^\dagger\epsilon^\dagger}^{(m^\dagger+1)\epsilon^\dagger} \boldsymbol{G}(h^\dagger - \tau^{\dagger\prime}) \begin{bmatrix} \boldsymbol{0} & \boldsymbol{0} \\ \boldsymbol{0} & \sqrt{2\gamma}\boldsymbol{I}_d \end{bmatrix} \mathrm{d} \begin{bmatrix} \boldsymbol{b}'_{t_n,n^\dagger h^\dagger + \tau^{\dagger\prime}} \\ \boldsymbol{b}_{t_n,n^\dagger h^\dagger + \tau^{\dagger\prime}} \end{bmatrix}.
$$

By Itô isometry, we have

$$
\int_{m^\dagger\epsilon^\dagger}^{(m^\dagger+1)\epsilon^\dagger} \boldsymbol{G}(h^\dagger - \tau^{\dagger\prime}) \begin{bmatrix} \boldsymbol{0} & \boldsymbol{0} \\ \boldsymbol{0} & \sqrt{2\gamma}\boldsymbol{I}_d \end{bmatrix} \mathrm{d} \begin{bmatrix} \boldsymbol{b}'_{t_n,n^\dagger h^\dagger + \tau^{\dagger\prime}} \\ \boldsymbol{b}_{t_n,n^\dagger h^\dagger + \tau^{\dagger\prime}} \end{bmatrix}
$$

$$
\sim \mathcal{N}\Bigg( \boldsymbol{0}, \begin{bmatrix} \boldsymbol{0} & \boldsymbol{0} \\ \boldsymbol{0} & \sqrt{2\gamma\boldsymbol{I}_d} \end{bmatrix} \boldsymbol{G}((M^\dagger - m^\dagger - 1)\epsilon^\dagger)^\top \big(\boldsymbol{G}(\epsilon^\dagger) - \boldsymbol{I}_d\big)^\top
$$

$$
\big(\boldsymbol{G}(\epsilon^\dagger) - \boldsymbol{I}_d\big)\boldsymbol{G}((M^\dagger - m^\dagger - 1)\epsilon^\dagger) \begin{bmatrix} \boldsymbol{0} & \boldsymbol{0} \\ \boldsymbol{0} & \sqrt{2\gamma\boldsymbol{I}_d} \end{bmatrix} \Bigg)
$$

$$
\sim \begin{bmatrix} \boldsymbol{0} \\ \mathcal{N}\big(\boldsymbol{0}, 2\gamma(1+\gamma^{-2})(1 - e^{-\gamma\epsilon^\dagger})^2 e^{-2\gamma(M^\dagger - m^\dagger + 1)\epsilon^\dagger)}\boldsymbol{I}_d\big) \end{bmatrix},
$$

as desired $\qquad\qquad\square$

**Definition C.10** (Interpolating corrector process). *For any $n^\dagger \in [0 : N^\dagger - 1]$, we define the interpo-lating corrector process $(\widehat{\boldsymbol{u}}^{(k^\dagger)}_{t_n,n^\dagger h^\dagger,\tau^\dagger})_{\tau^\dagger\in[0,h^\dagger]}$ as the solution to the following SDE recursively for $k^\dagger \in [0 : K^\dagger - 1]$:*

$$
\begin{cases} \mathrm{d}\widetilde{\boldsymbol{u}}^{(k+1)}_{t_n,n^\dagger h^\dagger,\tau^\dagger} = \widetilde{\boldsymbol{v}}^{(k+1)}_{t_n,n^\dagger h^\dagger,\tau^\dagger}\mathrm{d}\tau^\dagger, \\ \mathrm{d}\widetilde{\boldsymbol{v}}^{(k+1)}_{t_n,n^\dagger h^\dagger,\tau^\dagger} = -\gamma\widetilde{\boldsymbol{v}}^{(k+1)}_{t_n,n^\dagger h^\dagger,\tau^\dagger}\mathrm{d}\tau^\dagger - \boldsymbol{s}_{t_{n+1}}\big(\widetilde{\boldsymbol{u}}^{(k^\dagger)}_{t_n,n^\dagger h^\dagger,g_n(\tau^\dagger)}\big)\mathrm{d}\tau^\dagger + \sqrt{2\gamma}\mathrm{d}\boldsymbol{b}_{t_n,n^\dagger h^\dagger + \tau^\dagger} \end{cases} \tag{C.15}
$$

*with the initial condition*

$$
\begin{cases} \widetilde{\boldsymbol{u}}^{(0)}_{t_n,n^\dagger h^\dagger,\tau^\dagger} \equiv \widetilde{\boldsymbol{u}}_{t_n,n^\dagger h^\dagger} \\ \widetilde{\boldsymbol{v}}^{(0)}_{t_n,n^\dagger h^\dagger,\tau^\dagger} = \widetilde{\boldsymbol{v}}_{t_n,n^\dagger h^\dagger} \end{cases} \text{for } \tau^\dagger \in [0, h^\dagger], \text{ and } \begin{cases} \widetilde{\boldsymbol{u}}^{(k^\dagger)}_{t_n,n^\dagger h^\dagger,\tau^\dagger} \equiv \widetilde{\boldsymbol{u}}_{t_n,n^\dagger h^\dagger} \\ \widetilde{\boldsymbol{v}}^{(k^\dagger)}_{t_n,n^\dagger h^\dagger,0} \equiv \widetilde{\boldsymbol{v}}_{t_n,n^\dagger h^\dagger} \end{cases} \text{for } k \in [1 : K^\dagger],
$$

$$
\tag{C.16}
$$

*where*

$$
\widetilde{\boldsymbol{u}}_{t_n,n^\dagger h^\dagger} := \widetilde{\boldsymbol{u}}^{(K^\dagger)}_{(n^\dagger-1)h^\dagger,h^\dagger}, \quad \widetilde{\boldsymbol{v}}_{t_n,n^\dagger h^\dagger} := \widetilde{\boldsymbol{v}}^{(K^\dagger)}_{(n^\dagger-1)h^\dagger,h^\dagger}
$$

*for $n^\dagger \in [1 : N^\dagger - 1]$, and*

$$
\widetilde{\boldsymbol{u}}_{t_n,0} = \widetilde{\boldsymbol{y}}^{(K)}_{t_n,h_n}, \quad \widetilde{\boldsymbol{v}}_{t_n,0} \sim \mathcal{N}(0, \boldsymbol{I}_d).
$$

*We define the joint probability distribution of $(\widetilde{\boldsymbol{u}}_{t_n,t^\dagger}, \widetilde{\boldsymbol{v}}_{t_n,t^\dagger})$ at time $t$ as $\widetilde{\pi}_{t_n,t^\dagger}(\widetilde{\boldsymbol{u}}_{t_n,t^\dagger}, \widetilde{\boldsymbol{v}}_{t_n,t^\dagger})$ and its marginal on $\widetilde{\boldsymbol{u}}_{t_n,t^\dagger}$ as $\widetilde{\pi}^{\widetilde{\boldsymbol{u}}}_{t_n,t^\dagger}(\widetilde{\boldsymbol{u}}_{t_n,t^\dagger})$.*

We invoke Girsanov's theorem (Theorem A.4) again by the following procedure

1. Setting (A.2) as the auxiliary process (C.15) at iteration $K^\dagger$, where $\boldsymbol{b}_{t_n,t^\dagger}(\omega)$ is a Wiener process under the measure $Q$;

2. Defining another process $\widetilde{\boldsymbol{b}}_{t_n,n^\dagger h^\dagger + \tau^\dagger}$ governed by the following SDE:

$$\mathrm{d}\widetilde{\boldsymbol{b}}_{t_n,n^\dagger h^\dagger + \tau^\dagger} = \mathrm{d}\boldsymbol{b}_{t_n,n^\dagger h^\dagger + \tau^\dagger} - \boldsymbol{\phi}_{t_n,n^\dagger h^\dagger}(\tau^\dagger)\mathrm{d}\tau^\dagger, \tag{C.17}$$

where

$$\boldsymbol{\phi}_{t_n,n^\dagger h^\dagger}(\tau^\dagger) = \frac{1}{\sqrt{2\gamma}}\left(\boldsymbol{s}_{t_{n+1}}(\widetilde{\boldsymbol{u}}^{(K^\dagger - 1)}_{t_n,n^\dagger h^\dagger, \lfloor \frac{\tau^\dagger}{\epsilon^\dagger} \rfloor \epsilon^\dagger}) - \nabla\log\breve{p}_{t_{n+1}}(\widetilde{\boldsymbol{u}}^{(K^\dagger)}_{t_n,n^\dagger h^\dagger, \tau^\dagger})\right) \tag{C.18}$$

and computing the Radon-Nikodym derivative of the measure $P$ with respect to $Q$ as

$$\frac{\mathrm{d}P}{\mathrm{d}Q} = \exp\left(\int_0^{h^\dagger} \boldsymbol{\phi}_{t_n,n^\dagger h^\dagger}(\tau^\dagger)^\top \mathrm{d}\boldsymbol{b}_{t_n,n^\dagger h^\dagger + \tau^\dagger} - \frac{1}{2}\int_0^{h^\dagger} \|\boldsymbol{\phi}_{nh}(\tau^\dagger)\|^2 \mathrm{d}\tau^\dagger\right); \tag{C.19}$$

3. Concluding that (C.15) at iteration $K^\dagger$ under the measure $Q$ satisfies the following SDE:

$$\begin{cases} \mathrm{d}\widetilde{\boldsymbol{u}}^{(K^\dagger)}_{n^\dagger h^\dagger, \tau^\dagger} = \widetilde{\boldsymbol{v}}^{(K^\dagger)}_{n^\dagger h^\dagger, \tau^\dagger}\mathrm{d}\tau^\dagger \\ \mathrm{d}\widetilde{\boldsymbol{v}}^{(K^\dagger)}_{t_n,n^\dagger h^\dagger, \tau^\dagger} = -\gamma\widetilde{\boldsymbol{v}}^{(K^\dagger)}_{t_n,n^\dagger h^\dagger, \tau^\dagger}\mathrm{d}\tau^\dagger - \nabla\log\breve{p}_{t_{n+1}}(\widetilde{\boldsymbol{u}}^{(K^\dagger)}_{n^\dagger h^\dagger, \tau^\dagger})\mathrm{d}\tau^\dagger + \sqrt{2\gamma}\mathrm{d}\widetilde{\boldsymbol{b}}_{t_n,n^\dagger h^\dagger + \tau^\dagger}, \end{cases} \tag{C.20}$$

with $(\widetilde{\boldsymbol{b}}_{t_n,n^\dagger h^\dagger + \tau^\dagger})_{\tau^\dagger \geq 0}$ being a Wiener process under the measure $P$. If we replace $(\widetilde{\boldsymbol{u}}^{(K^\dagger)}_{n^\dagger h^\dagger, \tau^\dagger}, \widetilde{\boldsymbol{v}}^{(K^\dagger)}_{n^\dagger h^\dagger, \tau^\dagger})$ by $(\boldsymbol{u}_{t_n,n^\dagger h^\dagger + \tau^\dagger}, \boldsymbol{v}_{t_n,n^\dagger h^\dagger + \tau^\dagger})$, one should notice (C.20) is immediately the original backward SDE (C.10) with the true score function on $t \in [n^\dagger h^\dagger, (n+1)h^\dagger]$:

$$\begin{cases} \mathrm{d}\boldsymbol{u}_{t_n,n^\dagger h^\dagger + \tau^\dagger} = \boldsymbol{v}_{t_n,n^\dagger h^\dagger + \tau^\dagger}\mathrm{d}\tau^\dagger \\ \mathrm{d}\boldsymbol{v}_{t_n,n^\dagger h^\dagger + \tau^\dagger} = -\gamma\boldsymbol{v}_{t_n,n^\dagger h^\dagger + \tau^\dagger}\mathrm{d}\tau^\dagger - \nabla\log\breve{p}_{t_{n+1}}(\boldsymbol{u}_{t_n,n^\dagger h^\dagger + \tau^\dagger})\mathrm{d}\tau^\dagger + \sqrt{2\gamma}\mathrm{d}\widetilde{\boldsymbol{b}}_{t_n,n^\dagger h^\dagger + \tau^\dagger}. \end{cases} \tag{C.21}$$

We further define the joint probability distribution of $(\boldsymbol{u}_{t_n,t^\dagger}, \boldsymbol{v}_{t_n,t^\dagger})$ at time $t$ as $\pi_{t_n,t^\dagger}(\boldsymbol{u}_{t_n,t^\dagger}, \boldsymbol{v}_{t_n,t^\dagger})$ and its marginal on $\boldsymbol{u}_{t_n,t^\dagger}$ as $\pi^{\boldsymbol{u}}_{t_n,t^\dagger}(\boldsymbol{u}_{t_n,t^\dagger})$.

**Remark C.11.** *The application of Girsanov's theorem A.4 is by writing the system of SDEs in the matrix form.*

**Definition C.12** (Stationary process). *Under the $P$-measure that is defined by the Radon-Nikodym derivative (C.19), we may define a stationary underdamped Langevin process for $n^\dagger \in [0 : N^\dagger - 1]$ and $\tau^\dagger \in [0, h^\dagger]$ as*

$$\begin{cases} \mathrm{d}\boldsymbol{u}^*_{t_n,n^\dagger h^\dagger + \tau^\dagger} = \boldsymbol{v}^*_{t_n,n^\dagger h^\dagger + \tau^\dagger}\mathrm{d}\tau^\dagger, \\ \mathrm{d}\boldsymbol{v}^*_{t_n,n^\dagger h^\dagger + \tau^\dagger} = -\gamma\boldsymbol{v}^*_{t_n,n^\dagger h^\dagger + \tau^\dagger}\mathrm{d}\tau^\dagger - \nabla\log\breve{p}_{t_{n+1}}(\boldsymbol{u}^*_{n^\dagger h^\dagger + \tau^\dagger})\mathrm{d}\tau^\dagger + \sqrt{2\gamma}\mathrm{d}\widetilde{\boldsymbol{b}}_{t_n,n^\dagger h^\dagger + \tau^\dagger}, \end{cases} \tag{C.22}$$

*with the initial condition $\boldsymbol{u}^*_{t_n,n^\dagger h^\dagger} \sim \breve{p}_{t_{n+1}}$ and $\boldsymbol{v}^*_{t_n,n^\dagger h^\dagger} \sim \mathcal{N}(0, \boldsymbol{I}_d)$. We define the joint probability distribution of $(\boldsymbol{u}^*_{t_n,t^\dagger}, \boldsymbol{v}^*_{t_n,t^\dagger})$ at time $t$ as $\pi^*_{t_n,t^\dagger}(\boldsymbol{u}^*_{t_n,t^\dagger}, \boldsymbol{v}^*_{t_n,t^\dagger})$ and its marginal on $\boldsymbol{u}^*_{t_n,t^\dagger}$ as $\pi^{*,\boldsymbol{u}^*}_{t_n,t^\dagger}(\boldsymbol{u}^*_{t_n,t^\dagger})$.*

Thus, from Corollary A.5, we have that

$$D_{\mathrm{KL}}(\pi_{t_n,n^\dagger h^\dagger}\|\widetilde{\pi}_{t_n,n^\dagger h^\dagger})$$

$$\leq D_{\mathrm{KL}}(\pi_{t_n,(n-1)h^\dagger}\|\widetilde{\pi}_{t_n,(n-1)h^\dagger}) + \sum_{n=0}^{N^\dagger - 1} D_{\mathrm{KL}}(\pi_{t_n,n^\dagger h^\dagger:(n+1)h^\dagger}\|\widetilde{\pi}_{t_n,n^\dagger h^\dagger:(n+1)h^\dagger})$$

$$\leq D_{\mathrm{KL}}(\pi_{t_n,(n-1)h^\dagger}\|\widetilde{\pi}_{t_n,(n-1)h^\dagger})$$

$$+ \frac{1}{4\gamma}\mathbb{E}_P\left[\int_0^{h^\dagger} \left\|\boldsymbol{s}_{t_{n+1}}(\widetilde{\boldsymbol{u}}^{(K^\dagger - 1)}_{t_n,n^\dagger h^\dagger, \lfloor \frac{\tau^\dagger}{\epsilon^\dagger} \rfloor \epsilon^\dagger}) - \nabla\log\breve{p}_{t_{n+1}}(\widetilde{\boldsymbol{u}}^{(K^\dagger)}_{t_n,n^\dagger h^\dagger, \tau^\dagger})\right\|^2 \mathrm{d}\tau^\dagger\right]. \tag{C.23}$$

By triangle inequality, we have

$$\int_0^{h^\dagger} \left\| s_{t_{n+1}}(\widetilde{\boldsymbol{u}}_{t_n,n^\dagger h^\dagger,\lfloor\frac{\tau^\dagger}{\epsilon^\dagger}\rfloor\epsilon^\dagger}^{(K^\dagger-1)}) - \nabla\log\breve{p}_{t_{n+1}}(\widetilde{\boldsymbol{u}}_{t_n,n^\dagger h^\dagger,\tau^\dagger}^{(K^\dagger)}) \right\|^2 \, d\tau^\dagger$$

$$\leq 5\int_0^{h^\dagger} \left\| s_{t_{n+1}}(\widetilde{\boldsymbol{u}}_{t_n,n^\dagger h^\dagger,\lfloor\frac{\tau^\dagger}{\epsilon^\dagger}\rfloor\epsilon^\dagger}^{(K^\dagger-1)}) - s_{t_{n+1}}(\widetilde{\boldsymbol{u}}_{t_n,n^\dagger h^\dagger,\lfloor\frac{\tau^\dagger}{\epsilon^\dagger}\rfloor\epsilon^\dagger}^{(K^\dagger)}) \right\|^2 \, d\tau^\dagger$$

$$+5\int_0^{h^\dagger} \left\| s_{t_{n+1}}(\widetilde{\boldsymbol{u}}_{t_n,n^\dagger h^\dagger,\lfloor\frac{\tau^\dagger}{\epsilon^\dagger}\rfloor\epsilon^\dagger}^{(K^\dagger)}) - s_{t_{n+1}}(\boldsymbol{u}_{t_n,n^\dagger h^\dagger+\lfloor\frac{\tau^\dagger}{\epsilon^\dagger}\rfloor\epsilon^\dagger}^{*}) \right\|^2 \, d\tau^\dagger$$

$$+5\int_0^{h^\dagger} \left\| s_{t_{n+1}}(\boldsymbol{u}_{t_n,n^\dagger h^\dagger,\lfloor\frac{\tau^\dagger}{\epsilon^\dagger}\rfloor\epsilon^\dagger}^{*}) - \nabla\log\breve{p}_{t_{n+1}}(\boldsymbol{u}_{t_n,n^\dagger h^\dagger+\lfloor\frac{\tau^\dagger}{\epsilon^\dagger}\rfloor\epsilon^\dagger}^{*}) \right\|^2 \, d\tau^\dagger$$

$$+5\int_0^{h^\dagger} \left\| \nabla\log\breve{p}_{t_{n+1}}(\boldsymbol{u}_{t_n,n^\dagger h^\dagger+\lfloor\frac{\tau^\dagger}{\epsilon^\dagger}\rfloor\epsilon^\dagger}^{*}) - \nabla\log\breve{p}_{t_{n+1}}(\boldsymbol{u}_{t_n,n^\dagger h^\dagger,\tau^\dagger}^{*}) \right\|^2 \, d\tau^\dagger$$

$$+5\int_0^{h^\dagger} \left\| \nabla\log\breve{p}_{t_{n+1}}(\boldsymbol{u}_{t_n,n^\dagger h^\dagger+\tau^\dagger}^{*}) - \nabla\log\breve{p}_{t_{n+1}}(\widetilde{\boldsymbol{u}}_{t_n,n^\dagger h^\dagger,\tau^\dagger}^{(K^\dagger)}) \right\|^2 \, d\tau^\dagger$$

$$\leq 5L_s^2\int_0^{h^\dagger} \left\| \widetilde{\boldsymbol{u}}_{t_n,n^\dagger h^\dagger,\lfloor\frac{\tau^\dagger}{\epsilon^\dagger}\rfloor\epsilon^\dagger}^{(K^\dagger-1)} - \widetilde{\boldsymbol{u}}_{t_n,n^\dagger h^\dagger,\lfloor\frac{\tau^\dagger}{\epsilon^\dagger}\rfloor\epsilon^\dagger}^{(K^\dagger)} \right\|^2 \, d\tau^\dagger$$

$$+5\underbrace{\left( L_s^2\int_0^{h^\dagger} \left\| \widetilde{\boldsymbol{u}}_{t_n,n^\dagger h^\dagger,\lfloor\frac{\tau^\dagger}{\epsilon^\dagger}\rfloor\epsilon^\dagger}^{(K^\dagger)} - \boldsymbol{u}_{t_n,n^\dagger h^\dagger+\lfloor\frac{\tau^\dagger}{\epsilon^\dagger}\rfloor\epsilon^\dagger}^{*} \right\|^2 \, d\tau^\dagger + L_p^2\int_0^{h^\dagger} \left\| \widetilde{\boldsymbol{u}}_{t_n,n^\dagger h^\dagger,\tau^\dagger}^{(K^\dagger)} - \boldsymbol{u}_{t_n,n^\dagger h^\dagger+\tau^\dagger}^{*} \right\|^2 \, d\tau^\dagger \right)}_{:=E_{t_n,n^\dagger h^\dagger}}$$

$$+5h^\dagger\delta_\infty^2 + 5L_p^2\underbrace{\int_0^{h^\dagger} \left\| \boldsymbol{u}_{t_n,n^\dagger h^\dagger+\lfloor\frac{\tau^\dagger}{\epsilon^\dagger}\rfloor\epsilon^\dagger}^{*} - \boldsymbol{u}_{t_n,n^\dagger h^\dagger+\tau^\dagger}^{*} \right\|^2 \, d\tau^\dagger}_{:=F_{t_n,n^\dagger h^\dagger}},$$

(C.24)

where we used the Lipschitz continuity of the learned score function (Assumption 3.3) and the true score function (Assumption 3.4), and the $\delta_\infty$-accuracy of the learned score function at each time step (Assumption 3.1').

Now we proceed to bound the terms in the error decomposition (C.24). We first bound the $F_{t_n,n^\dagger h^\dagger}$ term by the following lemma:

**Lemma C.13.** *For any $n \in [0 : N-1]$ and $\tau^\dagger \in [0,h^\dagger]$, we have*

$$\mathbb{E}_P\left[ \left\| \boldsymbol{u}_{t_n,n^\dagger h^\dagger+\lfloor\frac{\tau^\dagger}{\epsilon^\dagger}\rfloor\epsilon^\dagger}^{*} - \boldsymbol{u}_{t_n,n^\dagger h^\dagger+\tau^\dagger}^{*} \right\|^2 \right] \leq d\epsilon^{\dagger 2},$$

*and therefore*

$$\mathbb{E}_P\left[ F_{t_n,n^\dagger h^\dagger} \right] \leq dh^\dagger\epsilon^{\dagger 2}.$$

*Proof.* By the definition of $(\boldsymbol{u}_{t_n,n^\dagger h^\dagger+\tau}^{*}, \boldsymbol{v}_{t_n,n^\dagger h^\dagger+\tau}^{*})$ as the stationary underdamped Langevin dynamics (C.22), we have

$$\mathbb{E}_P\left[ \left\| \boldsymbol{u}_{t_n,n^\dagger h^\dagger+\lfloor\frac{\tau^\dagger}{\epsilon^\dagger}\rfloor\epsilon^\dagger}^{*} - \boldsymbol{u}_{t_n,n^\dagger h^\dagger+\tau^\dagger}^{*} \right\|^2 \right] = \mathbb{E}_P\left[ \left\| \int_{\lfloor\frac{\tau^\dagger}{\epsilon^\dagger}\rfloor\epsilon^\dagger}^{\tau^\dagger} \boldsymbol{v}_{t_n,n^\dagger h^\dagger+\tau^{\dagger\prime}}^{*} d\tau^{\dagger\prime} \right\|^2 \right]$$

$$\leq \epsilon^\dagger \int_{\lfloor\frac{\tau^\dagger}{\epsilon^\dagger}\rfloor\epsilon^\dagger}^{\tau^\dagger} \mathbb{E}_P\left[ \left\| \boldsymbol{v}_{t_n,n^\dagger h^\dagger+\tau^{\dagger\prime}}^{*} \right\|^2 \right] d\tau^{\dagger\prime} \leq d\epsilon^{\dagger 2},$$

where the first inequality follows from Cauchy-Schwarz inequality and the last inequality is by the fact that

$$\boldsymbol{v}_{t_n,n^\dagger h^\dagger+\tau^{\dagger\prime}}^{*} \sim \mathcal{N}(0,\boldsymbol{I}_d), \quad \text{for any } \tau^{\dagger\prime} \in [0,h^\dagger].$$

Consequently, we have

$$\mathbb{E}_P\left[F_{t_n,n^\dagger h^\dagger}\right] = \int_0^{h^\dagger} \mathbb{E}_P\left[\left\|\boldsymbol{u}^*_{t_n,n^\dagger h^\dagger+\lfloor\frac{\tau^\dagger}{\epsilon^\dagger}\rfloor\epsilon^\dagger} - \boldsymbol{u}^*_{t_n,n^\dagger h^\dagger+\tau^\dagger}\right\|^2\right]d\tau^\dagger \leq dh^\dagger\epsilon^{\dagger 2}.$$

$\square$

The term $E_{t_n,n^\dagger h^\dagger}$ can be bounded with the following lemma:

**Lemma C.14.** *For any $n^\dagger \in [0 : N^\dagger - 1]$, suppose that $\gamma \lesssim L_p^{-1/2}$ and $T^\dagger \lesssim L_p^{-1/2}$, then we have the following inequality for any $\tau^\dagger \in [0, h^\dagger]$*

$$\mathbb{E}_P\left[\left\|\widetilde{\boldsymbol{u}}^{(K^\dagger)}_{t_n,n^\dagger h^\dagger,\tau^\dagger} - \boldsymbol{u}^*_{t_n,n^\dagger h^\dagger+\tau^\dagger}\right\|^2\right] \lesssim W_2^2(\widetilde{q}_{t_n,h_n}, \breve{p}_{t_{n+1}}),$$

*and therefore*

$$\mathbb{E}_P\left[E_{t_n,n^\dagger h^\dagger}\right] \lesssim h^\dagger(L_s^2 + L_p^2)W_2^2(\widetilde{q}_{t_n,h_n}, \breve{p}_{t_{n+1}}).$$

*Proof.* Recall that under the measure $P$, $\widetilde{\boldsymbol{u}}^{(K^\dagger)}_{t_n,n^\dagger h^\dagger,\tau^\dagger}$ follows the dynamics of $\boldsymbol{u}_{t_n,n^\dagger h^\dagger,\tau^\dagger}$ (C.21) for $\tau^\dagger \in [0, h^\dagger]$, which coincides with that of $\boldsymbol{u}^*_{t_n,n^\dagger h^\dagger+\tau^\dagger}$. As the only difference between the two processes $\boldsymbol{u}_{t_n,n^\dagger h^\dagger,\tau^\dagger}$ and $\boldsymbol{u}^*_{t_n,n^\dagger h^\dagger+\tau^\dagger}$ is the initial condition, we can invoke Lemma 10 proved in [111] to deduce that

$$\mathbb{E}_P\left[\left\|\widetilde{\boldsymbol{u}}^{(K^\dagger)}_{t_n,n^\dagger h^\dagger,\tau^\dagger} - \boldsymbol{u}^*_{t_n,n^\dagger h^\dagger+\tau^\dagger}\right\|^2\right] \lesssim W_2^2(\pi_{t_n,n^\dagger h^\dagger}, \breve{p}_{t_{n+1}}),$$

where the assumption that $\gamma \lesssim L_p^{-1/2}$ and $T^\dagger \lesssim L_p^{-1/2}$ is required.

Now notice that $\boldsymbol{u}^*_{t_n,n^\dagger h^\dagger+\tau^\dagger}$ and $\boldsymbol{u}_{t_n,n^\dagger h^\dagger,\tau^\dagger}$ also follow the same dynamics with the true score function for $\tau^\dagger \in [0, n^\dagger h^\dagger]$, for any coupling of $\boldsymbol{u}^*_{t_n,n^\dagger h^\dagger}$ and $\boldsymbol{u}_{t_n,n^\dagger h^\dagger}$, we have

$$W_2^2(\pi_{t_n,n^\dagger h^\dagger}, \breve{p}_{t_{n+1}}) \leq \mathbb{E}\left[\left\|\boldsymbol{u}_{t_n,n^\dagger h^\dagger} - \boldsymbol{u}^*_{t_n,n^\dagger h^\dagger}\right\|^2\right]$$
$$\leq W_2^2(\pi_{t_n,0}, \breve{p}_{t_{n+1}}) = W_2^2(\widetilde{q}_{t_n,h_n}, \breve{p}_{t_{n+1}}),$$

where the last equality is again by [111, Lemma 10].

Therefore, we have

$$\mathbb{E}_P\left[E_{t_n,n^\dagger h^\dagger}\right]$$
$$= \int_0^{h^\dagger} \mathbb{E}_P\left[L_s^2\left\|\widetilde{\boldsymbol{u}}^{(K^\dagger)}_{t_n,n^\dagger h^\dagger,\lfloor\frac{\tau^\dagger}{\epsilon^\dagger}\rfloor\epsilon^\dagger} - \boldsymbol{u}^*_{t_n,n^\dagger h^\dagger+\lfloor\frac{\tau^\dagger}{\epsilon^\dagger}\rfloor\epsilon^\dagger}\right\|^2 + L_p^2\left\|\widetilde{\boldsymbol{u}}^{(K^\dagger)}_{t_n,n^\dagger h^\dagger,\tau^\dagger} - \boldsymbol{u}^*_{t_n,n^\dagger h^\dagger+\tau^\dagger}\right\|^2\right]d\tau^\dagger$$
$$\leq h^\dagger(L_s^2 + L_p^2)W_2^2(\widetilde{q}_{t_n,h_n}, \breve{p}_{t_{n+1}}).$$

$\square$

Now, we provide lemmas that are used to bound the first term in (C.24).

**Lemma C.15.** *For any $n^\dagger \in [0 : N^\dagger - 1]$, we have the following estimate:*

$$\sup_{\tau^\dagger\in[0,h^\dagger]} \mathbb{E}_P\left[\left\|\widetilde{\boldsymbol{u}}^{(1)}_{t_n,n^\dagger h^\dagger,\tau^\dagger} - \widetilde{\boldsymbol{u}}^{(0)}_{t_n,n^\dagger h^\dagger,\tau^\dagger}\right\|^2\right]$$
$$\leq \frac{5L_s^2 h^\dagger e^{(3+\gamma)h^\dagger}}{2\gamma} \sup_{\tau^\dagger\in[0,h^\dagger]} \mathbb{E}_P\left[\left\|\widetilde{\boldsymbol{u}}^{(K^\dagger-1)}_{t_n,n^\dagger h^\dagger,\tau^\dagger} - \widetilde{\boldsymbol{u}}^{(K^\dagger)}_{t_n,n^\dagger h^\dagger,\tau^\dagger}\right\|^2\right]$$
$$+ \frac{5h^\dagger e^{(3+\gamma)h^\dagger}}{2\gamma}\mathbb{E}_P\left[E_{t_n,n^\dagger h^\dagger} + h^\dagger\delta_\infty^2 + L_p^2 F_{t_n,n^\dagger h^\dagger}\right] + h^{\dagger 2}e^{(3+\gamma)h^\dagger}\left(3\gamma d + M_s^2\right) + h^\dagger e^{2h^\dagger}d.$$

*Proof.* Let $\boldsymbol{\mu}_{t_n,n^\dagger h^\dagger,\tau^\dagger} := \widetilde{\boldsymbol{u}}^{(1)}_{t_n,n^\dagger h^\dagger,\tau^\dagger} - \widetilde{\boldsymbol{u}}^{(0)}_{t_n,n^\dagger h^\dagger,\tau^\dagger}$ and $\boldsymbol{\nu}_{t_n,n^\dagger h^\dagger,\tau^\dagger} := \widetilde{\boldsymbol{v}}^{(1)}_{t_n,n^\dagger h^\dagger,\tau^\dagger} - \widetilde{\boldsymbol{v}}^{(0)}_{t_n,n^\dagger h^\dagger,\tau^\dagger}$.
Then for $k = 0$, we may rewrite (C.15) as follows

$$
\begin{cases}
\mathrm{d}\boldsymbol{\mu}_{t_n,n^\dagger h^\dagger,\tau^\dagger} = \left(\boldsymbol{\nu}_{t_n,n^\dagger h^\dagger,\tau^\dagger} + \widetilde{\boldsymbol{v}}^{(0)}_{t_n,n^\dagger h^\dagger,\tau^\dagger}\right)\mathrm{d}\tau^\dagger \\
\mathrm{d}\boldsymbol{\nu}_{t_n,n^\dagger h^\dagger,\tau^\dagger} = -\gamma(\boldsymbol{\nu}_{t_n,n^\dagger h^\dagger,\tau^\dagger} + \widetilde{\boldsymbol{v}}^{(0)}_{t_n,n^\dagger h^\dagger,\tau^\dagger})\mathrm{d}\tau^\dagger - \boldsymbol{s}_{t_{n+1}}(\widetilde{\boldsymbol{u}}^{(0)}_{t_n,n^\dagger h^\dagger,\tau^\dagger})\mathrm{d}\tau^\dagger + \sqrt{2\gamma}\mathrm{d}\boldsymbol{b}_{t_n,n^\dagger h^\dagger+\tau^\dagger}
\end{cases}
$$
(C.25)

On the one hand, by using the first equation in (C.25), we may compute the derivative

$$
\frac{\mathrm{d}}{\mathrm{d}\tau^{\dagger'}}\left\|\boldsymbol{\mu}_{t_n,n^\dagger h^\dagger,\tau^{\dagger'}}\right\|^2 = 2\boldsymbol{\mu}^\top_{t_n,n^\dagger h^\dagger,\tau^{\dagger'}}\left(\boldsymbol{\nu}_{t_n,n^\dagger h^\dagger,\tau^{\dagger'}} + \widetilde{\boldsymbol{v}}^{(0)}_{t_n,n^\dagger h^\dagger,\tau^{\dagger'}}\right)
$$

and integrate it for $\tau^{\dagger'} \in [0, \tau^\dagger]$, which yields

$$
\left\|\boldsymbol{\mu}_{t_n,n^\dagger h^\dagger,\tau^\dagger}\right\|^2 = 2\int_0^{\tau^\dagger}\boldsymbol{\mu}^\top_{t_n,n^\dagger h^\dagger,\tau^{\dagger'}}(\boldsymbol{\nu}_{t_n,n^\dagger h^\dagger,\tau^{\dagger'}} + \widetilde{\boldsymbol{v}}^{(0)}_{t_n,n^\dagger h^\dagger,\tau^{\dagger'}})\mathrm{d}\tau^{\dagger'}
$$

$$
\leq 2\int_0^{\tau^\dagger}\left\|\boldsymbol{\mu}_{t_n,n^\dagger h^\dagger,\tau^{\dagger'}}\right\|^2\mathrm{d}\tau^{\dagger'} + \int_0^{\tau^\dagger}\left\|\boldsymbol{\nu}_{t_n,n^\dagger h^\dagger,\tau^{\dagger'}}\right\|^2\mathrm{d}\tau^{\dagger'} + \int_0^{\tau^\dagger}\left\|\widetilde{\boldsymbol{v}}^{(0)}_{t_n,n^\dagger h^\dagger,\tau^{\dagger'}}\right\|^2\mathrm{d}\tau^{\dagger'}.
$$

Applying Gronwall's inequality, we have

$$
\left\|\boldsymbol{\mu}_{t_n,n^\dagger h^\dagger,\tau^\dagger}\right\|^2 \leq e^{2\tau^\dagger}\left(\int_0^{\tau^\dagger}\left\|\boldsymbol{\nu}_{t_n,n^\dagger h^\dagger,\tau^{\dagger'}}\right\|^2\mathrm{d}\tau^{\dagger'} + \int_0^{\tau^\dagger}\left\|\widetilde{\boldsymbol{v}}^{(0)}_{t_n,n^\dagger h^\dagger,\tau^{\dagger'}}\right\|^2\mathrm{d}\tau^{\dagger'}\right).
$$

We then take expectation with respect to the path measure $P$ and then the supremum with respect to $\tau^\dagger \in [0, h^\dagger]$, implying that

$$
\sup_{\tau^\dagger\in[0,h^\dagger]}\mathbb{E}_P\left[\left\|\boldsymbol{\mu}_{t_n,n^\dagger h^\dagger,\tau^\dagger}\right\|^2\right]
$$

$$
\leq \sup_{\tau^\dagger\in[0,h^\dagger]}\left(e^{2\tau^\dagger}\int_0^{\tau^\dagger}\mathbb{E}_P\left[\left\|\boldsymbol{\nu}_{t_n,n^\dagger h^\dagger,\tau^{\dagger'}}\right\|^2\right]\mathrm{d}\tau^{\dagger'} + e^{2\tau^\dagger}\int_0^{\tau^\dagger}\mathbb{E}_P\left[\left\|\widetilde{\boldsymbol{v}}^{(0)}_{t_n,n^\dagger h^\dagger,\tau^{\dagger'}}\right\|^2\right]\mathrm{d}\tau^{\dagger'}\right)
$$

$$
\leq h^\dagger e^{2h^\dagger}\sup_{\tau^\dagger\in[0,h^\dagger]}\mathbb{E}_P\left[\left\|\boldsymbol{\nu}_{t_n,n^\dagger h^\dagger,\tau^{\dagger'}}\right\|^2\right] + h^\dagger e^{2h^\dagger}d.
$$
(C.26)

On the other hand, by applying Itô's lemma and plugging in the expression of $\boldsymbol{b}_{t_n,n^\dagger h^\dagger+\tau^\dagger}$ given by (C.17), we have

$$
\mathrm{d}\|\boldsymbol{\nu}_{t_n,n^\dagger h^\dagger,\tau^\dagger}\|^2
$$
$$
= -\left[2\gamma\|\boldsymbol{\nu}_{t_n,n^\dagger h^\dagger,\tau^\dagger}\|^2 + 2\gamma\boldsymbol{\nu}^\top_{t_n,n^\dagger h^\dagger,\tau^\dagger}\widetilde{\boldsymbol{v}}^{(0)}_{t_n,n^\dagger h^\dagger,\tau^\dagger} + 2\boldsymbol{\nu}^\top_{t_n,n^\dagger h^\dagger,\tau^\dagger}\boldsymbol{s}_{t_{n+1}}\left(\widetilde{\boldsymbol{u}}^{(0)}_{t_n,n^\dagger h^\dagger,\tau^\dagger}\right) - 2\gamma d\right]\mathrm{d}\tau^\dagger
$$
$$
+ 2\boldsymbol{\nu}^\top_{t_n,n^\dagger h^\dagger,\tau^\dagger}\sqrt{2\gamma}(\mathrm{d}\widetilde{\boldsymbol{b}}_{t_n,n^\dagger h^\dagger+\tau^\dagger} + \boldsymbol{\phi}_{t_n,n^\dagger h^\dagger}(\tau^\dagger)\mathrm{d}\tau^\dagger),
$$
(C.27)

Then similarly, we may compute the derivative of $\|\boldsymbol{\nu}_{t_n,n^\dagger h^\dagger,\tau^\dagger}\|^2$, integrate it for $\tau^\dagger \in [0, h^\dagger]$, and take the supremum with respect to $\tau^\dagger$ to obtain

$$
\mathbb{E}_P\left[\|\boldsymbol{\nu}_{t_n,n^\dagger h^\dagger,\tau^\dagger}\|^2\right]
$$
$$
= \mathbb{E}_P\left[-\int_0^{\tau^\dagger}\left(2\gamma\|\boldsymbol{\nu}_{t_n,n^\dagger h^\dagger,\tau^{\dagger'}}\|^2 + 2\gamma\boldsymbol{\nu}^\top_{t_n,n^\dagger h^\dagger,\tau^{\dagger'}}\widetilde{\boldsymbol{v}}^{(0)}_{t_n,n^\dagger h^\dagger,\tau^{\dagger'}} - 2\gamma d\right)\mathrm{d}\tau^{\dagger'}\right]
$$
$$
+ \mathbb{E}_P\left[-\int_0^{\tau^\dagger}2\boldsymbol{\nu}^\top_{t_n,n^\dagger h^\dagger,\tau^{\dagger'}}\boldsymbol{s}_{t_{n+1}}\left(\widetilde{\boldsymbol{u}}^{(0)}_{t_n,n^\dagger h^\dagger,\tau^{\dagger'}}\right)\mathrm{d}\tau^{\dagger'}\right]
$$
$$
+ 2\sqrt{2\gamma}\mathbb{E}_P\left[\int_0^{\tau^\dagger}\boldsymbol{\nu}^\top_{t_n,n^\dagger h^\dagger,\tau^{\dagger'}}\left(\mathrm{d}\widetilde{\boldsymbol{b}}_{t_n,n^\dagger h^\dagger+\tau^{\dagger'}} + \boldsymbol{\phi}_{t_n,n^\dagger h^\dagger}(\tau^{\dagger'})\mathrm{d}\tau^{\dagger'}\right)\right].
$$

By Itô's lemma, this equals to

$$\mathbb{E}_P\left[\|\boldsymbol{\nu}_{t_n,n^\dagger h^\dagger,\tau^\dagger}\|^2\right]$$

$$=\mathbb{E}_P\left[-\int_0^{\tau^\dagger}\left(2\gamma\|\boldsymbol{\nu}_{t_n,n^\dagger h^\dagger,\tau^{\dagger\prime}}\|^2+2\gamma\boldsymbol{\nu}_{t_n,n^\dagger h^\dagger,\tau^{\dagger\prime}}^\top\widetilde{\boldsymbol{v}}_{t_n,n^\dagger h^\dagger,\tau^{\dagger\prime}}^{(0)}-2\gamma d\right)\mathrm{d}\tau^{\dagger\prime}\right]$$

$$+\mathbb{E}_P\left[-\int_0^{\tau^\dagger}2\boldsymbol{\nu}_{t_n,n^\dagger h^\dagger,\tau^{\dagger\prime}}^\top\boldsymbol{s}_{t_{n+1}}\left(\widetilde{\boldsymbol{u}}_{t_n,n^\dagger h^\dagger,\tau^{\dagger\prime}}^{(0)}\right)+2\sqrt{2\gamma}\boldsymbol{\nu}_{t_n,n^\dagger h^\dagger,\tau^{\dagger\prime}}^\top\boldsymbol{\phi}_{t_n,n^\dagger h^\dagger}(\tau^{\dagger\prime})\mathrm{d}\tau^{\dagger\prime}\right].$$

Applying AM-GM gives

$$\mathbb{E}_P\left[\|\boldsymbol{\nu}_{t_n,n^\dagger h^\dagger,\tau^\dagger}\|^2\right]$$

$$\leq\int_0^{\tau^\dagger}\mathbb{E}_P\left[(1+\gamma)\|\boldsymbol{\nu}_{t_n,n^\dagger h^\dagger,\tau^{\dagger\prime}}\|^2+\|\boldsymbol{\phi}_{t_n,n^\dagger h^\dagger}(\tau^{\dagger\prime})\|^2\right]\mathrm{d}\tau^{\dagger\prime}$$

$$+\int_0^{\tau^\dagger}\mathbb{E}_P\left[\gamma\left\|\widetilde{\boldsymbol{v}}_{t_n,n^\dagger h^\dagger,\tau^{\dagger\prime}}^{(0)}\right\|^2+\left\|\boldsymbol{s}_{t_{n+1}}\left(\widetilde{\boldsymbol{u}}_{t_n,n^\dagger h^\dagger,\tau^{\dagger\prime}}^{(0)}\right)\right\|^2+2\gamma d\right]\mathrm{d}\tau^{\dagger\prime}$$

$$\leq\int_0^{\tau^\dagger}\mathbb{E}_P\left[(1+\gamma)\|\boldsymbol{\nu}_{t_n,n^\dagger h^\dagger,\tau^{\dagger\prime}}\|^2+\|\boldsymbol{\phi}_{t_n,n^\dagger h^\dagger}(\tau^{\dagger\prime})\|^2\right]\mathrm{d}\tau^{\dagger\prime}+\left(\gamma\mathbb{E}\left[\left\|\widetilde{\boldsymbol{v}}_{t_n,n^\dagger h^\dagger,0}^{(0)}\right\|^2\right]+M_{\boldsymbol{s}}^2+2\gamma d\right)\tau^\dagger$$

$$=(1+\gamma)\int_0^{\tau^\dagger}\mathbb{E}_P\left[\|\boldsymbol{\nu}_{t_n,n^\dagger h^\dagger,\tau^{\dagger\prime}}\|^2\right]\mathrm{d}\tau^{\dagger\prime}+\int_0^{\tau^\dagger}\mathbb{E}_P\left[\|\boldsymbol{\phi}_{t_n,n^\dagger h^\dagger}(\tau^{\dagger\prime})\|^2\right]\mathrm{d}\tau^{\dagger\prime}+\tau^\dagger\left(3\gamma d+M_{\boldsymbol{s}}^2\right),$$

where in the last equality, we used the initialization of the auxiliary corrector process $\widetilde{\boldsymbol{v}}_{t_n,n^\dagger h^\dagger,0}^{(0)}\sim\mathcal{N}(0,\boldsymbol{I}_d)$.

Again, we apply Gronwall's inequality to the above inequality and take the supremum with respect to $\tau^\dagger\in[0,h^\dagger]$ to obtain

$$\sup_{\tau^\dagger\in[0,h^\dagger]}\mathbb{E}_P\left[\|\boldsymbol{\nu}_{t_n,n^\dagger h^\dagger,\tau^\dagger}\|^2\right]$$

$$\leq e^{(1+\gamma)h^\dagger}\int_0^{h^\dagger}\mathbb{E}_P\left[\|\boldsymbol{\phi}_{t_n,n^\dagger h^\dagger}(\tau^\dagger)\|^2\right]\mathrm{d}\tau^\dagger+h^\dagger e^{(1+\gamma)h^\dagger}\left(3\gamma d+M_{\boldsymbol{s}}^2\right)$$

$$\leq\frac{e^{(1+\gamma)h^\dagger}}{2\gamma}\mathbb{E}_P\left[\int_0^{h^\dagger}\left\|\boldsymbol{s}_{t_{n+1}}(\widetilde{\boldsymbol{u}}_{t_n,n^\dagger h^\dagger,\lfloor\frac{\tau^\dagger}{\epsilon^\dagger}\rfloor\epsilon^\dagger}^{(K^\dagger-1)})-\nabla\log\breve{p}_{t_{n+1}}(\widetilde{\boldsymbol{u}}_{t_n,n^\dagger h^\dagger,\tau^\dagger}^{(K^\dagger)})\right\|^2\mathrm{d}\tau^\dagger\right] \tag{C.28}$$

$$+h^\dagger e^{(1+\gamma)h^\dagger}\left(3\gamma d+M_{\boldsymbol{s}}^2\right),$$

and for the difference term within the expectation, we decompose it again by the triangle inequality in (C.24), *i.e.*

$$\int_0^{h^\dagger}\left\|\boldsymbol{s}_{t_{n+1}}(\widetilde{\boldsymbol{u}}_{t_n,n^\dagger h^\dagger,\lfloor\frac{\tau^\dagger}{\epsilon^\dagger}\rfloor\epsilon^\dagger}^{(K^\dagger-1)})-\nabla\log\breve{p}_{t_{n+1}}(\widetilde{\boldsymbol{u}}_{t_n,n^\dagger h^\dagger,\tau^\dagger}^{(K^\dagger)})\right\|^2\mathrm{d}\tau^\dagger$$

$$\leq 5L_{\boldsymbol{s}}^2\int_0^{h^\dagger}\left\|\widetilde{\boldsymbol{u}}_{t_n,n^\dagger h^\dagger,\lfloor\frac{\tau^\dagger}{\epsilon^\dagger}\rfloor\epsilon^\dagger}^{(K^\dagger-1)}-\widetilde{\boldsymbol{u}}_{t_n,n^\dagger h^\dagger,\lfloor\frac{\tau^\dagger}{\epsilon^\dagger}\rfloor\epsilon^\dagger}^{(K^\dagger)}\right\|^2\mathrm{d}\tau^\dagger+5E_{t_n,n^\dagger h^\dagger}+5h^\dagger\delta_\infty^2+5L_p^2F_{t_n,n^\dagger h^\dagger},$$

to obtain that

$$\sup_{\tau^\dagger\in[0,h^\dagger]}\mathbb{E}_P\left[\|\boldsymbol{\nu}_{t_n,n^\dagger h^\dagger,\tau^\dagger}\|^2\right]$$

$$\leq\frac{5L_{\boldsymbol{s}}^2 e^{(1+\gamma)h^\dagger}}{2\gamma}\mathbb{E}_P\left[\int_0^{h^\dagger}\left\|\widetilde{\boldsymbol{u}}_{t_n,n^\dagger h^\dagger,\lfloor\frac{\tau^\dagger}{\epsilon^\dagger}\rfloor\epsilon^\dagger}^{(K^\dagger-1)}-\widetilde{\boldsymbol{u}}_{t_n,n^\dagger h^\dagger,\lfloor\frac{\tau^\dagger}{\epsilon^\dagger}\rfloor\epsilon^\dagger}^{(K^\dagger)}\right\|^2\mathrm{d}\tau^\dagger\right]$$

$$+\frac{5e^{(1+\gamma)h^\dagger}}{2\gamma}\mathbb{E}_P\left[E_{t_n,n^\dagger h^\dagger}+h^\dagger\delta_\infty^2+L_p^2F_{t_n,n^\dagger h^\dagger}\right]+h^\dagger e^{(1+\gamma)h^\dagger}\left(3\gamma d+M_{\boldsymbol{s}}^2\right)$$

$$\leq\frac{5L_{\boldsymbol{s}}^2 e^{(1+\gamma)h^\dagger}}{2\gamma}h^\dagger\sup_{\tau^\dagger\in[0,h^\dagger]}\mathbb{E}_P\left[\left\|\widetilde{\boldsymbol{u}}_{t_n,n^\dagger h^\dagger,\lfloor\frac{\tau^\dagger}{\epsilon^\dagger}\rfloor\epsilon^\dagger}^{(K^\dagger-1)}-\widetilde{\boldsymbol{u}}_{t_n,n^\dagger h^\dagger,\lfloor\frac{\tau^\dagger}{\epsilon^\dagger}\rfloor\epsilon^\dagger}^{(K^\dagger)}\right\|^2\right]$$

$$+\frac{5e^{(1+\gamma)h^\dagger}}{2\gamma}\mathbb{E}_P\left[E_{t_n,n^\dagger h^\dagger}+h^\dagger\delta_\infty^2+L_p^2F_{t_n,n^\dagger h^\dagger}\right]+h^\dagger e^{(1+\gamma)h^\dagger}\left(3\gamma d+M_{\boldsymbol{s}}^2\right),$$

substituting which into (C.26) completes our proof of this Lemma. $\qquad\square$

**Lemma C.16** (Exponential convergence of Picard iteration in the corrector step of PIADM-ODE).
*For any $n^\dagger \in [0, N^\dagger - 1]$, then the two ending terms $\widetilde{\boldsymbol{u}}_{n^\dagger h^\dagger, \tau^\dagger}^{(K^\dagger)}$ and $\widetilde{\boldsymbol{u}}_{n^\dagger h^\dagger, \tau^\dagger}^{(K^\dagger)}$ of the sequence $\{\widetilde{\boldsymbol{u}}_{n^\dagger h^\dagger, \tau^\dagger}^{(k^\dagger)}\}_{k^\dagger \in [0:K^\dagger - 1]}$ satisfy the following exponential convergence rate*

$$\sup_{\tau^\dagger \in [0,h^\dagger]} \mathbb{E}_P \left[ \left\| \widetilde{\boldsymbol{u}}_{n^\dagger h^\dagger, \tau^\dagger}^{(K^\dagger)} - \widetilde{\boldsymbol{u}}_{n^\dagger h^\dagger, \tau^\dagger}^{(K^\dagger - 1)} \right\|^2 \right]$$

$$\leq C_{K^\dagger} \left( \frac{5 h^\dagger e^{(3+\gamma)h^\dagger}}{2\gamma} \mathbb{E}_P \left[ E_{t_n, n^\dagger h^\dagger} + h^\dagger \delta_\infty^2 + L_p^2 F_{t_n, n^\dagger h^\dagger} \right] + h^{\dagger 2} e^{(3+\gamma)h^\dagger} \left( 3\gamma d + M_s^2 \right) + h^\dagger e^{2h^\dagger} d \right),$$
(C.29)

*where the coefficient*

$$C_{K^\dagger} = \left( \frac{L_s^2 h^{\dagger 2} e^{h^\dagger}}{2\gamma} \right)^{K^\dagger - 1} \Bigg/ \left( 1 - \frac{5 L_s^2 h^\dagger e^{(3+\gamma)h^\dagger}}{2\gamma} \left( \frac{L_s^2 h^{\dagger 2} e^{h^\dagger}}{2\gamma} \right)^{K^\dagger - 1} \right).$$

*Proof.* We subtract the dynamics of $\widetilde{\boldsymbol{u}}_{n^\dagger h^\dagger, \tau^\dagger}^{(k+1)}$ and $\widetilde{\boldsymbol{u}}_{n^\dagger h^\dagger, \tau^\dagger}^{(k)}$ in (C.15) to obtain

$$\mathrm{d} \left( \widetilde{\boldsymbol{u}}_{n^\dagger h^\dagger, \tau^\dagger}^{(k+1)} - \widetilde{\boldsymbol{u}}_{n^\dagger h^\dagger, \tau^\dagger}^{(k^\dagger)} \right) = \left( \widetilde{\boldsymbol{v}}_{n^\dagger h^\dagger, \tau^\dagger}^{(k+1)} - \widetilde{\boldsymbol{v}}_{n^\dagger h^\dagger, \tau^\dagger}^{(k^\dagger)} \right) \mathrm{d}\tau^\dagger.$$

Then, we use the formula above to compute the derivative

$$\frac{\mathrm{d}}{\mathrm{d}\tau^{\dagger'}} \left\| \widetilde{\boldsymbol{u}}_{n^\dagger h^\dagger, \tau^{\dagger'}}^{(k+1)} - \widetilde{\boldsymbol{u}}_{n^\dagger h^\dagger, \tau^{\dagger'}}^{(k^\dagger)} \right\|^2 = 2 \left( \widetilde{\boldsymbol{u}}_{n^\dagger h^\dagger, \tau^{\dagger'}}^{(k+1)} - \widetilde{\boldsymbol{u}}_{n^\dagger h^\dagger, \tau^{\dagger'}}^{(k^\dagger)} \right)^\top \left( \widetilde{\boldsymbol{v}}_{n^\dagger h^\dagger, \tau^{\dagger'}}^{(k+1)} - \widetilde{\boldsymbol{v}}_{n^\dagger h^\dagger, \tau^{\dagger'}}^{(k^\dagger)} \right)$$

and integrate for $\tau^{\dagger'} \in [0, \tau^\dagger]$ to obtain

$$\left\| \widetilde{\boldsymbol{u}}_{n^\dagger h^\dagger, \tau^\dagger}^{(k+1)} - \widetilde{\boldsymbol{u}}_{n^\dagger h^\dagger, \tau^\dagger}^{(k^\dagger)} \right\|^2$$

$$= 2 \int_0^{\tau^\dagger} \left( \widetilde{\boldsymbol{u}}_{n^\dagger h^\dagger, \tau^{\dagger'}}^{(k+1)} - \widetilde{\boldsymbol{u}}_{n^\dagger h^\dagger, \tau^{\dagger'}}^{(k^\dagger)} \right)^\top \left( \widetilde{\boldsymbol{v}}_{n^\dagger h^\dagger, \tau^{\dagger'}}^{(k+1)} - \widetilde{\boldsymbol{v}}_{n^\dagger h^\dagger, \tau^{\dagger'}}^{(k^\dagger)} \right) \mathrm{d}\tau^{\dagger'}$$

$$\leq \int_0^{\tau^\dagger} \left\| \widetilde{\boldsymbol{u}}_{n^\dagger h^\dagger, \tau^{\dagger'}}^{(k+1)} - \widetilde{\boldsymbol{u}}_{n^\dagger h^\dagger, \tau^{\dagger'}}^{(k^\dagger)} \right\|^2 \mathrm{d}\tau^{\dagger'} + \int_0^{\tau^\dagger} \left\| \widetilde{\boldsymbol{v}}_{n^\dagger h^\dagger, \tau^{\dagger'}}^{(k+1)} - \widetilde{\boldsymbol{v}}_{n^\dagger h^\dagger, \tau^{\dagger'}}^{(k^\dagger)} \right\|^2 \mathrm{d}\tau^{\dagger'}$$

Applying Grönwall's inequality gives us that

$$\left\| \widetilde{\boldsymbol{u}}_{n^\dagger h^\dagger, \tau^\dagger}^{(k+1)} - \widetilde{\boldsymbol{u}}_{n^\dagger h^\dagger, \tau^\dagger}^{(k^\dagger)} \right\|^2 \leq e^{\tau^\dagger} \int_0^{\tau^\dagger} \left\| \widetilde{\boldsymbol{v}}_{n^\dagger h^\dagger, \tau^{\dagger'}}^{(k+1)} - \widetilde{\boldsymbol{v}}_{n^\dagger h^\dagger, \tau^{\dagger'}}^{(k^\dagger)} \right\|^2 \mathrm{d}\tau^{\dagger'}$$

and taking the supremum with respect to $\tau^\dagger \in [0, h^\dagger]$ on both sides above implies

$$\sup_{\tau^\dagger \in [0,h^\dagger]} \mathbb{E}_P \left[ \left\| \widetilde{\boldsymbol{u}}_{n^\dagger h^\dagger, \tau^\dagger}^{(k+1)} - \widetilde{\boldsymbol{u}}_{n^\dagger h^\dagger, \tau^\dagger}^{(k^\dagger)} \right\|^2 \right] \leq h^\dagger e^{h^\dagger} \sup_{\tau^\dagger \in [0,h^\dagger]} \mathbb{E}_P \left[ \left\| \widetilde{\boldsymbol{v}}_{n^\dagger h^\dagger, \tau^{\dagger'}}^{(k+1)} - \widetilde{\boldsymbol{v}}_{n^\dagger h^\dagger, \tau^{\dagger'}}^{(k^\dagger)} \right\|^2 \right].$$
(C.30)

We then apply a similar argument for $\widetilde{\boldsymbol{v}}_{n^\dagger h^\dagger, \tau^\dagger}^{(k+1)} - \widetilde{\boldsymbol{v}}_{n^\dagger h^\dagger, \tau^\dagger}^{(k^\dagger)}$ as well

$$\mathrm{d} \left( \widetilde{\boldsymbol{v}}_{t_n, n^\dagger h^\dagger, \tau^\dagger}^{(k+1)} - \widetilde{\boldsymbol{v}}_{t_n, n^\dagger h^\dagger, \tau^\dagger}^{(k^\dagger)} \right)$$

$$= -\gamma \left( \widetilde{\boldsymbol{v}}_{t_n, n^\dagger h^\dagger, \tau^\dagger}^{(k+1)} - \widetilde{\boldsymbol{v}}_{t_n, n^\dagger h^\dagger, \tau^\dagger}^{(k^\dagger)} \right) \mathrm{d}\tau^\dagger - \left( \boldsymbol{s}_{t_{n+1}} (\widetilde{\boldsymbol{u}}_{t_n, n^\dagger h^\dagger, \lfloor \frac{\tau^\dagger}{\epsilon^\dagger} \rfloor \epsilon^\dagger}^{(k^\dagger)}) - \boldsymbol{s}_{t_{n+1}} (\widetilde{\boldsymbol{u}}_{t_n, n^\dagger h^\dagger, \lfloor \frac{\tau^\dagger}{\epsilon^\dagger} \rfloor \epsilon^\dagger}^{(k-1)}) \right) \mathrm{d}\tau^\dagger,$$

integrate which for $\tau^\dagger \in [0, \tau^\dagger]$ to obtain

$$\left\| \widetilde{\boldsymbol{v}}_{n^\dagger h^\dagger, \tau^\dagger}^{(k+1)} - \widetilde{\boldsymbol{v}}_{n^\dagger h^\dagger, \tau^\dagger}^{(k^\dagger)} \right\|^2$$

$$= - \int_0^{\tau^\dagger} 2\gamma \left\| \widetilde{\boldsymbol{v}}_{n^\dagger h^\dagger, \tau^{\dagger\prime}}^{(k+1)} - \widetilde{\boldsymbol{v}}_{n^\dagger h^\dagger, \tau^{\dagger\prime}}^{(k^\dagger)} \right\|^2 \mathrm{d}\tau^{\dagger\prime}$$

$$- 2 \int_0^{\tau^\dagger} \left( \widetilde{\boldsymbol{v}}_{n^\dagger h^\dagger, \tau^{\dagger\prime}}^{(k+1)} - \widetilde{\boldsymbol{v}}_{n^\dagger h^\dagger, \tau^{\dagger\prime}}^{(k^\dagger)} \right)^\top \left( \boldsymbol{s}_{t_{n+1}}(\widetilde{\boldsymbol{u}}_{t_n, n^\dagger h^\dagger, \lfloor \frac{\tau^{\dagger\prime}}{\epsilon^\dagger} \rfloor \epsilon^\dagger}^{(k^\dagger)}) - \boldsymbol{s}_{t_{n+1}}(\widetilde{\boldsymbol{u}}_{t_n, n^\dagger h^\dagger, \lfloor \frac{\tau^{\dagger\prime}}{\epsilon^\dagger} \rfloor \epsilon^\dagger}^{(k-1)}) \right) \mathrm{d}\tau^{\dagger\prime}$$

$$\leq \frac{1}{2\gamma} \int_0^{\tau^\dagger} \left\| \boldsymbol{s}_{t_{n+1}}(\widetilde{\boldsymbol{u}}_{t_n, n^\dagger h^\dagger, \lfloor \frac{\tau^{\dagger\prime}}{\epsilon^\dagger} \rfloor \epsilon^\dagger}^{(k^\dagger)}) - \boldsymbol{s}_{t_{n+1}}(\widetilde{\boldsymbol{u}}_{t_n, n^\dagger h^\dagger, \lfloor \frac{\tau^{\dagger\prime}}{\epsilon^\dagger} \rfloor \epsilon^\dagger}^{(k-1)}) \right\|^2 \mathrm{d}\tau^{\dagger\prime}$$

$$\leq \frac{L_{\boldsymbol{s}}^2}{2\gamma} \int_0^{\tau^\dagger} \left\| \widetilde{\boldsymbol{u}}_{t_n, n^\dagger h^\dagger, \lfloor \frac{\tau^{\dagger\prime}}{\epsilon^\dagger} \rfloor \epsilon^\dagger}^{(k^\dagger)} - \widetilde{\boldsymbol{u}}_{t_n, n^\dagger h^\dagger, \lfloor \frac{\tau^{\dagger\prime}}{\epsilon^\dagger} \rfloor \epsilon^\dagger}^{(k-1)} \right\|^2 \mathrm{d}\tau^{\dagger\prime}.$$

And then taking the supremum with respect to $\tau^\dagger \in [0, h^\dagger]$ on both sides above implies

$$\sup_{\tau^\dagger \in [0, h^\dagger]} \mathbb{E}_P \left[ \left\| \widetilde{\boldsymbol{v}}_{n^\dagger h^\dagger, \tau^\dagger}^{(k+1)} - \widetilde{\boldsymbol{v}}_{n^\dagger h^\dagger, \tau^\dagger}^{(k^\dagger)} \right\|^2 \right] \leq \frac{h^\dagger L_{\boldsymbol{s}}^2}{2\gamma} \sup_{\tau^\dagger \in [0, h^\dagger]} \mathbb{E}_P \left[ \left\| \widetilde{\boldsymbol{u}}_{t_n, n^\dagger h^\dagger, \tau^\dagger}^{(k^\dagger)} - \widetilde{\boldsymbol{u}}_{t_n, n^\dagger h^\dagger, \tau^\dagger}^{(k-1)} \right\|^2 \right]$$

(C.31)

Substituting (C.31) into (C.30) and iterating over $k \in [1 : K^\dagger - 1]$, we obtain that

$$\sup_{\tau^\dagger \in [0, h^\dagger]} \mathbb{E}_P \left[ \left\| \widetilde{\boldsymbol{u}}_{n^\dagger h^\dagger, \tau^\dagger}^{(K^\dagger)} - \widetilde{\boldsymbol{u}}_{n^\dagger h^\dagger, \tau^\dagger}^{(K^\dagger - 1)} \right\|^2 \right] \leq \frac{L_{\boldsymbol{s}}^2 h^{\dagger 2} e^{h^\dagger}}{2\gamma} \sup_{\tau^\dagger \in [0, h^\dagger]} \mathbb{E}_P \left[ \left\| \widetilde{\boldsymbol{u}}_{t_n, n^\dagger h^\dagger, \tau^\dagger}^{(K^\dagger - 1)} - \widetilde{\boldsymbol{u}}_{t_n, n^\dagger h^\dagger, \tau^\dagger}^{(K^\dagger - 2)} \right\|^2 \right]$$

$$\leq \left( \frac{L_{\boldsymbol{s}}^2 h^{\dagger 2} e^{h^\dagger}}{2\gamma} \right)^{K^\dagger - 1} \sup_{\tau^\dagger \in [0, h^\dagger]} \mathbb{E}_P \left[ \left\| \widetilde{\boldsymbol{u}}_{t_n, n^\dagger h^\dagger, \tau^\dagger}^{(1)} - \widetilde{\boldsymbol{u}}_{t_n, n^\dagger h^\dagger, \tau^\dagger}^{(0)} \right\|^2 \right]$$

$$\leq \left( \frac{L_{\boldsymbol{s}}^2 h^{\dagger 2} e^{h^\dagger}}{2\gamma} \right)^{K^\dagger - 1} \frac{5 h^\dagger e^{(3+\gamma) h^\dagger}}{2\gamma} \mathbb{E}_P \left[ E_{t_n, n^\dagger h^\dagger} + h^\dagger \delta_\infty^2 + L_p^2 F_{t_n, n^\dagger h^\dagger} \right]$$

$$+ \left( \frac{L_{\boldsymbol{s}}^2 h^{\dagger 2} e^{h^\dagger}}{2\gamma} \right)^{K^\dagger - 1} \left( h^{\dagger 2} e^{(3+\gamma) h^\dagger} \left( 3\gamma d + M_{\boldsymbol{s}}^2 \right) + h^\dagger e^{2 h^\dagger} d \right)$$

$$+ \left( \frac{L_{\boldsymbol{s}}^2 h^{\dagger 2} e^{h^\dagger}}{2\gamma} \right)^{K^\dagger - 1} \frac{5 L_{\boldsymbol{s}}^2 h^\dagger e^{(3+\gamma) h^\dagger}}{2\gamma} \sup_{\tau^\dagger \in [0, h^\dagger]} \mathbb{E}_P \left[ \left\| \widetilde{\boldsymbol{u}}_{t_n, n^\dagger h^\dagger, \tau^\dagger}^{(K^\dagger - 1)} - \widetilde{\boldsymbol{u}}_{t_n, n^\dagger h^\dagger, \tau^\dagger}^{(K^\dagger)} \right\|^2 \right],$$

where we plug in the results from Lemma C.15 in the last inequality. Rearranging the inequality above completes our proof. □

**Theorem C.17.** *Under Assumptions 3.1', 3.2, 3.3, and 3.4, given the following choices of the order of the parameters*

$$T^\dagger = \mathcal{O}(1), \quad N^\dagger = \mathcal{O}(1), \quad h^\dagger = \Theta(1)$$

$$M^\dagger = \Theta(d^{1/2} \delta^{-1}), \quad \epsilon^\dagger = \Theta(d^{-1/2} \delta), \quad K^\dagger = \mathcal{O}(\log(d\delta^{-2}))$$

*and let*

$$\frac{L_{\boldsymbol{s}}^2 h^{\dagger 2} e^{h^\dagger}}{2\gamma} \ll 1, \quad \gamma \lesssim L_p^{-1/2}, \quad T^\dagger \lesssim L_p^{-1/2} \wedge L_{\boldsymbol{s}}^{-1/2}, \quad \delta_\infty \lesssim \delta$$

*then the distribution $\widetilde{\pi}_{t_n, T^\dagger}$ satisfies the following error bound:*

$$D_{\mathrm{KL}}(\pi_{t_n, T^\dagger} \| \widetilde{\pi}_{t_n, T^\dagger}) \lesssim T^\dagger W_2^2(\widetilde{q}_{t_n, h_n}, \breve{p}_{t_{n+1}}) + T^\dagger \delta_\infty^2 + d T^\dagger \epsilon^{\dagger 2} + e^{-K^\dagger} T^\dagger h^\dagger d$$

$$\lesssim W_2^2(\widetilde{q}_{t_n, h_n}, \breve{p}_{t_{n+1}}) + \delta^2,$$

*with a total of $K^\dagger N^\dagger = \mathcal{O}\left( \log(d\delta^{-2}) \right)$ approximate time complexity and $M = \Theta\left( d^{1/2} \delta^{-2} \right)$ space complexity for parallelizable $\delta$-accurate score function computations.*

*Proof.* Now, we continue the computation by plugging the decomposition in (C.24) and all the error bounds derived above into the equation. First for the last term in (C.23)

$$
\mathbb{E}_P\left[\int_0^{h^\dagger}\left\|\boldsymbol{s}_{t_{n+1}}(\widetilde{\boldsymbol{u}}_{t_n,n^\dagger h^\dagger,\lfloor\frac{\tau^\dagger}{\epsilon^\dagger}\rfloor\epsilon^\dagger}^{(K^\dagger-1)})-\nabla\log\breve{p}_{t_{n+1}}(\widetilde{\boldsymbol{u}}_{t_n,n^\dagger h^\dagger,\tau^\dagger}^{(K^\dagger)})\right\|^2\mathrm{d}\tau^\dagger\right]
$$

$$
\leq 5L_{\boldsymbol{s}}^2 h^\dagger\sup_{\tau^\dagger\in[0,h^\dagger]}\mathbb{E}_P\left[\left\|\widetilde{\boldsymbol{u}}_{t_n,n^\dagger h^\dagger,\lfloor\frac{\tau^\dagger}{\epsilon^\dagger}\rfloor\epsilon^\dagger}^{(K^\dagger-1)}-\widetilde{\boldsymbol{u}}_{t_n,n^\dagger h^\dagger,\lfloor\frac{\tau^\dagger}{\epsilon^\dagger}\rfloor\epsilon^\dagger}^{(K^\dagger)}\right\|^2\right]+5\mathbb{E}_P\left[E_{t_n,n^\dagger h^\dagger}+h^\dagger\delta_\infty^2+L_p^2 F_{t_n,n^\dagger h^\dagger}\right]
$$

$$
\leq 5\left(1+L_{\boldsymbol{s}}^2 h^\dagger C_{K^\dagger}\frac{5h^\dagger e^{(3+\gamma)h^\dagger}}{2\gamma}\right)\mathbb{E}_P\left[E_{t_n,n^\dagger h^\dagger}+h^\dagger\delta_\infty^2+L_p^2 F_{t_n,n^\dagger h^\dagger}\right]
$$

$$
+5L_{\boldsymbol{s}}^2 h^\dagger C_{K^\dagger}\left(h^{\dagger 2}e^{(3+\gamma)h^\dagger}\left(3\gamma d+M_{\boldsymbol{s}}^2\right)+h^\dagger e^{2h^\dagger}d\right),
$$

where the last inequality is by Lemma C.16. We further substitute Lemma C.14 and C.13 into (C.23) to obtain

$$
D_{\mathrm{KL}}(\pi_{t_n,n^\dagger h^\dagger}\|\widetilde{\pi}_{t_n,n^\dagger h^\dagger})
$$

$$
\leq D_{\mathrm{KL}}(\pi_{t_n,(n-1)h^\dagger}\|\widetilde{\pi}_{t_n,(n-1)h^\dagger})+5\frac{2\gamma+L_{\boldsymbol{s}}^2 C_{K^\dagger}5h^{\dagger 2}e^{(3+\gamma)h^\dagger}}{4\gamma^2}\mathbb{E}_P\left[E_{t_n,n^\dagger h^\dagger}+h^\dagger\delta_\infty^2+L_p^2 F_{t_n,n^\dagger h^\dagger}\right]
$$

$$
+\frac{5L_{\boldsymbol{s}}^2 h^\dagger C_{K^\dagger}}{2\gamma}\left(h^{\dagger 2}e^{(3+\gamma)h^\dagger}\left(3\gamma d+M_{\boldsymbol{s}}^2\right)+h^\dagger e^{2h^\dagger}d\right)
$$

$$
\lesssim D_{\mathrm{KL}}(\pi_{t_n,(n-1)h^\dagger}\|\widetilde{\pi}_{t_n,(n-1)h^\dagger})
$$

$$
+5\frac{2\gamma+L_{\boldsymbol{s}}^2 C_{K^\dagger}5h^{\dagger 2}e^{(3+\gamma)h^\dagger}}{4\gamma^2}\left(h^\dagger(L_{\boldsymbol{s}}^2+L_p^2)W_2^2(\widetilde{q}_{t_n,h_n},\breve{p}_{t_{n+1}})+h^\dagger\delta_\infty^2+dh^\dagger\epsilon^{\dagger 2}\right)
$$

$$
+\frac{5L_{\boldsymbol{s}}^2 h^\dagger C_{K^\dagger}}{2\gamma}\left(h^{\dagger 2}e^{(3+\gamma)h^\dagger}\left(3\gamma d+M_{\boldsymbol{s}}^2\right)+h^\dagger e^{2h^\dagger}d\right)
$$

$$
\lesssim D_{\mathrm{KL}}(\pi_{t_n,(n-1)h^\dagger}\|\widetilde{\pi}_{t_n,(n-1)h^\dagger})+h^\dagger W_2^2(\widetilde{q}_{t_n,h_n},\breve{p}_{t_{n+1}})+h^\dagger\delta_\infty^2+dh^\dagger\epsilon^{\dagger 2}+e^{-K^\dagger}h^{\dagger 2}d,
$$

and then sum over $n$ to obtain

$$
D_{\mathrm{KL}}(\pi_{t_n,T^\dagger}\|\widetilde{\pi}_{t_n,T^\dagger})=D_{\mathrm{KL}}(\pi_{t_n,N^\dagger h^\dagger}\|\widetilde{\pi}_{t_n,N^\dagger h^\dagger})
$$

$$
\lesssim D_{\mathrm{KL}}(\pi_{t_n,0}\|\widetilde{\pi}_{t_n,0})+N^\dagger h^\dagger W_2^2(\widetilde{q}_{t_n,h_n},\breve{p}_{t_{n+1}})+N^\dagger h^\dagger\delta_\infty^2+dN^\dagger h^\dagger\epsilon^{\dagger 2}+e^{-K^\dagger}N^\dagger h^{\dagger 2}d
$$

$$
=T^\dagger W_2^2(\widetilde{q}_{t_n,h_n},\breve{p}_{t_{n+1}})+T^\dagger\delta_\infty^2+dT^\dagger\epsilon^{\dagger 2}+e^{-K^\dagger}T^\dagger h^\dagger d.
$$

Then, it is straightforward to see that when the following order of the parameters holds

$$
T^\dagger=\mathcal{O}(1),\quad h^\dagger=\Theta(1),\quad N^\dagger=\mathcal{O}(1),
$$

$$
\epsilon^\dagger=\Theta(d^{-1/2}\delta),\quad M^\dagger=\mathcal{O}(d^{1/2}\delta^{-1}),\quad K^\dagger=\mathcal{O}(\log(d\delta^{-2}))
$$

and $\delta_\infty\leq\delta$, we have

$$
D_{\mathrm{KL}}(\pi_{t_n,T^\dagger}\|\widetilde{\pi}_{t_n,T^\dagger})\lesssim W_2^2(\widetilde{q}_{t_n,h_n},\breve{p}_{t_{n+1}})+\delta^2.
$$

$\square$

**Lemma C.18.** *Suppose $T^\dagger\lesssim L_p^{-1/2}$, then we have*

$$
\mathrm{TV}(\pi_{t_n,T^\dagger},\breve{p}_{t_{n+1}})\leq\sqrt{D_{\mathrm{KL}}(\pi_{t_n,T^\dagger}\|\breve{p}_{t_{n+1}})}\lesssim\frac{1}{L_p^{\frac{1}{4}}(T^\dagger)^{\frac{3}{2}}}W_2(\pi_{t_n,0},\breve{p}_{t_{n+1}})\lesssim W_2(\widetilde{q}_{t_n,h_n},\breve{p}_{t_{n+1}}).
$$

*Proof.* A complete proof of the Lemma above is presented in [111, Lemma 9], which is derived based on [145, Corollary 4.7 (1) ].

$\square$

## C.4 Overall Error Bound

We are now ready to prove Theorem 3.5.

*Proof of Theorem 3.5.* Notice that the interpolating corrector process $(\widetilde{\boldsymbol{u}}_{t_n, n^\dagger h^\dagger, \tau^\dagger}, \widetilde{\boldsymbol{v}}_{t_n, n^\dagger h^\dagger, \tau^\dagger})$ is constructed to follow the same dynamics as the auxiliary corrector process $(\widehat{\boldsymbol{u}}_{t_n, n^\dagger h^\dagger, \tau^\dagger}, \widehat{\boldsymbol{v}}_{t_n, n^\dagger h^\dagger, \tau^\dagger})$ in the corrector step. Therefore, we have by data processing inequality that

$$\mathrm{TV}(\widehat{\pi}^{\widehat{\boldsymbol{u}}}_{t_n, T^\dagger}, \widetilde{\pi}^{\widetilde{\boldsymbol{u}}}_{t_n, T^\dagger}) \leq \mathrm{TV}(\widehat{\pi}^{\widehat{\boldsymbol{u}}}_{t_n, 0}, \widetilde{\pi}^{\widetilde{\boldsymbol{u}}}_{t_n, 0}) = \mathrm{TV}(\widehat{q}_{t_n, h_n}, \widetilde{q}_{t_n, h_n}), \tag{C.32}$$

and again, since the interpolating predictor process $\widetilde{\boldsymbol{y}}_{t_n, n^\dagger h^\dagger}$ is constructed to follow the same dynamics as the auxiliary predictor process $\widehat{\boldsymbol{y}}_{t_n, n^\dagger h^\dagger}$ in the predictor step, we further have by data processing inequality that

$$\mathrm{TV}(\widehat{q}_{t_n, h_n}, \widetilde{q}_{t_n, h_n}) \leq \mathrm{TV}(\widehat{q}_{t_n, 0}, \widetilde{q}_{t_n, 0}) = \mathrm{TV}(\widehat{q}_{t_n}, \breve{p}_{t_n}). \tag{C.33}$$

Furthermore, applying triangle inequality, Pinsker's inequality along with Theorem C.17 and Theorem C.7 proved above, we may upper bound the second term above as follows

$$\mathrm{TV}(\pi_{t_n, T^\dagger}, \widetilde{\pi}_{t_n, T^\dagger})^2 \lesssim D_{\mathrm{KL}}(\pi_{t_n, T^\dagger} \| \widetilde{\pi}_{t_n, T^\dagger}) \lesssim W_2^2(\widetilde{q}_{t_n, h_n}, \breve{p}_{t_{n+1}}) + \delta^2 \tag{C.34}$$

Summarizing the above inequalities, we have

$$\begin{aligned}
\mathrm{TV}(\widetilde{\pi}^{\widetilde{\boldsymbol{u}}}_{t_n, T^\dagger}, \breve{p}_{t_{n+1}})^2 &= \mathrm{TV}(\widetilde{\pi}^{\widetilde{\boldsymbol{u}}}_{t_n, T^\dagger}, \pi^{*, \boldsymbol{u}^*}_{t_n, T^\dagger})^2 \\
&\leq \mathrm{TV}(\widetilde{\pi}^{\widetilde{\boldsymbol{u}}}_{t_n, T^\dagger}, \pi^{\boldsymbol{u}}_{t_n, T^\dagger})^2 + \mathrm{TV}(\pi^{\boldsymbol{u}}_{t_n, T^\dagger}, \pi^{*, \boldsymbol{u}^*}_{t_n, T^\dagger})^2 \\
&\leq \mathrm{TV}(\widetilde{\pi}_{t_n, T^\dagger}, \pi_{t_n, T^\dagger})^2 + \mathrm{TV}(\pi_{t_n, T^\dagger}, \pi^*_{t_n, T^\dagger})^2 \\
&\leq \mathrm{TV}(\widetilde{\pi}_{t_n, T^\dagger}, \pi_{t_n, T^\dagger})^2 + \mathrm{TV}(\pi_{t_n, T^\dagger}, \breve{p}_{t_{n+1}})^2 \\
&\lesssim W_2^2(\widetilde{q}_{t_n, h_n}, \breve{p}_{t_{n+1}}) + \delta^2 + W_2^2(\widetilde{q}_{t_n, h_n}, \breve{p}_{t_{n+1}}) \\
&\lesssim d e^{-K} + h_n^2 \delta_\infty^2 + d\epsilon^2 h_n^2 + \delta^2,
\end{aligned} \tag{C.35}$$

where the second last inequality is deduced from Theorem C.17 and Lemma C.18 and the last inequality is derived via Theorem C.7. Therefore, for any $n \in [0 : N-1]$, applying triangle inequality along with data processing inequality (*cf.* Theorem A.1) yields

$$\begin{aligned}
\mathrm{TV}(\widehat{q}_{t_{n+1}}, \breve{p}_{t_{n+1}}) &= \mathrm{TV}(\widehat{\pi}^{\widehat{\boldsymbol{u}}}_{t_n, T^\dagger}, \breve{p}_{t_{n+1}}) \\
&\leq \mathrm{TV}(\widehat{\pi}^{\widehat{\boldsymbol{u}}}_{t_n, T^\dagger}, \widetilde{\pi}^{\widetilde{\boldsymbol{u}}}_{t_n, T^\dagger}) + \mathrm{TV}(\widetilde{\pi}^{\widetilde{\boldsymbol{u}}}_{t_n, T^\dagger}, \breve{p}_{t_{n+1}}) \\
&\leq \mathrm{TV}(q_{t_n}, \breve{p}_{t_n}) + d^{1/2} e^{-K/2} + h_n \delta_\infty + d^{1/2} \epsilon h_n + \delta.
\end{aligned} \tag{C.36}$$

where the last inequality is derived by plugging in (C.32), (C.33) and (C.36). Applying Lemma A.9 and summing the inequalities above further give us that

$$\begin{aligned}
\mathrm{TV}(\widehat{q}_{t_N}, p_\eta) &= \mathrm{TV}(\widehat{q}_{t_N}, \breve{p}_{t_N}) \\
&\lesssim \mathrm{TV}(\widehat{q}_0, \breve{p}_0) + \sum_{n=0}^{N-1} \left( d^{1/2} e^{-\frac{K}{2}} + h_n \delta + d^{1/2} \epsilon h_n + \delta \right) \\
&\lesssim d^{1/2} e^{-T/2} + N d^{1/2} e^{-K/2} + T \delta_\infty + d^{1/2} \epsilon T + \delta N.
\end{aligned} \tag{C.37}$$

By setting the parameters

$$T = \mathcal{O}(\log(d\delta^{-2})), \quad h = \Theta(1), \quad N = \mathcal{O}(\log(d\delta^{-2})),$$

$$\epsilon = \Theta\left(d^{-1/2} \delta \log^{-1}(d^{-1/2}\delta^{-1})\right), \quad M = \mathcal{O}(d^{1/2}\delta^{-1} \log(d^{1/2}\delta^{-1})), \quad K = \widetilde{\mathcal{O}}(\log(d\delta^{-2})),$$

and letting $\delta_\infty \lesssim \delta T^{-1} \lesssim \delta \log^{-1}(d\delta^{-2})$, we finally obtained the upper bound

$$\mathrm{TV}(\widehat{q}_{t_N}, p_\eta)^2 \lesssim d e^{-T} + N^2 d e^{-K} + \delta^2 + d\epsilon^2 T^2 \leq \delta^2$$

as desired. $\qquad\square$

