# OpenReview forum: "Accelerating Diffusion Models with Parallel Sampling: Inference at Sub-Linear Time Complexity"
_NeurIPS.cc/2024/Conference — NeurIPS 2024 spotlight_

### Official Review · Reviewer_FgeD · 2024-07-10

**Soundness:** 3
**Presentation:** 2
**Contribution:** 2
**Rating:** 6
**Confidence:** 4

**Summary:**

This work improves the sample complexity of diffusion models by using the parallel sampling technique and achieves $\tilde{O}(\text{poly} \log d)$ results for reverse SDE and PFODE settings at the same time. To achieve these results, this work provides a general version of Girsanov’s theorem to deal with the additional dependence introduced by the Picard iteration.

**Strengths:**

1.	The general version of Girsanov’s theorem is novel and suitable for the parallel sampling setting, which would raise independent interest.

**Weaknesses:**

1. For the PIADM type algorithm, the framework is similar to Algorithm 2 of [1], which is used to analyze ULMC. It seems that this work replaces the ULMC process with the  denoised process. Then, this work relaxes the LSI assumption of data by using a similar technique to [2]. It would be helpful to discuss the technique challenge except for the general Girsanov’s theorem.

2. For the PFODE setting, the PFODE-ULMC type algorithm is introduced and analyzed by [3]. Using this technique, [3] achieve a similar $\sqrt{d}$ improvement. It would be better to discuss the technique novelty compared to [3].

[1] Anari, N., Chewi, S., & Vuong, T. D. (2024). Fast parallel sampling under isoperimetry. arXiv preprint arXiv:2401.09016.

[2] Benton, J., Bortoli, V. D., Doucet, A., & Deligiannidis, G. (2024). Nearly d-linear convergence bounds for diffusion models via stochastic localization.

[3] Chen, S., Chewi, S., Lee, H., Li, Y., Lu, J., & Salim, A. (2024). The probability flow ode is provably fast. Advances in Neural Information Processing Systems, 36.

**Questions:**

Please see the Weakness part.

**Limitations:**

The authors adequately addressed the limitations and societal impact.

---

> ### Author Rebuttal · Authors · 2024-08-06
>
> We would like to thank the reviewer’s comments and suggestions on our work. In the following paragraphs, we address the reviewer’s concerns.
>
> ---
>
> ### Regarding comparison with [1, 2]
> We appreciate the reviewer’s suggestions to deepen the discussions on our results relative to those in [1, 2]. Along with the comparisons already detailed in the second bullet point of Remark 3.4, we wish to offer the following additional remarks:
>
> In contrast to [1], which limits parallel sampling techniques to log-concave sampling within Langevin dynamics, our approach significantly applies these techniques to denoising diffusion models - a topic of widespread interest within the ML community. To the best of our knowledge, our work is the first to provide parallel algorithms for diffusion models that are rigorously analyzed and proven to achieve $O( \mathrm{poly} \log d)$ time complexity.
>
> As aptly pointed out by the reviewer, unlike the strategies in [1], our analysis does not rely on LSI-type assumptions on the data distribution, making it applicable to almost any distributions of interest, given a sufficiently accurate score approximation. This flexibility underpins both the empirical success of diffusion models [8] and the theoretical advancements in [2, 3] - further validating our results under the parallel sampling framework.
>
> We would like to emphasize that the aim of our work is not merely the sophisticated generalization and application of the Girsanov’s theorem, which the reviewer recognized as “novel and suitable”, and “raises independent interest”, but also its coordinated integration into the algorithmic design of denoising diffusion models that answers the open question of the benefit of parallel sampling theoretically in this context.
>
> Although primarily theoretical, our PIADM-SDE algorithm remains practically feasible, and our analysis incorporates practical inference strategies such as exponential integrator, early stopping, etc., which hold promise for real-world applications, a potential recognized by Reviewer sp2J. Further discussion on PIADM-ODE provides possible directions for reducing storage complexity while preserving a sub-linear rate. We will refine our paper to more clearly articulate these contributions.
>
> ### Regarding comparison with [3]
> We appreciate the reviewer’s question regarding our work’s comparison with [3]. As aforementioned, our exploration of PIADM-ODE, building upon our PIADM-SDE findings, aims to investigate the potential reduction in storage requirements by replacing the SDE with the probability flow ODE formulation, which has been demonstrated as “provably faster” in terms of $d$ in [3].
>
> As discussed in Section 3.2.1 (Algorithm) and Section 3.2.4 (Proof Sketch), our analysis addresses not only the absence of a data processing inequality for 2-Wasserstein distance as in [3], but also the $O(\sqrt{d})$ time complexity introduced by a naive implementation of the underdamped Langevin corrector, which would compromise our sublinear time efficiency. Leveraging techniques, such as the generalized Girsanov’s Theorem initially developed for our PIADM-SDE, we successfully mitigate these challenges. This allows us to show that the improvement of time efficiency from $O(d)$ to $O(\sqrt{d})$ with probability flow ODE in [3] can be adapted to reduce storage complexity from $O(d^2)$ to $O(d^{3/2})$ in our parallel algorithms, as detailed in Theorem 3.5.
>
> We believe our results not only echo those found in [3] but also introduce significant additional technical contributions. We will refine our paper to further elaborate on these points in the revised version.
>
> ---
>
> Finally, we sincerely thank the reviewer for their insightful comments. Should our clarifications and modifications address their concerns, we would be grateful if the reviewer could possibly consider reevaluation of our work based on our theoretical contributions and we are more than happy to answer any follow-up questions.
>
> ### References
> [1] Anari, N., Chewi, S., & Vuong, T. D. (2024). Fast parallel sampling under isoperimetry. arXiv preprint arXiv:2401.09016.
>
> [2] Benton, J., Bortoli, V. D., Doucet, A., & Deligiannidis, G. (2024). Nearly d-linear convergence bounds for diffusion models via stochastic localization.
>
> [3] Chen, S., Chewi, S., Lee, H., Li, Y., Lu, J., & Salim, A. (2024). The probability flow ode is provably fast. Advances in Neural Information Processing Systems, 36.
>
> [4] Andy Shih, Suneel Belkhale, Stefano Ermon, Dorsa Sadigh, and Nima Anari. Parallel sampling of diffusion models. Advances in Neural Information Processing Systems, 36, 2024.
>
> [5] Chen, H., Lee, H., & Lu, J. (2023, July). Improved analysis of score-based generative modeling: User-friendly bounds under minimal smoothness assumptions. In International Conference on Machine Learning (pp. 4735-4763). PMLR.
>
> [6] Lee, H., Lu, J., & Tan, Y. (2022). Convergence for score-based generative modeling with polynomial complexity. Advances in Neural Information Processing Systems, 35, 22870-22882.
>
> [7] Lee, H., Lu, J., & Tan, Y. (2023, February). Convergence of score-based generative modeling for general data distributions. In International Conference on Algorithmic Learning Theory (pp. 946-985). PMLR.
>
> [8] Yang, Ling, et al. "Diffusion models: A comprehensive survey of methods and applications. arXiv 2022." arXiv preprint arXiv:2209.00796 (2022).

---

> > ### Comment · Reviewer_FgeD · 2024-08-08
> >
> > Thanks for the detailed discussion on the previous works. It would be helpful to add these discussions and highlight the contribution of the technique in the next version. Since this discussion addresses my concerns, I will raise my score to $6$.

---

> > > ### Author Response · Authors · 2024-08-08
> > > **Thank you so much!**
> > >
> > > We would like to thank the reviewer again for your time and efforts! Your insightful suggestions and comments have greatly helped improve the quality of our manuscript. In case you have any other question, please don't hesitate to let us know.

---

### Official Review · Reviewer_jWja · 2024-07-11

**Soundness:** 3
**Presentation:** 3
**Contribution:** 3
**Rating:** 7
**Confidence:** 4

**Summary:**

In this article, the authors propose parallel methods for diffusion models, achieving a poly-logarithmic complexity for the first time under commonly used assumptions. By applying existing Picard Iterations to both the SDE and probability flow ODE of diffusion models, the backward process can be efficiently solved within a period of $O(1)$ length, effectively transferring sequential computations of $O(d)$ into parallelizable iterations with a depth of $O(\log d)$. However, the author's analysis is not trivial. They employ a more intricate mathematical framework for stochastic processes and general forms of Girsanov's theorem to bound the error in each block.

**Strengths:**

- I believe this paper is technically solid and addresses an important open question regarding parallel methods for diffusion models.
- This paper is well-written, and the proof is clear and comprehensive.
- I think that the sketches and ideas presented in the proof will be useful for future research, although it is similar to the parallel method used in sampling.

**Weaknesses:**

- The authors apply Picard iterations to diffusion models without providing a comparison to other parallel methods, such as randomized midpoint methods. Including such comparisons would enhance the evaluation of their approach.
- Please compare the total query complexity of the proposed method to that of sequential methods.

Minor:
- Line 85: Please add a reference for the Picard iteration.
- Line 89: Please include a reference for the claim of exponentially fast convergence.
- Last equation on page 4: Specify the distribution that is being assumed for this equation.
- Lemma A.6: Ensure that $_F$ is included.
- Provide an explanation regarding the different parameter choices for the predictor and corrector in Theorem 3.5.

**Questions:**

See Weaknesses

---

> ### Author Rebuttal · Authors · 2024-08-06
>
> We are grateful for the reviewer’s insightful comments and appreciation for our work. We have listed our answers to the questions raised in the review as follows:
>
> ---
>
> ### Regarding comparison with other parallel sampling methods
> We appreciate the reviewer’s suggestion regarding the necessity of contrasting the Picard iteration and other parallel methods. To address this, we will revise our paper accordingly, particularly enhancing the introduction and expanding the discussion in Section 2.2 (Parallel Sampling). Our aim is to provide a more comprehensive outline of parallel methods developed for various tasks and to elucidate the rationale behind selecting the Picard iteration for our study.
>
> Specifically for the randomized midpoint method [1, 2], we acknowledge its potential for achieving better query complexity under additional isoperimetric assumptions on the target density, such as the logarithmic Sobolev inequality. This observation will guide our future research efforts. We will incorporate these insights and recent advances along this research direction into our revised paper, ensuring a thorough evaluation and justification of the methodologies employed in our work.
> ### Regarding total query complexity
> We thank the reviewer for the suggestion. In response, we have included a remark in Section 3 that compares the total query complexity of our proposed methods (PIADM-SDE and PIADM-ODE) with their sequential counterparts. We wish to clarify that the core idea behind our acceleration approach is to offset the number of sequential evaluations with parallel iterations. Leveraging the exponential convergence properties of Picard iterations, our algorithm manages to achieve $O(\mathrm{poly} \log d)$ time complexity, at the cost of a potentially higher total number of score queries. We will make additional revisions to our paper to further elucidate this aspect.
> ### Regarding minor issues
> We appreciate the reviewer’s meticulous attention in identifying the typos and missing references (on line 85, on line 89, Lemma A.6) and have accordingly revised our paper.
> - For the last equation on page 4, we have inserted an explanatory line below equation (2.2) to more clearly define the true density associated with the reversed SDE. We have also expanded the “Step Size Scheme” paragraph in Section 3.1.1 to enhance understanding of Lemma A.6, A.7, A.8, along with their corollary, Lemma B.7 on which the equation in the reviewer’s equestion is based.
> - Regarding the two sets of parameter choices for the predictor and corrector presented in Theorem 3.5, we have added a paragraph dedicated to elaborate on the reasoning behind our parameter selections, including $h$, $\epsilon$, $M$, $K$, etc.. We specifically address the conditions that allow for the block size $h$ to be of constant order, how the magnitude of the step size $\epsilon$ aligns with that in previous work [3], and the relationships between the scaling of $T$ in the predictor step and $T^\dagger$ in the corrector, etc., within the ODE formulation.
>
> We believe these revisions will significantly enhance the clarity and coherence of the presentation of our paper.
>
> ---
>
> Finally, we sincerely thank the reviewer for all your time and efforts. Your suggestions have tremendously helped us improve the presentation of our paper.
>
> ### References
> [1] Shen, R., & Lee, Y. T. (2019). The randomized midpoint method for log-concave sampling. Advances in Neural Information Processing Systems, 32.
>
> [2] Yu, L., & Dalalyana, A. (2024). Parallelized midpoint randomization for langevin monte carlo. arXiv preprint arXiv:2402.14434.
>
> [3] Joe Benton, Valentin De Bortoli, Arnaud Doucet, and George Deligiannidis. Linear convergence bounds for diffusion models via stochastic localization. arXiv preprint arXiv:2308.03686, 2023.

---

### Official Review · Reviewer_sp2J · 2024-07-12

**Soundness:** 4
**Presentation:** 4
**Contribution:** 4
**Rating:** 8
**Confidence:** 4

**Summary:**

Denoising diffusion models generate samples from a complex target distribution by solving stochastic/ordinary differential initialized at the Gaussian noise. Solving these differential equations is typically done by simulating diffusion solutions. This is inherently sequential since simulating the solutions at time t_i requires the solution at time t_{i-1}.

An alternative approach to solve these diffusions is to use Picard Iterations (PI), which involve constructing a contraction mapping where the solution is the fixed point. Notably, a PI iteration updates the entire path, making parallelization amenable.

This paper introduces how to sample from a diffusion model incorporating PI into the pipeline, allowing practitioners to utilize their parallel processes efficiently. They show that a PI can speed up the run time in O(poly(log d)) in dimension compared to O(d) runtime of serial implementations of diffusion models.

**Strengths:**

I think the topic is well-motivated, and the authors do a great job of presenting the ideas in the paper. The idea feels quite natural and the algorithms developed have practical utility, which is uncommon for a theory paper. The theory is also strong, I can tell a lot of care has gone into the proofs.

**Weaknesses:**

I understand it is a theory paper, but I was disappointed to see no experiments, or empirical verification of the theory. The main result is the poly(log(d)) run-time, which would have been nice to see how this hold in practice. The algorithms presented in this paper seem fairly easy to implement. Experimenting with even a Gaussian toy model where analytical solutions are known would have made this paper more convincing to practitioners.

**Questions:**

Can you provide some intuition on how the constants scale, and how long does it take for the asymptotic to kick in?

Can you provide some intuition on how to tune hyperparameters, including a number of blocks, discretization steps per block, and depth?

**Limitations:**

The authors adequately addressed the limitations.

---

> ### Author Rebuttal · Authors · 2024-08-06
>
> We thank the reviewer for the appreciation of our work and the valuable feedback. Below are our responses to the questions raised in the review:
>
> ---
> ## Weaknesses
>
> ### Regarding numerical experiments
> We appreciate the reviewer’s suggestion and recognize the value of empirical verification of our theoretical results.
>
> Our decision to focus primarily on theoretical analysis was influenced by existing literature, particularly recent studies in references [1, 2], which have extensively explored and positively demonstrated the acceleration of diffusion model inference through parallel sampling. Therefore, we chose to concentrate on advancing the theoretical discussion to avoid repetition and build directly on these established empirical findings.
>
> Moreover, as mentioned in Section 2.3, the verification of the sublinear acceleration achieved by our proposed parallel algorithms might require large-scale experiments with data dimensions $d \gg \log d$ and thus substantial engineering efforts that could be of independent research interest. In light of this, we will expand Section 2.3 to further discuss the applicability of our results and to clarify the assumptions required for implementation.
>
> ---
>
> ## Questions
> ### Regarding asymptotics
> We thank the reviewer for the insightful question regarding the constant scalings and asymptotic behavior.
> In our work, the scaling of constants in the two main theorems (Theorem 3.3 and 3.5) is influenced by the constants ($L_{\boldsymbol{s}}$ and $M_{\boldsymbol{s}}$) specified in Assumption 3.3., These constants are in turn derived from the properties of the ground-truth density from which we aim to generate samples.
>
> It is important to note that the actual onset of asymptotic behavior may depend not only on these constants, but also the specific implementation and the empirical setup. As such, further empirical studies are needed to precisely determine when the asymptotics will take effect under various conditions. We plan to explore this aspect in future work to provide a clearer understanding of the dynamics involved.
>
> ### Regarding hyperparameters
> We appreciate the reviewer’s question concerning the tuning of the hyperparameters. We acknowledge the tuning of hyperparameters, although involving considerable engineering effort, is crucial for the actual performance of the algorithms and may vary significantly depending on specific problems and the desired accuracy of the model. In this context, our theoretical work aims to provide insights into the optimal scaling and relative magnitudes of these hyperparameters, potentially facilitating the tuning process through a more structured approach.
>
> Additionally, several empirical studies [1, 2] have implemented various kinds of parallel algorithms, incorporating advanced tuning techniques. For instance, the algorithm in [1] employs a striding window of adaptive length to tune the step sizes. These implementations serve as excellent references for observing the practical impacts of hyperparameter settings in real-world scenarios. We are eager to further explore and validate our theoretical findings by experimenting with different combinations of these hyperparameters in empirical settings in subsequent research efforts.
>
> ---
>
> Finally, we would like to thank the reviewer for all the helpful comments and suggestions, which have greatly helped us improve the quality of our manuscript.
>
> ### References
> [1] Shih, A., Belkhale, S., Ermon, S., Sadigh, D., & Anari, N. (2024). Parallel sampling of diffusion models. Advances in Neural Information Processing Systems, 36.
>
> [2] Tang, Z., Tang, J., Luo, H., Wang, F., & Chang, T. H. (2024, January). Accelerating parallel sampling of diffusion models. In Forty-first International Conference on Machine Learning.

---

### Official Review · Reviewer_GP1D · 2024-07-15

**Soundness:** 3
**Presentation:** 4
**Contribution:** 3
**Rating:** 7
**Confidence:** 4

**Summary:**

In this paper, the authors propose to analyse the error in parallel sampling for diffusion models. This represents the first theoretical analysis of parallel sampling for diffusion models. The initial methodology was proposed in ParaDiGMS [1]. In this paper, the authors propose some incremental improvements on the samplers used in ParaDiGMS. The main contribution resides in the control of the error in parallel sampling. Notably, instead of deriving a global time complexity which depends on $O(d)$ they derive rates that are of order $O(\mathrm{poly log}(d))$. This improvement comes from the parallel sampling nature of the scheme they analyse. The authors not only deal with the SDE (Stochastic Differential Equation) sampler but also with ODE (Ordinary Differential Equation) samplers. Parallel sampling for ULA (Unadjusted Langevin Algorithm) was already theoretically analysed in [2] but to the best of my knowledge this is the first rigorous work on the complexity of parallel sampling for diffusion models.

[1] Shih et al. (2024) -- Parallel Sampling of diffusion models

[2] Anari et al. (2024) -- Fast parallel sampling under isoperimetry

**Strengths:**

* This paper is very well-written. I found the flow of the paper to be quite easy to follow.

* I have read the proofs in the case of the SDE samplers and they are correct. I also commend the authors on adding a proof sketch in the main paper. This makes the whole paper much easier to read and the proofs become more intuitive.

* The rates obtained by the authors are quite compelling. In particular the $O(\mathrm{poly log}(d))$. time complexity echoes the rates obtained in [1].

* The authors not only analyse the SDE sampler but also the ODE sampler which is usually more complicated than the SDE one. I am pleasantly surprised by the gains they obtain in terms of memory complexity.

[1] Anari et al. (2024) -- Fast parallel sampling under isoperimetry

**Weaknesses:**

* My main complain is that the results of Theorem 3.3 and Theorem 3.5 are given in terms of a given level of optimisation $\delta^$. This gives a specific choice of constant for the discretisation stepsize $\epsilon$, the time $T$, the block size $h$, the number of blocks $N$, the number of parallel iterations $K$ and $M$ the number of steps in each block. However, it would also be interesting to give a bound that depends on those quantities. What I am thinking about is that then one could optimize the bound under time complexity and space complexity constraints (for example with a requirement that $d M \leq M_0$ where $M_0$ is the available memory of the system). Such a bound would be more interesting to gain insights on what we really gain by moving to parallel sampling.

* While I'm interested in the ODE framework I am a bit confused by the explanation on how the authors have managed to reduce the $O(d^2)$ memory complexity to $O(d^{3/2})$. The explanation of l.281 "The reduction of space complexity by the probability flow ODE implementation is intuitively owing to the fact that the probability flow ODE process is a deterministic process in time rather than a
stochastic process as in the SDE implementation." is not very intuitive to me. It would be nice to point out where the improvement comes from in the proof sketch (3.2.4).

* Regarding the ODE sampler, I am not sure that the bounds with SDE are directly comparable since the proof techniques use different metrics (even though the KL can be related to the TV using Pinsker's inequality). To  get comparable rates it would have been better to provide a unified framework in which both the SDE and the ODE samplers are analysed under the same techniques (and assumptions).

* Assumption 3.4 is extremely strong and not satisfied in most applications. It seems that compared to [1] which analyses ODE samplers, Assumption 3.4 is not required there. Similarly for Assumption 3.3. These assumptions prevent the model to explode near the data which is something that is observed in practice (and would happen theoretically). I would have liked a more detailed comparisons between the differences of assumptions required for the analysis of sequential and parallel samplers of diffusion models.

[1] Chen et al. (2023) -- "Restoration-Degradation Beyond Linear Diffusions: A Non-Asymptotic Analysis For DDIM-Type Samplers"

**Questions:**

* "We propose new parallel inference algorithms for diffusion models using parallel sampling" (in the TLDR). It seems that one of the claim of the authors is that they also propose a new sampler. Compared to ParaDiGMS [1], the methodological innovation seems pretty incremental (exponential integrator, shrinking step size, early stopping, see Section 1.1 "Contributions"). Is there something more? If there is a key methodological contribution, this should be assessed rigorously against ParaDiGMS in experimental settings. I realize that this is a theoretical paper so I would suggest to tame down the claims of methodological novelty and instead focus on the theoretical analysis.

* I would be interested to see if the improvements obtained by the authors can be translated to the manifold hypothesis setting of [2, 3]. In particular, can the $O(\mathrm{poly log}(d))$ rate be kept for this setting and/or improved to replace the dimension of the space by the dimension of the implicit manifold?

* See "Weaknesses" for other questions and concerns.

[1] Shih et al. (2024) -- Parallel Sampling of diffusion models

[2] De Bortoli (2022) -- Convergence of denoising diffusion models under the manifold hypothesis

[3] Yang et al. (2023) -- The Convergence of Variance Exploding Diffusion Models under the Manifold Hypothesis

**Limitations:**

In the paper the limitations are discussed in Section 4, i.e. "Although we anticipate implementing diffusion models in parallel may introduce engineering challenges, e.g. scalability, hardware compatibility, memory bandwidth, etc., we believe that our theoretical contributions lay a solid foundation that not only supports but also motivates the empirical development of parallel inference algorithms for diffusion models since advancements continue in GPU power and memory efficiency." I think there are other limitations that need to be discussed regarding the current work and the theoretical assumptions that the authors needed to make in order to derive their results. While I don't think these limitations hinder the results of the paper they should be clearly laid out.

---

> ### Author Rebuttal · Authors · 2024-08-06
>
> We thank the reviewer for the detailed feedback, constructive suggestions, and kind affirmation of our work. We would like to address the reviewer’s comments in a one-by-one manner below.
>
> ---
>
> ## Weaknesses
> ### Regarding the presentation of results
> We thank the reviewer’s suggestions regarding the presentation of results in Theorem 3.3 and Theorem 3.5. Acknowledging your suggestion, we will revise our paper to include such a bound with explicit dependencies on the step size $\epsilon$, the number of iterations $K$, etc., enhancing the utility and applicability of our results.
>
> ### Regarding memory reduction with ODE formulation
> Thank you for pointing out this confusion. Here is an example to illustrate the intuition behind the acceleration of ODE-based samplers: Consider a 1D Itô process $dx_t = b(x_t) \ dt + \sqrt{2\sigma} \ dw_t,
> $ and by Itô's formula for $f \in C^2$, we obtain
> $$f(x_t) - f(x_0) = \int \left( f'(x_t) b(x_t) + \sigma f''(x_t) \right) \ dt + \int f'(x_t) \sqrt{2\sigma} \ dw_t,$$
> and thus
> $$\mathbb{E} \left[ f(x_t) - f(x_0) \right]^2 \lesssim \mathbb{E} \left[ \int \left( f'(x_t) b(x_t) + \sigma f''(x_t) \right) \ dt \right]^2 + \mathbb{E} \left[ \int f'(x_t) \sqrt{2\sigma} \ dw_t \right]^2 \leq t \mathbb{E} \left[ \int \left( f'(x_t) b(x_t) + \sigma f''(x_t) \right)^2 \ dt \right] + 2\sigma \mathbb{E} \left[ \int f'(x_t)^2 \ dt \right] \sim O(t^2) + \sigma O(t),$$
> where the second-to-last inequality utilizes the Cauchy-Schwarz inequality and Itô's isometry. This derivation suggests that when $\sigma > 0$ (in the case of SDEs), the discretization error with step size $\epsilon$ ($\mathbb{E}[f(x_\epsilon) - f(x_0)]^2$) is of order $O(\epsilon)$, whereas it is of order $O(\epsilon^2)$ when $\sigma = 0$ (for ODEs). The improved step size dependency thus allows for better storage complexity.
>
> For more rigorous arguments, please refer to Lemma B.7 and C.5. We will expand the corresponding discussions in the paper to improve clarity.
>
> ### Regarding the comparability of SDE and ODE formulations
> We appreciate the reviewer’s insightful observation. While ODEs do not permit the application of Girsanov’s theorem, we have to make more restrictive assumptions and some compromises in the algorithm. Notwithstanding, we have tried our best to align the results as closely as possible by giving KL bounds for SDE and TV$^2$ bound for ODE. Indeed, as the reviewer suggests, the bound in Theorem 3.3 could be relaxed to a TV$^2$ bound using just one step of Pinsker’s inequality. Creating a unified framework to analyze both samplers under the same techniques and assumptions would be both desirable and challenging, and we intend to pursue this direction in future work.
>
> ###  Regarding Assumptions
> We acknowledge the reviewer’s comments on the assumptions. We agree with the reviewer that Assumption 3.4 is less favorable when it comes to wider applicability and is possibly due to technicality as we only require it for the ODE formulation. We regard an improvement on our current bound, by replacing Assumption 3.4 with Assumption 1.5 in [1] (arxiv version), as viable. Assumption 3.3, which is implementable by simply adding penalization terms based on the weights of the neural network, is often adopted for numerical stability. We believe the possible blow-up near the data end could be partially averted with an appropriate early stopping time $\eta$ and is of distinct research interest that we would like to study in future work. We will expand the discussions on the assumptions regarding the reviewer’s comments.
>
> ---
>
> ## Questions
> ### Regarding the algorithm compared to ParaDiGMS
> We thank the reviewer for the feedback. We will adjust our paper to more accurately emphasize the theoretical nature of our work, focusing on the specific analytical benefits our methods provide. We believe this will ensure our contributions are properly positioned within the existing literature, and we plan to rigorously assess their practical impacts in future empirical studies.
>
> ### Regarding the manifold hypothesis
> We thank the reviewer for raising this potential improvement with the manifold hypothesis. We believe this hypothesis can be integrated into our framework harmoniously, whereby the $O(\mathrm{poly}\log d)$ rate would be improved to $O(\mathrm{poly}\log \mathrm{diam}(M))$, with $\mathrm{diam}(M)$ being the diameter of the data manifold that only depends on the intrinsic dimension $p$ of the manifold. This would be a significant result, especially for applications in computer vision, where we have $p\ll d$. We will add related discussions in the paper and possibly conduct further theoretical analysis on this aspect in future works.
>
> ---
>
> ## Limitations
> ### Regarding the discussion of limitations
> We appreciate the reviewer for pointing out the necessity of a more comprehensive discussion on the theoretical assumptions underlying our results. We will expand our discussion in Section 3, focusing on how the assumptions might affect the generalizability and applicability of our findings. Noting that the performance of parallel algorithms can be greatly influenced by practical implementation and hardware constraints, we recognize that scaling may deviate from the theory, particularly when unexpected communication and synchronization issues introduce considerable delays. To further elaborate on these issues, we will enhance our discussion in Section 2.3 to more explicitly outline the approximations and assumptions made during our analysis of the time complexity of parallel algorithms.
>
> ---
>
> We thank again for the reviewer’s detailed comments and valuable suggestions. We hope our responses have resolved their concern and our modifications to the paper are up to their expectations.
>
> ### References
> [1] Chen, S., Daras, G., & Dimakis, A. (2023, July). Restoration-degradation beyond linear diffusions: A non-asymptotic analysis for ddim-type samplers. In International Conference on Machine Learning (pp. 4462-4484). PMLR.

---

> > ### Comment · Reviewer_GP1D · 2024-08-12
> >
> > Thanks a lot for your rebuttal. I would like to keep my score. I think the paper is good. Again, I am still a bit confused by the ODE memory reduction (and the provided answer does not really clarify things for me). I guess my question is the following. Do you think that the memory reduction is an artifact of the proof (i.e. some of the bounds are loose in the SDE case) or do you think this is something that could be actually verified experimentally?

---

> > > ### Author Response · Authors · 2024-08-12
> > > **Respone to Reviewer GP1D**
> > >
> > > We would like to thank the reviewer for the response and appreciation for our work. As for now, we can only obtain the memory reduction for the ODE case via mathematical proof. To the best of our knowledge, there has been no empirical work comparing the ODE implementation with the SDE implementation under the parallel sampling framework. We think it would be interesting to compare them to see if such memory reduction can be achieved in future work.

---

### Decision · Program_Chairs · 2024-09-25

**Decision:**

Accept (spotlight)

**Comment:**

This paper presents a novel parallel sampling method for diffusion models, achieving a poly-logarithmic time complexity. The authors leverage Picard Iterations (PI) to solve both the SDE and probability flow ODE associated with diffusion models, enabling efficient parallelization of the sampling process.

The reviews highlight the paper's strengths in addressing a significant problem, presenting a novel approach, and providing a solid theoretical analysis.

However, reviewers suggest that a more detailed comparison with other parallel methods like the randomized midpoint method and ParaDiGMS should be included. Additionally, they were also disappointed with the absence of empirical verification of the theoretical results. While a theory paper, they strongly encourage the inclusion of experiments to demonstrate the practical utility of the proposed methods. I concur with them and strongly recommend the authors include such experiments in the final version of the paper.

Overall, the paper presents an excellent contribution to the field of diffusion models. However, the authors should address carefully the concerns raised by reviewers regarding clarity, empirical validation, and comparison with existing methods while preparing the final version of the paper. This will significantly enhance the paper's impact.